# Structural insights into the interplay between microtubule polymerases, γ-tubulin complexes and their receptors

Anjun Zheng [1,3], Bram J. A. Vermeulen [1,3], Martin Würtz [1,2,3], Annett Neuner[1], Nicole Lübbehusen [1], Matthias P. Mayer [1], Elmar Schiebel [1] & Stefan Pfeffer [1]

The γ-tubulin ring complex (γ-TuRC) is a structural template for controlled nucleation of microtubules from α/β-tubulin heterodimers. At the cytoplasmic side of the yeast spindle pole body, the CM1-containing receptor protein Spc72 promotes γ-TuRC assembly from seven γ-tubulin small complexes (γ-TuSCs) and recruits the microtubule polymerase Stu2, yet their molecular interplay remains unclear. Here, we determine the cryo-EM structure of the *Candida albicans* cytoplasmic nucleation unit at 3.6 Å resolution, revealing how the γ-TuRC is assembled and conformationally primed for microtubule nucleation by the dimerised Spc72 CM1 motif. Two coiled-coil regions of Spc72 interact with the conserved C-terminal α-helix of Stu2 and thereby position the α/β-tubulin-binding TOG domains of Stu2 in the vicinity of the microtubule assembly site. Collectively, we reveal the function of CM1 motifs in γ-TuSC oligomerisation and the recruitment of microtubule polymerases to the γ-TuRC.

Microtubules (MTs) are polar cytoskeletal filaments composed of α/β-tubulin heterodimers with a central role in a range of cellular processes, including chromosome separation, ciliary function, intracellular trafficking and organelle positioning[1]. All of these processes require strict spatiotemporal regulation of MT nucleation, the de novo generation of MTs[2–4]. This is achieved in vivo by γ-tubulin complexes (γ-TuCs)[2], which serve as structural templates during the MT nucleation reaction.

In vertebrates, MTs are nucleated by the γ-tubulin ring complex (γ-TuRC)[5–7], comprising γ-tubulin, five different paralogous γ-tubulin complex proteins (GCP2-6), actin and the small mitotic spindle organizing proteins Mzt1 and Mzt2. Recently, high-resolution cryo-EM reconstructions provided structural insights into the molecular architecture of the vertebrate γ-TuRC[5–7]. These structures revealed a 14-spoked assembly in which the GCP paralogs are presenting γ-tubulin molecules in a helical arrangement. The exposed γ-tubulin molecules recruit α/β-tubulin heterodimers by interacting with the α-tubulin

subunits, thereby realising the function of γ-TuCs as structural templates for MT nucleation.

For ascomycete yeasts, MT nucleation occurs mainly from oligomers of the compositionally much simpler γ-tubulin small complex (γ-TuSC)[8,9]. Cryo-EM structures of γ-TuSCs from *Candida albicans* and *Saccharomyces cerevisiae* revealed the 2-spoked architecture of isolated γ-TuSCs, comprising Spc97 and Spc98 (homologues of human GCP2 and GCP3, respectively), each presenting one molecule of γ-tubulin[10,11]. At the spindle pole body (SPB), the major fungal MT-organising center (MTOC) embedded in the nuclear envelope[12], the γ-TuSC assembles into ring-like oligomers. Cryo-EM studies have investigated the structure and conformational landscape of γ-TuSC rings at the nuclear side of the SPB in *S. cerevisiae*. When reconstituted, the majority of nuclear γ-TuSC oligomers were characterised by an open conformation[13,14], in which γ-tubulin molecules were spaced further apart than in a 13-protofilament MT, and only a minority fraction of γ-TuSC rings sampled a MT-compatible and more active closed

[1]Zentrum für Molekulare Biologie der Universität Heidelberg (ZMBH), Heidelberg, Germany. [2]European Molecular Biology Laboratory (EMBL), Heidelberg Meyerhofstraße 1, Heidelberg, Germany. [3]These authors contributed equally: Anjun Zheng, Bram J. A. Vermeulen, Martin Würtz. ✉e-mail: e.schiebel@zmbh.uni-heidelberg.de; s.pfeffer@zmbh.uni-heidelberg.de

arrangement[10,13]. In contrast, γ-TuSC rings imaged in native SPBs were observed exclusively in the closed conformation while capping MTs[15], where the conformation was presumably stabilised by lateral interactions between MT protofilaments.

Two classes of γ-TuC-activating proteins are conserved from fungi to humans[16] and required for the formation of an active MT nucleation unit. The first class of proteins, γ-TuC receptors, contain γ-TuC-binding CM1 (centrosomin motif 1) motifs and are sufficient to enhance MT nucleation of γ-TuCs in vitro[17,18] and in vivo[19]. Whereas the CM1 motif of CDK5RAP2 binds to pre-assembled γ-TuRCs in vertebrates, the CM1-containing receptor proteins Spc110 and Spc72 promote oligomerisation of yeast γ-TuSC on the nuclear and cytoplasmic side of the SPB, respectively[20]. The interaction between CM1 motifs and γ-TuCs has been structurally resolved in both vertebrates (CDK5RAP2[21–23]) and budding yeast (Spc110[10,15]), revealing a surprising heterogeneity in terms of CM1 motif oligomeric state and binding mode. However, how these binding modes can be translated to other CM1-containing proteins, such as Spc72, which has a CM1 motif distinct from Spc110[24], and how they promote γ-TuSC oligomerisation remains unclear. The second class of γ-TuC-activating proteins conserved throughout eukaryotes[25–27] are MT polymerases of the XMAP215/chTOG/Stu2 family. XMAP215/chTOG/Stu2 family members bind both γ-TuCs[28,29] and α/β-tubulin heterodimers[30] and are pivotal for MT nucleation in a range of biological contexts[29,31,32]. While S. cerevisiae Stu2 was shown to interact with Spc72[29,33], the structural basis for XMAP215/chTOG/Stu2 recruitment to MT nucleation sites remains enigmatic.

Combining high-resolution cryo-EM single particle analysis, hydrogen-$^1$H/$^2$H-exchange mass spectrometry (HX-MS), neural network-based structure predictions and structure-guided interaction analysis, we here reveal the structure and molecular architecture of the active MT nucleation unit at the cytoplasmic side of the C. albicans

SPB[29], comprising a ring-like oligomer of γ-TuSC bound to the CM1-containing receptor protein Spc72 and the MT polymerase Stu2. Our analysis provides in-depth structural and mechanistic insights into CM1-driven oligomerisation and conformational activation of the γ-TuSC into a nucleation competent MT template, evolutionary conservation of dimeric CM1 motif binding and recruitment of the MT polymerase Stu2[33–36] through its C-terminal helix conserved from fungi to vertebrates[29,31,32].

## Results

### Reconstitution of the cytoplasmic *C. albicans* nucleation unit by insect cell co-expression

Aiming to elucidate the architecture of the cytoplasmic MT (cMT) nucleation unit in C. albicans, we established insect-cell co-expression of the γ-TuSC components γ-tubulin, Spc97 and Spc98, together with a FLAG-tagged version of the MT polymerase Stu2 and a C-terminally truncated version of the γ-TuC receptor protein Spc72 (Spc72$^{1-599}$; note that tags on Spc72 are indicated in the figures), which is lacking the SPB-binding C-terminal domain (Supplementary Fig. 1a), but still interacts with the γ-TuSC and Stu2 (Supplementary Fig. 1b). Truncation of Spc72 was required to circumvent the high aggregation propensity of full-length Spc72 during co-expression. FLAG-Stu2-purified complexes obtained from insect cell expression contained all γ-TuSC components and Spc72$^{1-599}$, as judged by Coomassie blue staining (Fig. 1a). Successful higher order complex formation of FLAG-purified complexes was confirmed by size-exclusion chromatography (SEC, Supplementary Fig. 1c–f) as well as negative stain EM analysis, in which ordered ring-like oligomers of γ-TuSCs could be observed (Supplementary Fig. 1g, h).

We also explored the co-expression of Mzt1, a protein facilitating the targeting of γ-TuCs to MTOCs across different species, including

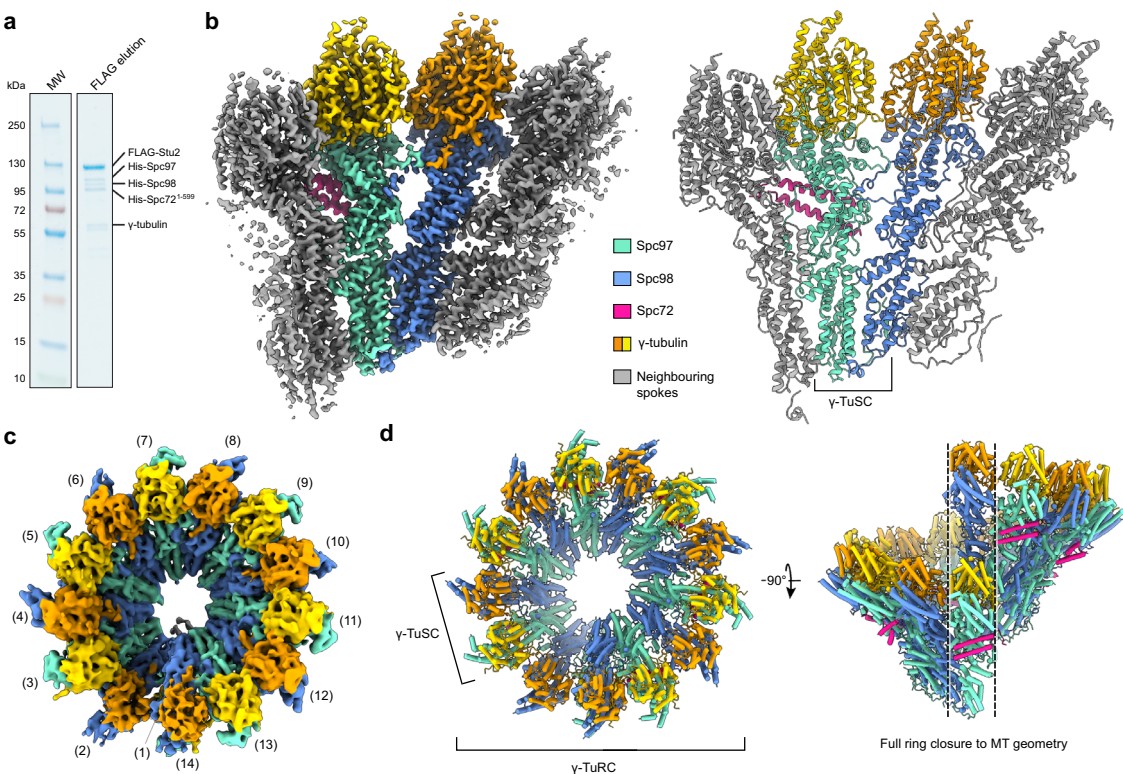

**Fig. 1 | Overall architecture of the γ-TuRC from *C. albicans*. a** Coomassie blue-stained SDS-PAGE gel of FLAG-Stu2-purified γ-TuSC/His-Spc72$^{1-599}$/FLAG-Stu2 complex. *N* > 3 biologically independent experiments. **b** Cryo-EM structure (left) and atomic model (right) of the γ-TuSC within larger oligomers, showing a γ-TuSC unit in colour with one spoke of the adjacent γ-TuSC units on either side in grey.

Colouring scheme is indicated. **c** Cryo-EM reconstruction of 14-spoked γ-TuSC rings obtained using multi-body refinement. **d** Model of the 14-spoked γ-TuSC ring created by rigid-body docking multiple copies of the γ-TuSC model from panel (**b**) into the cryo-EM reconstruction, highlighting full conformational closure to MT geometry (right). Source data are provided in the Source Data file.

the SPB of *C. albicans*[20,37–42]. His-GFP-Mzt1 effectively bound to the γ-TuSC-Spc72[1-599] complex purified through FLAG-Spc97 (Supplementary Fig. 2a). Using size-exclusion chromatography (SEC, Supplementary Fig. 2b, c) combined with negative stain EM analysis (Supplementary Fig. 2d, e), we could not find evidence that co-expression of Mzt1 impacts oligomer formation or the overall architecture of γ-TuSC rings. However, handling of Mzt1-containing complexes was more challenging and negative stain EM analysis consistently indicated that the number of γ-TuSC oligomers suitable for EM analysis was relatively low (Supplementary Fig. 2d). We therefore focused high-resolution cryo-EM analysis on γ-TuSC oligomers formed in the absence of Mzt1.

## The *C. albicans* cytoplasmic nucleation unit is conformationally primed for MT nucleation

Aiming to elucidate structural details of γ-TuSC oligomerisation, as well as Spc72 and Stu2 binding to the γ-TuSC oligomer, we next subjected the γ-TuSC/Spc72[1-599]/FLAG-Stu2 complex purified via FLAG-Stu2 to cryo-EM single particle analysis (Supplementary Fig. 3a, Supplementary Fig. 4). As the MT nucleation unit comprises higher oligomers of structurally identical γ-TuSCs, two complementary data processing strategies were pursued: in the first branch, all γ-TuSC units were treated as individual asymmetric units (Supplementary Fig. 4a–c), resolving the structure of the γ-TuSC unit within higher oligomers to 3.6 Å (Fig. 1b, Supplementary Fig. 3b-d), which was sufficient to build and refine a full atomic model of γ-TuSC components (Fig. 1b, Supplementary Table 1). In the second branch, 14-spoked oligomers of γ-TuSCs were selected through multiple rounds of focused particle sorting (Supplementary Fig. 4d) and refined to resolve the full ring of seven γ-TuSC units to 8.2 Å (Fig. 1c, Supplementary Fig. 3e-g). The atomic model for individual γ-TuSC units within higher oligomers obtained in the first processing branch could be seamlessly docked into the reconstruction of the 14-spoked γ-TuSC ring, which allowed for detailed analysis of γ-tubulin arrangement in the cytoplasmic nucleation unit (Fig. 1d). In the presence of Spc72[1-599], the γ-TuSC assembles into a ring with helical parameters that are indistinguishable from fungal 13 protofilament MTs (Table 1). Thus, in complex with Spc72[1-599], the *C. albicans* γ-TuSC stoichiometrically assumes a closed, MT-compatible conformation primed for efficient MT nucleation, unlike the budding yeast γ-TuSC in complex with the CM1-containing protein Spc110[10].

## γ-TuSC oligomerisation is paired with large conformational changes and the formation of additional interfaces

In the cryo-EM reconstruction of the non-oligomerised *C. albicans* γ-TuSC (PDB 7ANZ), the relative arrangement of γ-tubulin units strongly deviates from the configuration of α/β-tubulin units in MTs[11]. Comparing the models of non-oligomerised and oligomerised γ-TuSCs revealed extensive conformational changes within the γ-TuSC unit (Fig. 2a). While the N-terminal parts of GRIP1 domains are structurally unaffected by oligomerisation (residues 91-308: $C_\alpha$ RMSD 1.86 Å (Spc97), residues 150-306: $C_\alpha$ RMSD 1.42 Å (Spc98)), the C-terminal parts undergo a hinge-like rigid-body rotation that guides the GRIP2 domains and the associated γ-tubulins into a MT-compatible conformation, in which the γ-tubulin molecules are in much closer

proximity as compared to the isolated γ-TuSC (Fig. 2b). This conformational change is accompanied by remodelling of an ascomycete yeast-specific extended interface between the GRIP2 domains[11] (Fig. 2c) and the formation of an additional interface between the γ-tubulin molecules on both spokes, primarily electrostatic in nature (Fig. 2d). The substantial conformational change of γ-TuSC units to a MT-compatible conformation upon oligomerisation likely plays a key role in keeping the non-oligomerised γ-TuSC inactive until Spc72 recruits it to the SPB and induces γ-TuSC oligomerisation. Notably, the conformational changes in γ-TuSC units induced by *C. albicans* Spc72[1-599] (this study) and *S. cerevisiae* Spc110[1-220] [10] are overall similar (Supplementary Fig. 5a, b). However, while the closed γ-TuSC conformation is stoichiometrically achieved in *C. albicans* upon Spc72[1-599]-induced oligomerisation, Spc110[1-220]-induced oligomerisation of *S. cerevisiae* γ-TuSCs appears to stabilise an intermediate state, in which the closed state is sampled only in a minority fraction[10].

Upon oligomerisation, an interface forms between adjacent *C. albicans* γ-TuSC units. Similar to the intra-γ-TuSC interface, the GRIP1 domains and γ-tubulin molecules contribute the largest contact area between adjacent γ-TuSC units. These interactions are complemented by a small Spc97 helix, flanked by long loops, docking phenylalanine at position 721 into a strongly hydrophobic pocket in Spc98 (Fig. 2e). This helix was not visible in the cryo-EM reconstruction of the isolated γ-TuSC from *C. albicans*[11], indicating that it is stabilised in a fixed position only when interacting with the neighbouring γ-TuSC unit. This flexible character makes it a prime candidate for initiating the interaction between γ-TuSC units, before the more extensive GRIP1 and γ-tubulin interfaces form. Lastly, the extension of a coiled-coil helix, later identified as the CM1 motif of Spc72, contributes a small contact with the GRIP2 domain of Spc98 in the neighbouring γ-TuSC unit (see below).

In conclusion, oligomerisation of γ-TuSC units in the presence of Spc72[1-599] is coupled to large scale conformational changes that stoichiometrically transition the γ-TuSC into a closed MT compatible conformation. This closed conformation is stabilised by the formation of additional inter- and intra-γ-TuSC interfaces.

## The Spc72 CM1 motif binds as a dimeric coiled-coil and is essential for γ-TuSC oligomerisation and function

Our high-resolution cryo-EM reconstruction of the γ-TuSC unit within higher oligomers displayed density to the amino acid side chain level for two α-helices forming a coiled-coil on the outer surface of the Spc97 GRIP2 domain (Fig. 3a, Supplementary Fig. 6a). We used the input sequence-free model building functionality of ModelAngelo to identify the α-helices in a completely unbiased manner. For both α-helices, ModelAngelo predicted a sequence highly similar to the CM1 motif of Spc72 (Supplementary Fig. 6a), revealing that the CM1 motif of Spc72 binds as a parallel dimeric coiled coil, comparable to human CDK5RAP2[21], but, surprisingly, different from monomeric CM1 binding in budding yeast Spc110[10]. The inner helix of the dimeric CM1 motif (CM1$_{in}$) is coordinated by four distinct interfaces towards Spc97 (Fig. 3b, Supplementary Fig. 6b, c), including hydrophobic contacts with two short α-helices in an insertion of the Spc97 GRIP1 domain that is unresolved in the isolated γ-TuSC. The outer helix (CM1$_{out}$) additionally contributes to one of those interfaces (Supplementary Fig. 6b, c). These interactions are consistent with previous

**Table 1 | Geometric parameters of γ-tubulin within the *C. albicans* γ-TuRC in complex with Spc72[1-599]**

|  | *C. albicans* γ-TuRC with Spc72[1-599] | *S. cerevisiae* 13 protofilament MT |
|---|---|---|
| Distance to helical axis (Å) | 102.5 ± 1.5 | 101.8 ± 0.1 |
| Rotation around axis per spoke (degrees) | 27.6 ± 0.7 | 27.6 ± 0.0 |
| Helical pitch increment per spoke (Å) | 9.2 ± 0.8 | 9.6 ± 0.1 |

Coordinates were taken at Gln12 of γ-tubulin subunits and compared to coordinates taken at Gln15 of α-tubulin subunits in a 13 protofilament MT (*S. cerevisiae*, PDB 5W3F fit into EMD-8758[79]). n = 13 individual tubulins, shown as mean ± standard deviation.

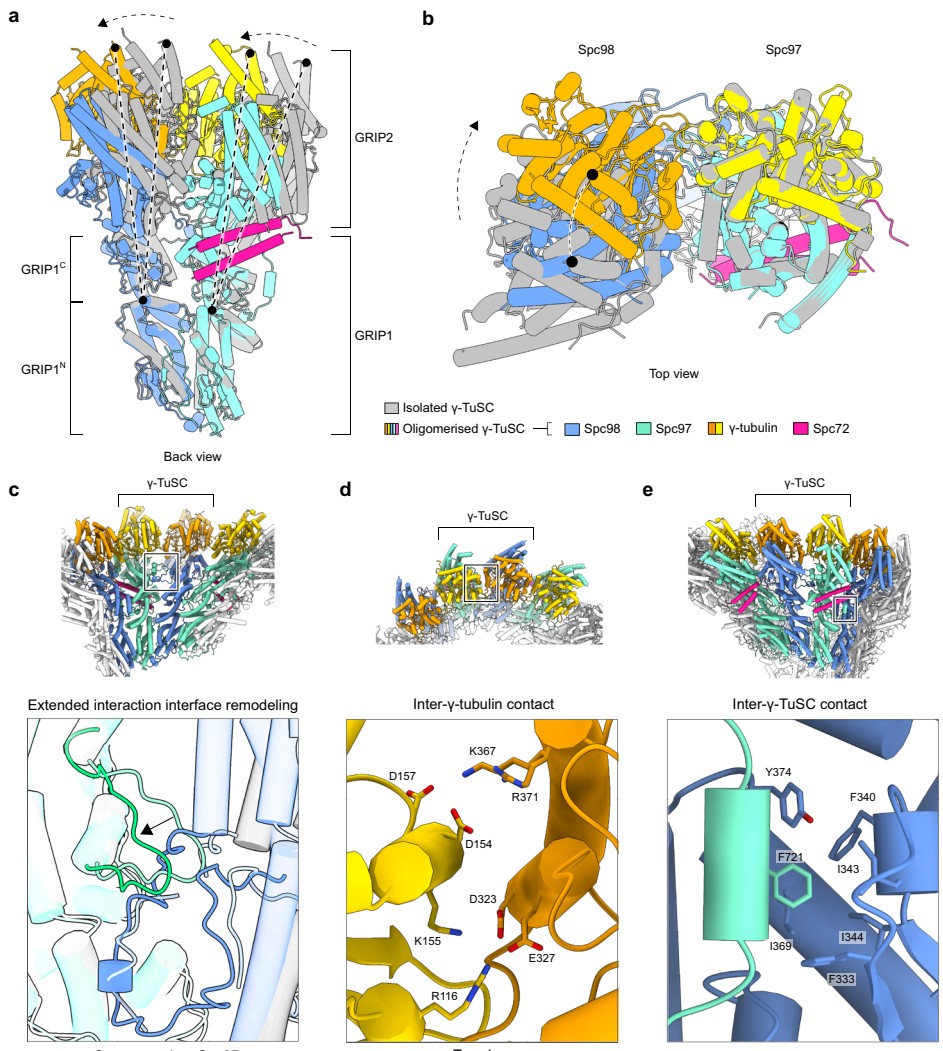

**Fig. 2 | Conformational changes and interactions involved in γ-TuSC oligo-merisation. a** The C-terminal parts of Spc97/98 GRIP1 domains (GRIP1ᶜ), full GRIP2 domains and γ-tubulins undergo a hinge motion relative to the N-terminal part of the GRIP1 (GRIP1ᴺ) domain upon γ-TuSC oligomerisation. Colouring scheme is indicated at the bottom of panel (**b**). **b** During the conformational change shown in panel (**a**), the γ-tubulin molecules move closer together (top view). Models were superposed on the GRIP1ᶜ/GRIP2 domain of Spc97 with γ-tubulin. **c** Remodeling of the extended interface between GRIP2 domains upon oligomerisation. Colouring as indicated in panel (**b**). **d** Electrostatic interactions at the interface between two γ-tubulin molecules in the oligomeric γ-TuSC. **e** Docking of Spc97 Phe721 in a strongly hydrophobic pocket of Spc98 in the neighbouring γ-TuSC.

mutagenesis studies on Spc72 binding to the γ-TuSC (Supplementary Fig. 6d)[20]. Moreover, the CM1 motif of Spc72 contributes hydrogen bond contacts towards the GRIP2 domain of Spc98 in the neighbouring γ-TuSC unit (Fig. 3c; T234-V500 (backbone to backbone), T236-D497, S238-S471) and may thus contribute to γ-TuSC oligomerisation. A similar γ-TuSC-bridging function was also observed for the monomeric CM1 motif in *S. cerevisiae* Spc110, but compared to *C. albicans* Spc72, the binding site of the *S. cerevisiae* Spc110 CM1 motif is substantially shifted towards Spc98 of the neighbouring γ-TuSC unit (Supplementary Fig. 6e).

Thus, to test whether the interface between the Spc72 CM1 motif and the neighbouring γ-TuSC unit contributes to γ-TuSC oligomerisation in a similar way as the *S. cerevisiae* Spc110 CM1 motif despite their divergent binding modes, we generated two point mutations in Spc72 (T234P, T236A, referred to as the PA mutant) on the very N-terminal end of the CM1 motif, aiming to specifically disrupt the backbone and side chain hydrogen bonding network at the interface with Spc98 of the neighbouring spoke (Fig. 3c). To quantify γ-TuSC oligomerisation, we subjected complexes containing wild-type Spc72[1-599] or mutant Spc72[1-599,PA] to SEC and analysed protein

distribution using SDS-PAGE and Western blotting. For the wild-type γ-TuSC/Spc72[1-599]/FLAG-Stu2 complex, proteins are mostly present in fractions corresponding to oligomeric γ-TuSC, while the majority of proteins shifts to fractions corresponding to non-oligomerised γ-TuSC for the mutant γ-TuSC/Spc72[1-599,PA]/FLAG-Stu2 complex (Fig. 3d, e). This validates a general role of Spc72 for γ-TuSC oligomerisation and indicates that the interaction between the Spc72 CM1 motif and Spc98 of neighbouring γ-TuSC units is central to γ-TuSC oligomerisation.

To assess the impact of the PA mutations on *SPC72* function in vivo, we engineered a mutant version of *SPC72* in the yeast *S. cerevisiae*. Given that amino acids T234 and T236 in *C. albicans* Spc72 are not conserved in *S. cerevisiae* Spc72, we identified the corresponding region (P55-N62) in *S. cerevisiae* based on sequence alignment (Supplementary Fig. 7a) and then deleted codons P55-N62 of *SPC72* (*spc72[ΔP55-N62]*). The strain *spc72[ΔP55-N62]* exhibited cell viability in two distinct *S. cerevisiae* backgrounds (W303, S288C) from 16 °C to 30 °C (Supplementary Fig. 7b), with little to no effect on growth rate. Because it is known that mutations in yeast can lead to defective cMT function and organisation without noticeably affecting cell doubling time[43], we next analysed the phenotype of *spc72[ΔP55-N62]* W303 cells

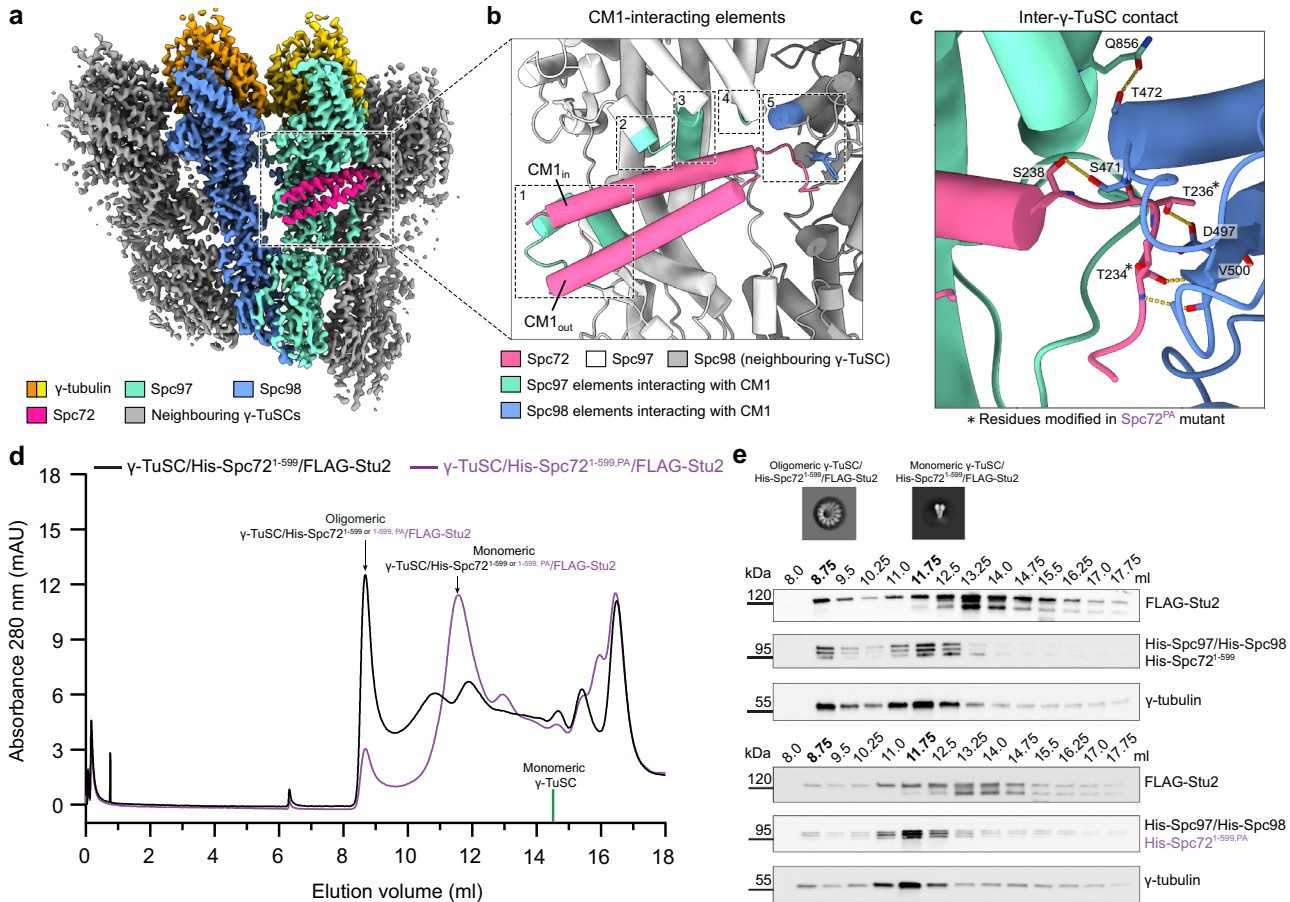

**Fig. 3 | CM1 motif-mediated interactions are central to γ-TuSC oligomerisation.**
**a** Back view of the high-resolution reconstruction of the γ-TuSC within higher oligomers. **b** Overview of the five interaction elements between the Spc72 CM1 motif and the γ-TuSC. Inner (CM1$_{in}$) and outer (CM1$_{out}$) helices of the CM1 coiled-coil are indicated. **c** Detailed depiction of the interaction between the Spc72 CM1 motif and Spc98 on the neighboring γ-TuSC. Mutated residues (T234P, T236A) in the PA mutant are indicated by asterisks. **d** Size-exclusion chromatography (SEC) profile indicating that Spc72-induced oligomerisation of the γ-TuSC is reduced in His-Spc72$^{1-599,PA}$ compared to wild-type His-Spc72$^{1-599}$. Fractions corresponding to monomeric and oligomeric γ-TuSC/His-Spc72$^{1-599}$/FLAG-Stu2 and γ-TuSC/His-Spc72$^{1-599,PA}$/FLAG-Stu2 are indicated. Elution volume of free monomeric γ-TuSC is

indicated at 14.5 ml (green line, see Supplementary Fig. 1e, f). $N = 2$ biologically independent experiments. **e** Visualisation of the protein content in fractions from SEC profiles for γ-TuSC/His-Spc72$^{1-599}$/FLAG-Stu2 (top) and for γ-TuSC/His-Spc72$^{1-599,PA}$/FLAG-Stu2 (bottom) as shown in panel (**d**) by Western blotting. Representative negative stain 2D classes for monomeric and oligomeric γ-TuSC (obtained from wild-type γ-TuSC/His-Spc72$^{1-599}$/FLAG-Stu2 complex) are shown. Western blotting was performed with antibodies against FLAG-Stu2, His-Spc97/His-Spc98/His-Spc72$^{1-599}$/His-Spc72$^{1-599,PA}$ and γ-tubulin (home-made anti-*C. albicans* anti-γ-tubulin antibodies). $N = 2$ biologically independent experiments. Source data are provided in the Source Data file.

compared to *SPC72* control cells with a focus on cMTs, which are expected to be affected by mutations in Spc72. For the analysis, we incubated cells at 16 °C, because defects in the MT cytoskeleton are frequently more apparent at lower temperatures[43]. Compared to wild-type *SPC72* control cells, *spc72$^{ΔPS5-N62}$* cells either lacked detectable cMTs (Supplementary Fig. 7c, yellow arrows) or showed elongated cMTs (Supplementary Fig. 7c; yellow asterisks) when tracing GFP-Tub1 (α-tubulin) and Spc42-mCherry (SPB component) signals. The aberrant cMTs affected the positioning of the nucleus (Supplementary Fig. 7d, e) and increased the frequency of cells with misaligned or mispositioned spindle in the mother cell body ~2.5 fold (Supplementary Fig. 7c, f). These observations suggest that *spc72$^{ΔPS5-N62}$* cells cannot assemble a fully functional γ-TuRC, which leads to cMT defects and aligns with the functional significance of the inter-γ-TuSC contacts facilitated by Spc72 (Fig. 3d).

### CM1 motif dimerisation and γ-TuRC interactions are evolutionarily conserved
The observation that the *C. albicans* Spc72 CM1 motif associates with Spc97 as a dimer was surprising, given that the CM1 of the

evolutionarily closely related *S. cerevisiae* Spc110 receptor binds in monomeric form[10]. This prompted us to investigate the mode of CM1 motif binding across evolution by inspecting the conservation of coiled-coil interface residues and the predicted propensity for coiled-coil formation in CM1 motifs using DeepCoil2[44] predictions over a wide range of ascomycetes and other representative eukaryote species, including vertebrates (Fig. 4a, Supplementary Fig. 8a). Among all sequences analysed, monomeric binding of the CM1 motif was a unique feature of *S. cerevisiae* Spc110. In all other CM1 motifs analysed, including human CDK5RAP2, *S. cerevisiae* Spc72 and *C. albicans* Spc110, CM1 dimerisation appears to be conserved.

Having identified dimeric CM1 motif binding to γ-tubulin complexes as an evolutionarily conserved feature, we next investigated whether the general binding mechanism of the CM1 dimer is conserved as well, by comparing the dimeric CM1 motif structure as associated with the *C. albicans* cMT nucleation unit and the human γ-TuRC[5–7]. Extension of atomic models for the human CDK5RAP2 CM1 motif bound to spoke 13 in the γ-TuRC[6,21] (Supplementary Fig. 8b, c) revealed that indeed all interactions observed for *C. albicans* Spc72 also have their equivalents in the human system (Fig. 4b). However, in contrast

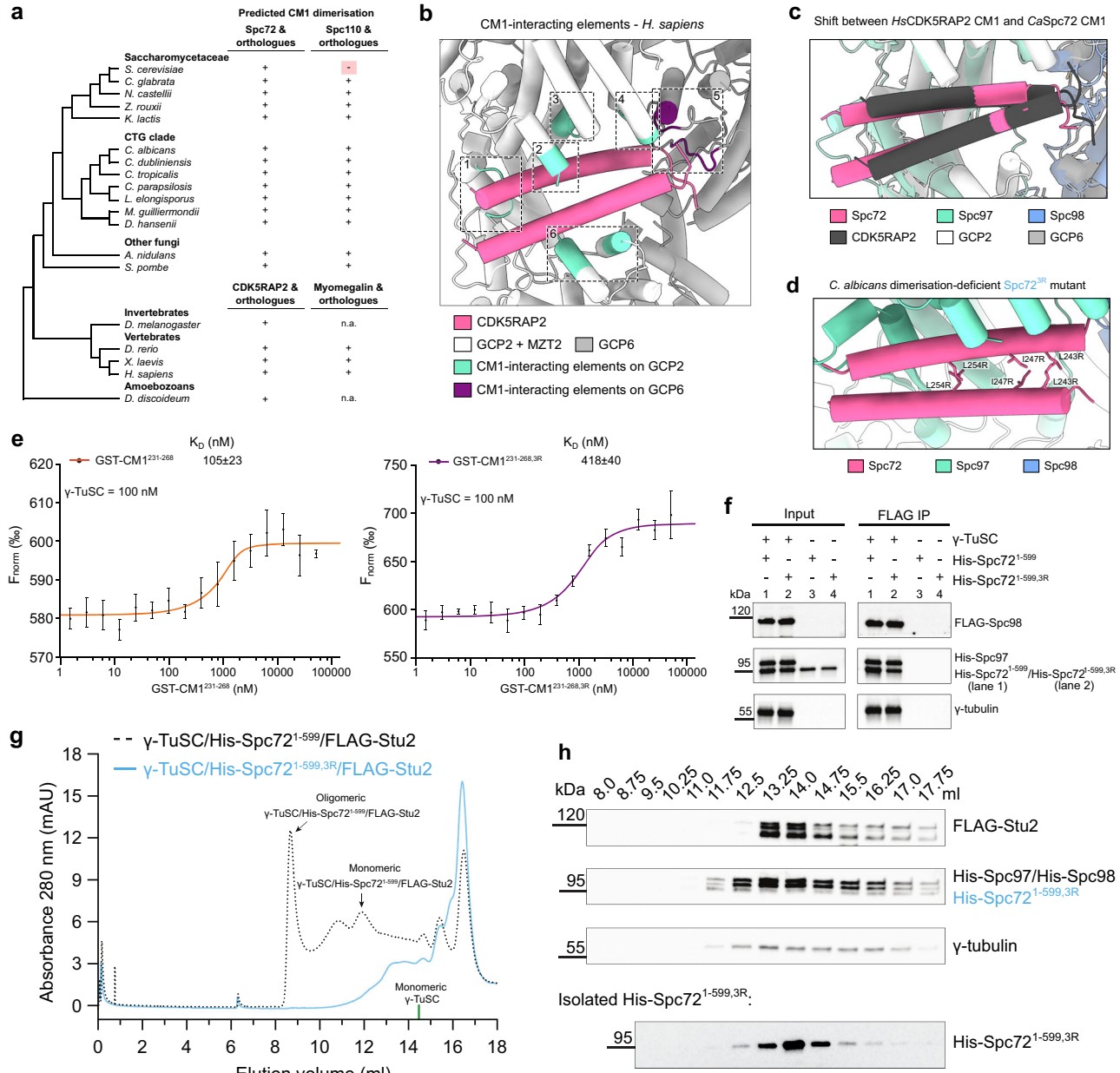

**Fig. 4 | CM1 dimerisation is evolutionary conserved and plays an important role in γ-TuSC binding and oligomerisation. a** Predicted dimerisation of CM1 motifs in fungal Spc72 and Spc110 orthologues, as well as CDK5RAP2 and myomegalin orthologues in other eukaryotes. Dimerisation was predicted using DeepCoil2[44] and based on the conservation of residues at the coiled-coil interface (Supplementary Fig. 8a). **b** Visualisation of elements on human GCP2 (Spc97 homologue) and GCP6 interacting with the CDK5RAP2 CM1 motif. **c** Comparison between the positioning of CM1 motifs in *C. albicans* Spc72 and human CDK5RAP2 on Spc97 and GCP2, respectively. Structures were aligned on the CM1-interacting elements of Spc97 and GCP2, respectively. Colouring in panel (**b**) and (**c**) as indicated. **d** Visualisation of residues L243, I247, and L254 in the dimerisation interface that are mutated to disrupt the dimerisation of the Spc72 CM1 motif in the 3R mutant. **e** Binding affinities of wild-type and 3R-mutated GST-CM1[231-268] with the γ-TuSC, measured with microscale thermophoresis (MST). Data are shown as mean ± SD. *N* = 2 biologically independent experiments. **f** Pull-down of His-Spc72[1-599] and His-Spc72[1-599,3R] by the γ-TuSC through FLAG-tagged Spc98. Western blotting was performed with antibodies against FLAG-Spc98, His-Spc97/His-Spc72[1-599]/His-Spc72[1-599,3R] and γ-tubulin. *N* = 3

biologically independent experiments. **g** SEC chromatograms of γ-TuSC/His-Spc72[1-599]/FLAG-Stu2 (also shown in Fig. 3d) and γ-TuSC/His-Spc72[1-599,3R]/FLAG-Stu2. Fractions corresponding to monomeric and oligomeric γ-TuSC/His-Spc72[1-599]/Stu2 are indicated. Elution volume of monomeric γ-TuSC is indicated at 14.5 ml (indicated by a green line, Supplementary Fig. 1e, f). **h** Top: Visualisation of the protein content in fractions from the SEC profile of γ-TuSC/His-Spc72[1-599,3R]/FLAG-Stu2 shown in panel (**g**) by Western blotting. γ-TuSC is primarily present as uncomplexed monomer, with a minor shoulder of γ-TuSC/His-Spc72[1-599,3R] complex. Bottom: Visualisation of the protein content in fractions from the SEC profile of isolated His-Spc72[1-599,3R] by Western blotting. Note that the elution volumes of monomeric γ-TuSC and isolated Spc72[1-599,3R] coincide at 12–14 ml. Western blotting was performed with antibodies against FLAG-Stu2, His-Spc97/His-Spc98/His-Spc72[1-599,3R] and γ-tubulin. Western blot for γ-TuSC/His-Spc72[1-599]/FLAG-Stu2 is shown in Fig. 3e. Experiments in panels (**g**) and (**h**) were performed in N = 2 biologically independent experiments. Source data are provided in the Source Data file.

to the general conservation of structural elements involved in dimeric CM1 motif binding, the interactions are divergent at the residue level and the CM1 motif-containing α-helices are shifted by 8-14 Å (inner and outer helix, respectively) between the two systems to accommodate structural differences in the relative positioning of CM1-interacting spokes in *C. albicans* and the human γ-TuRC (Fig. 4c).

Cumulatively, while dimerisation of CM1 motifs and general structural elements in CM1 motif binding are conserved, interactions are divergent on the residue level to adapt to variations of γ-tubulin complex architecture.

## CM1 dimerisation may play an important role in γ-TuSC binding and in vivo function of Spc72

To test whether dimerisation of the Spc72 CM1 motif is important for γ-TuSC oligomerisation and Spc72 function, we introduced three amino acid substitutions (L243 R/I247R/L254R, 3R mutant) in the *C. albicans* Spc72 CM1 motif at the coiled-coil interface to disrupt $CM1_{in}$-$CM1_{out}$ dimerisation, while completely preserving the binding interface between $CM1_{in}$ and the γ-TuSC (Fig. 4d; Spc72$^{1-599}$ 3R mutant; referred to as Spc72$^{1-599,3R}$). To verify that the CM1 motif of Spc72 is capable of dimerisation and that dimerisation is disrupted by the 3R mutations, we expressed and purified wild-type MBP-Spc72$^{231-268}$ and mutated MBP-Spc72$^{231-268,3R}$ (Supplementary Fig. 9a, hereafter referred to as MBP-CM1$^{231-268}$ and MBP-CM1$^{231-268,3R}$). As predicted, SEC analysis (at 100 μM) revealed that MBP-CM1$^{231-268,3R}$ was exclusively in monomeric form, while MBP-CM1$^{231-268}$ predominantly existed as a dimer (Supplementary Fig. 9b). However, at lower concentrations as used in mass photometry ( < 40 nM), MBP-CM1$^{231-268}$ formed monomers, similar to MBP-CM1$^{231-268,3R}$ (Supplementary Fig. 9c, d). In Spc72$^{1-599}$, the CM1 motif is directly followed by a coiled-coil region that most likely promotes CM1 dimerisation by keeping CM1 helices in spatial proximity. To mimic this effect and promote CM1$^{231-268}$ dimerisation also under low concentrations, we expressed the same CM1 variants with an N-terminal GST tag (GST-CM1$^{231-268}$ and GST-CM1$^{231-268,3R}$) (Supplementary Fig. 9e). As expected, the GST tag promoted dimerisation in both constructs and CM1-mediated dimerisation of dimers induced tetramer formation for a substantial portion of proteins in case of GST-CM1$^{231-268}$, as observed by mass photometry (Supplementary Fig. 9f, g). A low level of tetramer formation was also observed in case of GST-CM1$^{231-268,3R}$, which may suggest that CM1 motif dimerisation was strongly reduced but not completely abolished by the 3R mutation under these experimental conditions.

To characterise the interaction between γ-TuSCs and wild-type or mutated CM1, we conducted MicroScale Thermophoresis (MST) measurements. We observed that the $K_D$ value for the binding of GST-CM1$^{231-268,3R}$ to the γ-TuSC is ~4 times higher than that for wild-type GST-CM1$^{231-268}$ (Fig. 4e), which suggests that CM1$_{out}$ significantly contributes to the Spc72-γ-TuSC interaction (Supplementary Fig. 6c). Notably, a low level of CM1$^{231-268,3R}$ dimerisation may contribute to the remaining γ-TuSC binding observed for GST-CM1$^{231-268,3R}$. Although we were unable to determine the affinity of Spc72$^{1-599}$ and Spc72$^{1-599,3R}$ for the γ-TuSC using MST due to protein aggregation at the concentrations needed to saturate γ-TuSC binding, co-IP confirmed a $1.7 \pm 0.2$ fold ($n = 3$, mean ± SD) lower binding efficiency of Spc72$^{1-599,3R}$ to γ-TuSC compared to Spc72$^{1-599}$ (Fig. 4f). Consistently, oligomerisation of the γ-TuSC was drastically reduced in the presence of Spc72$^{1-599,3R}$ (Fig. 4g, h). While we cannot exclude entirely that alternative defects induced by the 3R mutant may contribute to the observed effects, this likely suggests that dimerisation of the Spc72 CM1 motif plays a central role for establishing high affinity binding to γ-TuSCs, which is essential for the formation of γ-TuSC oligomers in *C. albicans*.

Based on the *C. albicans* Spc72 CM1 dimerisation interface, we identified the corresponding residues in *S. cerevisiae* Spc72 (Fig. 5a) and introduced three amino acid substitutions (L67R/L71R/I78R, 3R

mutations) to disrupt Spc72 CM1 dimerisation (*spc72$^{3R}$*). We first tested whether spc72$^{3R}$ is dominant lethal, as would be expected if it is non-functional for inducing γ-TuSC oligomerisation but competes with wild-type Spc72 for binding to the SPB. Indeed, galactose-induced expression of *GAL1-spc72$^{3R}$* in *S. cerevisiae* impaired growth, whereas no such effect was observed with *GAL1-SPC72* when compared to the empty vector control (Fig. 5b). Additionally, *spc72$^{3R}$* failed to complement for *SPC72* in a plasmid shuffle experiment in both S288C and W303 cells, suggesting that Spc72 CM1 dimerisation is important for its function (Fig. 5c).

The severe growth impairment of *spc72$^{3R}$* cells, even in the W303 strain background, where *SPC72* is not essential for viability but still strongly impacts cell growth, is likely to cause secondary effects due to cell death (Fig. 5c). To analyse the phenotype of *spc72$^{3R}$* cells, we therefore employed IAA (auxin)-induced degradation[45] of wild-type Spc72-IAA7 (auxin-inducible degron) in cells that also expressed either *SPC72* or *spc72$^{3R}$* from the *SPC72* promoter. Three hours after the addition of degron inducer IAA, Spc72-IAA7 was effectively degraded (Fig. 5d), allowing for the analysis of the phenotype associated with the *spc72$^{3R}$* allele without secondary effects caused by dying cells. *SPC72-IAA7 spc72$^{3R}$* cells exhibited defects in cMT organisation, misaligned and broken spindles, as well as multiple spindles within the mother cell body (Fig. 5e, f). This phenotype is consistent with defective cMTs causing mitotic cell cycle arrest due to activation of the spindle position checkpoint[46,47]. The defects in *SPC72-IAA7 spc72$^{3R}$* cells were comparable to *SPC72-IAA7* cells harbouring the empty pRS315 plasmid, but strongly reduced in *SPC72-IAA7 SPC72* cells, indicating that *spc72$^{3R}$* is strongly defective in mediating cMT nucleation by the γ-TuSC.

In summary, our data suggest that dimerisation of the Spc72 CM1 motif plays an important role in binding to γ-TuSCs and, as a consequence, for γ-TuSC oligomerisation, activation and the formation of functional cMTs in yeast.

## Stu2 interacts with Spc72 coiled-coil segments

We next aimed to understand how Spc72 recruits the MT polymerase Stu2 to the γ-TuSC. Despite our purification strategy using FLAG-Stu2 as bait, no clear density could be observed for Stu2 in the cryo-EM reconstruction of the cytoplasmic nucleation unit. This indicates that interactions with flexible segments of the γ-TuSC or Spc72 may underlie Stu2 recruitment, although it cannot be entirely excluded that Stu2 has dissociated from the γ-TuSC/Spc72$^{1-599}$ complex during cryo-EM sample preparation. Thus, to obtain insights into the structural basis for Stu2 binding, we first dissected the interaction between Stu2 and Spc72$^{1-599}$ using pull-down experiments in combination with AlphaFold2-based structure predictions.

Consistent with data from *S. cerevisiae*[29], a C-terminally truncated version of *C. albicans* Stu2 (FLAG-Stu2$^{Δ894-924}$, Supplementary Fig. 10a) was unable to bind to Spc72$^{1-599}$ (Supplementary Fig. 10b, lane 5), while wild-type FLAG-Stu2 could pull down Spc72$^{1-599}$ (Fig. 6a, lane 1). To exclude a contribution of the Stu2 coiled-coil region (666-768) to the Spc72$^{1-599}$-Stu2 interaction, we compared Spc72$^{1-599}$ pull-down by FLAG-Stu2$^{Δ666-768}$ and FLAG-Stu2, observing no difference in pull-down efficiency (Fig. 6a, lane 1 and Supplementary Fig. 10b, lane 1). Thus, Stu2 interacts with Spc72$^{1-599}$ via its C-terminal α-helix.

To identify binding sites of Stu2 on Spc72$^{1-599}$, we next generated Spc72 fragments based on Spc72 functional domains and AlphaFold2 predictions. In addition to a long unstructured N-terminal region (1-230) and the CM1 motif (231-268), Spc72$^{1-599}$ was predicted to consist of two flexibly linked modules of dimerising coiled-coil regions (293-418 and 434-560) (Supplementary Fig. 11a and Supplementary Fig. 12a, b). While Spc72$^{1-239}$ and Spc72$^{1-290}$ could not be pulled down by FLAG-Stu2, co-IP efficiency of Spc72$^{291-599}$ was comparable to Spc72$^{1-599}$ (Fig. 6a, lanes 1-4), indicating that the Spc72 coiled-coil segments are central to the Stu2-Spc72$^{1-599}$ interaction.

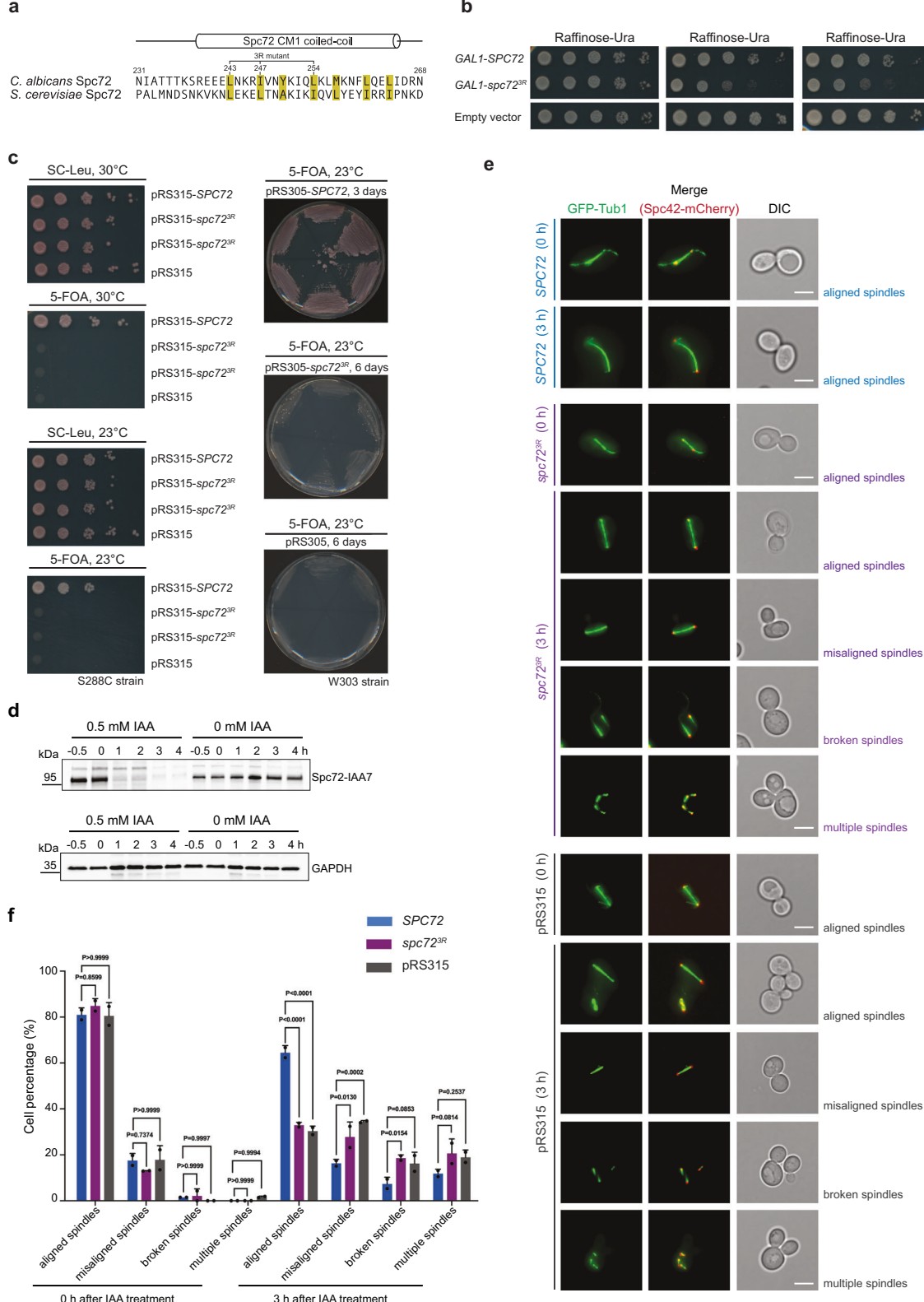

## Two binding sites on Spc72 flexibly tether Stu2 to the γ-TuSC/Spc72 complex

To narrow down the region in Spc72$^{1-599}$ that interacts with Stu2, we used AlphaFold2 to predict structures of Spc72$^{1-599}$ in complex with the C-terminal Stu2 helix. AlphaFold2 predicted two potential interaction sites, involving the C-terminal helix of Stu2 and either Spc72$^{305-345}$ or Spc72$^{435-475}$, located on the first and second Spc72 coiled coil module respectively (Fig. 6b, c, Supplementary Fig. 11b, c, Supplementary Fig. 12c-f). To test these predictions experimentally, we expressed structure-guided Spc72 fragments in *E. coli* and analysed for interaction with insect cell-expressed Stu2 and Stu2$^{Δ894-924}$ (Fig. 6b, Supplementary Fig. 10c). The Spc72 fragments tested were spanning the first or second coiled-coil module (291-428 and 429-599, respectively), as well as more defined segments corresponding to the two predicted

**Fig. 5 | CM1 dimerisation plays a central role for Spc72 function in vivo. a** Sequence alignment of the Spc72 CM1 motif in *C. albicans* and *S. cerevisiae*, highlighting hydrophobic residues at the dimerisation interface in yellow. Sequence numbering and secondary structure are indicated based on *C. albicans* Spc72. Residues mutated to arginine in the 3R mutant are indicated, showing their conservation in *S. cerevisiae* that allows preparation of the *spc72³R* mutant in *S. cerevisiae*. **b** pRS316-*GAL1-SPC72*, pRS316-*GAL1-spc72³R*, and pRS316-*GAL1* were transformed into a modified ESM356-1 *S. cerevisiae* strain (YAJZ023, see Table S3), plated on SC-Ura plates cultured at 30 °C for 2 days. Cells were 10-fold serially diluted to test cell growth. The transformants were plated on raffinose plates lacking uracil (Ura) and containing increasing galactose concentrations (0%, 1%, and 2%). Plates were incubated at 30 °C for 2 days. *N* = 2 biologically independent experiments. **c** Viability assay of *S. cerevisiae* cells expressing wild-type *SPC72* or the *spc72³R* mutant. Cells were grown for 2 days at 30 °C or 3 to 6 days at 23° C. Viability tests were performed three times. **d** *SPC72-IAA7* degron cells were treated with 0.5 mM IAA or without for 0, 1, 2, 3, and 4 h. Cells harvested 0.5 h before the IAA treatment were used as a control. Spc72 expression level was analysed by Western blotting using an anti-Spc72 antibody. GAPDH was used as a loading control. *N* = 2 biologically independent experiments. **e** *SPC72-IAA7* cells containing pRS315-*SPC72*, pRS315-*spc72³R* or pRS315 plasmids were analysed after 0 or 3 h of 0.5 mM IAA treatment by microscopy. GFP-Tub1 (marking MTs) and Spc42-mCherry (marking the SPB) were used for visualisation. Scale bars: 5 μm. DIC: differential interference contrast. *N* = 3 biologically independent experiments. Phenotypic analysis was performed by three independent replicates. **f** Different MT phenotypes from panel (e) were quantified. *N* = 2 biologically independent experiments. Data shown as mean ± SD. n > 50 cells per experiment. Data are shown as the mean ± SD. Statistical significance was determined using a 2-way ANOVA test (Šídák multiple comparison test). *P*-values are indicated on the graphs. Source data are provided in the Source Data file.

interaction interfaces (300-350 and 430-480, respectively). While we observed clear interaction of FLAG-Stu2 with Spc72⁴²⁹⁻⁵⁹⁹ and Spc72⁴³⁰⁻⁴⁸⁰ in the IP experiment (Fig. 6b, lanes 4 and 5), no interaction could be observed with FLAG-Stu2^Δ894-924 (Supplementary Fig. 10c, lanes 4 and 5). Additionally, we observed weaker binding of Stu2 to the second interaction site predicted by AlphaFold2 (Spc72²⁹¹⁻⁴²⁸) (Fig. 6b, lane 2 and 3 and Fig. 6c). Cumulatively, these experiments established that Stu2⁸²⁴⁻⁹²⁴ interacts with two coiled-coil regions in Spc72 (Spc72⁴³⁰⁻⁴⁸⁰ and Spc72³⁰⁰⁻³⁵⁰).

To further validate the two predicted binding sites of Stu2 on Spc72, we used hydrogen exchange mass spectrometry (HX-MS), which reports on stabilisation of protein secondary structure elements originating from protein-protein interactions[48]. We achieved a near-complete coverage of coiled-coil modules 1 and 2 (CC1 and CC2) in Spc72²⁹¹⁻⁵⁹⁹ (Supplementary Fig. 13a, b), which contain the two binding sites predicted by AlphaFold2 and confirmed using co-IP experiments (Fig. 6b, c). Upon addition of FLAG-Stu2, we observed strong and specific protection of Spc72 peptides located in the primary and secondary Stu2 binding sites (Fig. 6d, Supplementary Fig. 13c–e). Notably, strong protection of the secondary Stu2 binding site was achieved only under higher Stu2 concentrations (2:1 molar ratio), which suggests lower affinity compared to the primary binding site, in line with our co-IP experiments (Supplementary Fig. 13e, Fig. 6b). For FLAG-Stu2, HX-MS reached good coverage of the TOG domains (Supplementary Fig. 14a-e, Supplementary Fig. 12g, h), but large parts of the C-terminal Stu2 regions, including the Spc72-binding Stu2 α-helix, could not be detected by mass spectrometry of peptic peptides, preventing us from more detailed analysis of the Spc72-Stu2 interaction from the side of Stu2.

Finally, to support the predicted interactions, we generated point mutations at the predicted interfaces between Spc72³⁰⁰⁻³⁵⁰, Spc72⁴³⁰⁻⁴⁸⁰ and Stu2⁸⁸²⁻⁹²⁴ aiming to disrupt the interaction between the two proteins. Both interfaces were predicted to involve conserved hydrophobic residues in Stu2, flanked by conserved positively charged residues interacting with negatively charged amino acids on the Spc72 coiled-coil modules (Supplementary Fig. 11b–h). Fully consistent with the first interaction predicted by AlphaFold2, binding of FLAG-Stu2⁸⁸²⁻⁹²⁴ to Spc72⁴³⁰⁻⁴⁸⁰ was strongly affected by mutations in Spc72⁴³⁰⁻⁴⁸⁰ (EDID: E455A, D456A, I458A, D462A, Supplementary Fig. 10d, lanes 3 and 4) and, vice-versa, interaction of a mutated FLAG-Stu2⁸⁸²⁻⁹²⁴ variant (LIM: L906A, I910A, M913A) with wild-type Spc72⁴³⁰⁻⁴⁸⁰ was almost fully abrogated (Supplementary Fig. 10d, lanes 4 and 8). Similarly, the interaction of Spc72³⁰⁰⁻³⁵⁰ with FLAG-Stu2⁸⁸²⁻⁹²⁴ was reduced by mutations in Spc72³⁰⁰⁻³⁵⁰ (ELLY: E317R, L319A, L321R, Y326A) (Supplementary Fig. 10d, lanes 1 and 2) and the mutated FLAG-Stu2⁸⁸²⁻⁹²⁴ LIM variant displayed strongly weakened binding to Spc72³⁰⁰⁻³⁵⁰ (Supplementary Fig. 10d, lanes 2 and 6). This indicates that the predicted interface residues on Spc72 and Stu2 are directly involved in the Spc72-Stu2 interaction. However, the weak interaction remaining for the Spc72³⁰⁰⁻³⁵⁰ ELLY mutant may suggest that the secondary binding site involves residues additional to those mutated.

Finally, having established that two Stu2 binding sites are present on Spc72, we aimed to address the molar stoichiometry of Stu2 and Spc72 in the complex. We analysed the oligomerisation status of Spc72²⁹¹⁻⁵⁹⁹ and FLAG-Stu2, as well as the Spc72²⁹¹⁻⁵⁹⁹/FLAG-Stu2 complex using mass photometry and SEC-MALS. By themselves, Stu2 and Spc72²⁹¹⁻⁵⁹⁹ formed stable homo-dimers in isolation across the range of concentrations used in these experiments (Supplementary Fig. 15a–c). When incubated, Spc72²⁹¹⁻⁵⁹⁹ and FLAG-Stu2 formed heterogeneous complexes at high molecular weight, consistent with the formation of large Spc72²⁹¹⁻⁵⁹⁹-Stu2 networks due to multivalent interactions (Supplementary Fig. 15d, e). However, when quantifying the band intensities for Stu2 and Spc72¹⁻⁵⁹⁹ in Coomassie-stained gels of γ-TuSC/Spc72¹⁻⁵⁹⁹/FLAG-Stu2 complex purified by SEC after co-expression (Supplementary Fig. 1d), the relative intensities suggested that each Spc72 dimer binds slightly below two Stu2 dimers, consistent with the presence of multiple Stu2 binding sites on Spc72 (Fig. 6e). As expected from cryo-EM analysis, band intensities for Spc97, Spc98, γ-tubulin and Spc72¹⁻⁵⁹⁹ were approximately in a 1:1:2:2 ratio, validating this approach (Fig. 6e).

Cumulatively, by combining systematic co-IP experiments, mutational analysis, HX-MS and AlphaFold2-based structure predictions, we revealed that Stu2 is tethered to the cytoplasmic nucleation unit by flexibly linked coiled-coil modules, which are anchored to the γ-TuSC ring by the dimerised Spc72 CM1 motif (Fig. 6f).

## Discussion

Conformational changes of γ-TuSC units in ring-like oligomers directly control the arrangement of the associated γ-tubulin molecules and therefore are considered a central mechanism in the regulation of MT nucleation activity. Our cryo-EM reconstruction of the *C. albicans* γ-TuSC oligomer in complex with Spc72¹⁻⁵⁹⁹ reveals a complex that stoichiometrically assumes a geometry fully compatible with the MT lattice. In contrast, the closely related *S. cerevisiae* γ-TuSC oligomer in complex with an N-terminal segment of Spc110 [10,13,14] on the nuclear side of the SPB adopts such a closed geometry only in a minor population, in the presence of artificial crosslinks [10,13] or when capping a MT[15]. While the distinct effects of *S. cerevisiae* Spc110 and *C. albicans* Spc72 on the conformation of oligomerised γ-TuSCs may stem from the differences in CM1 motif binding, it may also reflect different regulatory mechanisms at the cytoplasmic and nuclear side of the yeast SPB. In the cytoplasm, Spc72-mediated γ-TuSC oligomerisation alone may be sufficient to obtain a MT nucleation competent γ-TuSC oligomer and regulation of MT formation could thus be achieved on the level of Spc72 recruitment to the SPB[49] or Stu2 binding to Spc72. Nuclear MT assembly, on the other hand, could require phosphorylation of Spc110 or other γ-TuSC components to achieve conformational closure of the γ-TuSC oligomer [50,51], or may only be induced during

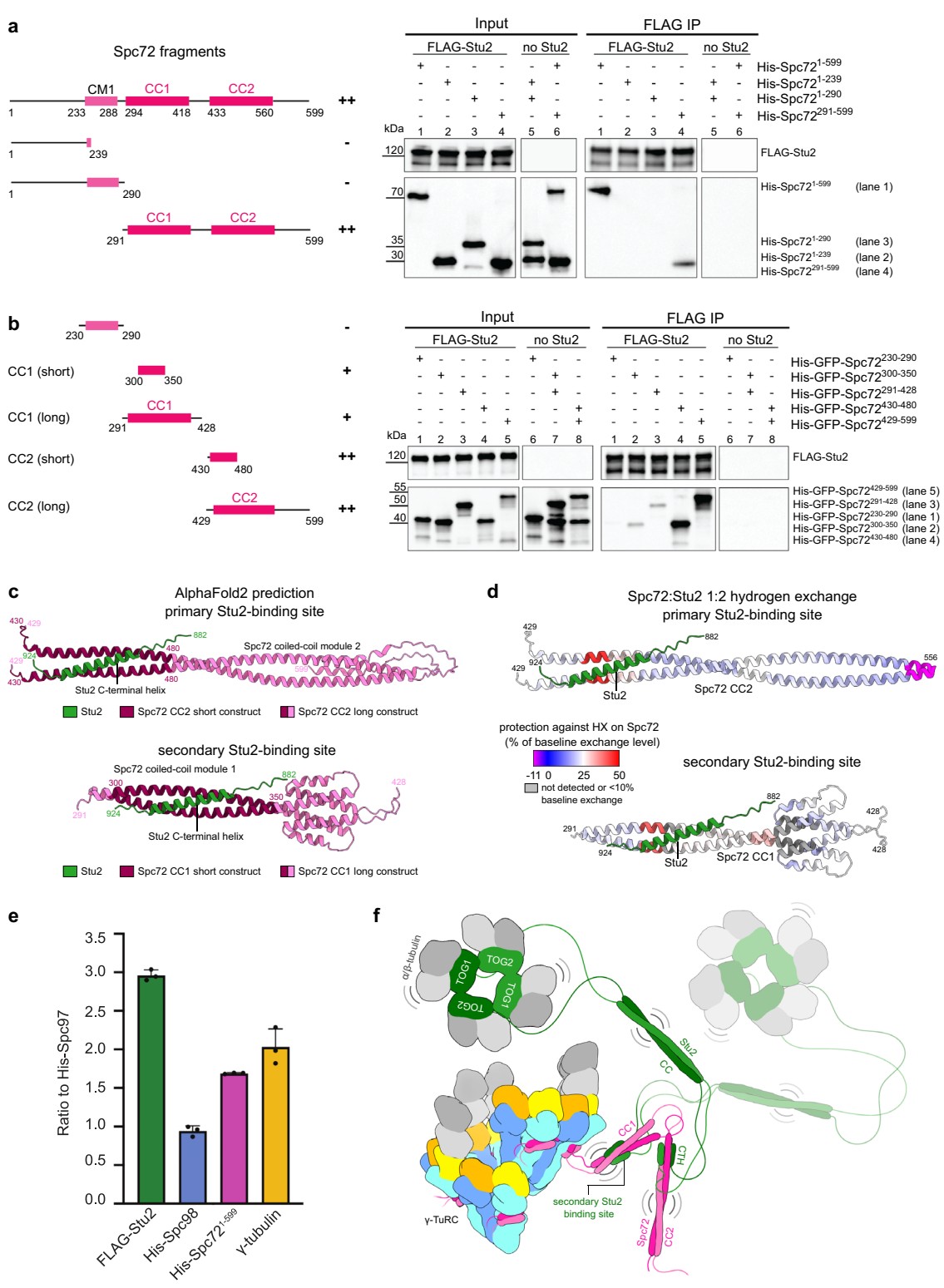

nucleation by binding of α/β-tubulin stabilised by a yet unidentified coiled-coil protein[15].

Mzt1 proteins are central structural elements in the vertebrate γ-TuRC, indispensable for correct complex assembly and stability[20,21,52]. In contrast, Mzt1 is completely absent in the genome of *S. cerevisiae*[20], representing an organism with a strongly reduced MT nucleation system. While Mzt1 is encoded in *C. albicans*, our cryo-EM reconstruction suggests that it is dispensable for the formation of stable γ-TuSC rings with MT-compatible geometry. This indicates that in *C.*

*albicans*, Mzt1 may mostly function in γ-TuSC recruitment to the SPB, as is the case in *Aspergillus nidulans*[53].

In *C. albicans*, the Spc72 CM1 motif binds to the γ-TuSC as a dimer. Sequence and structural analyses indicate that dimeric CM1 motif binding to γ-TuCs is a deeply conserved feature across evolution, whereas monomeric CM1 motif binding in Spc110 of *S. cerevisiae* is unique among all organisms included in the analysis. Conservation of dimeric CM1 motif binding raised the question of what role the outer α-helix of the dimeric CM1 motif (CM1$_{out}$) may play. Our structural and

**Fig. 6 | The C-terminal α-helix of Stu2 has two binding sites on coiled-coil modules of Spc72. a, b** (Left) Schemes highlighting different functional domains of Spc72 used for immunoprecipitation experiments. (Right) Anti-FLAG immunoprecipitation experiments with or without FLAG-Stu2 and different Spc72 fragments. Plus (+) and minus (-) indicate the ability of each construct to be pulled down by Stu2. *N* = 2 biologically independent experiments. Corresponding input and FLAG IP lanes were labelled with the same number to aid comparison. **c** Highest-ranked AlphaFold2 predictions consistent with the primary and secondary Stu2 (green) binding site on Spc72 coiled-coil modules 2 and 1, respectively. Only Spc72 sequences used in pull-down experiments are shown. Colouring scheme as indicated. **d** Protection against hydrogen exchange of His-Spc72[291-599] in the presence of Stu2 (His-Spc72[291-599]:FLAG-Stu2 1:2 molar ratio), as detected using HX-MS. Spc72[429-556] (upper panel) and Spc72[291-428] (lower panel) are coloured according to the hydrogen exchange in the presence of FLAG-Stu2 as a percentage of the hydrogen exchange in the absence of FLAG-Stu2. Only peptides with a minimum of

10% baseline exchange are shown. Stu2 is coloured in green; Spc72[291-599] is visualised using the AlphaFold2 prediction of the primary Stu2 binding site. For the secondary binding site, Stu2 was placed on Spc72 CC1 by aligning CC1 in predictions of the primary and secondary binding site. **e** Ratio of FLAG-Stu2, His-Spc98, His-Spc72[1-599], and γ-tubulin to His-Spc97 in the oligomeric γ-TuSC/FLAG-Stu2/His-Spc72[1-599] complex, quantified based on SDS-PAGE gel densitometry (fraction 8.5, as shown in Supplementary Fig. 1d) and normalised for protein molecular weight. *N* = 3 biologically independent experiments. Data are shown as mean ± SD. **f** Model for the architecture of the MT nucleation unit, highlighting the recruitment of the C-terminal helix of Stu2 (CTH) on flexibly-linked coiled-coil modules (CC1 and CC2) of Spc72 to allow the delivery of α/β-tubulin by the TOG domains[80] (TOG1 and TOG2) of Stu2, which are linked to the CTH by its dimerising coiled-coil (CC). The second dimer of Stu2 is shown in semi-transparent. Source data are provided in the Source Data file.

MST data indicate that CM1[out] is most likely important for high-affinity γ-TuSC binding of CM1 motifs, and as a consequence for γ-TuSC oligomerisation and in vivo function. Notably, *S. cerevisiae* Spc110, the only protein for which monomeric CM1 motif binding was observed, contains a loop and a coiled-coil region C-terminal to the CM1 motif, both of which bind to Spc97 GRIP1[10] and may thus compensate for the loss of CM1[out] in high-affinity γ-TuSC binding. Additionally, CM1 motif dimerisation may impact the mechanical properties of the inner CM1 helix (CM1[in]), e.g., by increasing its rigidity and stiffness, which may be relevant for induction of conformational changes in the γ-TuSC. This would be consistent with a model in which the CM1 motif of mammalian CDK5RAP2 acts as part of a molecular wedge that forces the γ-TuSC into a closed conformation, as proposed by Xu et al[22]. Independent of a function in γ-TuSC oligomerisation, CM1[out] may also broadly serve as a binding site for recruitment of factors, as observed for the GCP2/Mzt2 module in the human γ-TuRC[21].

Beyond dimerisation at the level of the CM1 motif, higher-order oligomerisation of γ-TuC receptor proteins imposed by the SPB components Spc42[54] or Nud1[55] likely additionally contributes to γ-TuSC oligomerisation at physiological γ-TuSC concentrations[56]. Therefore, unlike in mammalian cells where γ-TuRC assembles in the cytoplasm without the assistance of a CM1-containing protein, γ-TuSC oligomerisation in yeast occurs exclusively within the specialised environment of the SPB, facilitated by layers of CM1-associated proteins.

In addition to CM1-containing proteins, MT polymerases of the Stu2/XMAP215/chTOG family promote MT nucleation throughout eukaryotes, but the mechanism of their recruitment to MT nucleation centers has remained unclear. We reveal that the evolutionarily conserved C-terminal α-helix of Stu2 binds to coiled-coil regions of Spc72 adjacent to the SPB binding domain using systematic structure-guided pull-down experiments, site-directed mutational analysis and HX-MS. The targeting function of the C-terminal conserved helix in Stu2/XMAP215/chTOG family proteins is likely conserved. This is supported by observations that the larger C-terminal region of human chTOG, which includes the conserved C-terminal α-helix, is essential in localising chTOG to interphase centrosomes, among other cellular locations[32]. Similarly, a larger C-terminal fragment of *Xenopus laevis* XMAP215 binds to the γ-TuRC, suggesting that the C-terminal α-helix may also be involved in recruitment in vertebrates.

The Stu2-binding regions on Spc72 identified by pull-down experiments were not resolved in our cryo-EM reconstruction, indicating that they are flexibly associated with the CM1 motif. This enhanced flexibility, complemented by disordered linkers within Spc72 and Stu2, may augment the functionality of Stu2 by expanding the range of accessible sites to which it can deliver α/β-tubulin subunits, thereby promoting MT nucleation. This function may be further enhanced by the presence of multiple Stu2-interacting sites on Spc72, allowing the recruitment of more than one Stu2 dimer and thereby locally increasing α/β-tubulin subunit concentration at the γ-TuRC. Recruitment of Stu2

to multiple Spc72 binding sites flexibly linked to the γ-TuSC at different effective distances may also benefit the recruitment of α/β-tubulin subunits at different stages of MT nucleation, where the necessity for α/β-tubulin delivery to γ-tubulin subunits is superseded by the need for α/β-tubulin delivery to the first layers of the nascent MT.

Interestingly, studies have revealed that a conserved region of both XMAP215 and Stu2, N-terminal to the α-helix interacting with Spc72, can directly bind to γ-TuCs[29,31]. However, no density for Stu2 was observed on the oligomerised γ-TuSC in our cryo-EM reconstruction. Although Stu2 dissociation or destabilisation during cryo-EM sample preparation cannot be excluded entirely, this implies that additional interactions between Stu2 and the γ-TuRC could form at distinct functional stages in the MT nucleation process, e.g., upon the association of the initial αβ-tubulin subunits with the γ-TuRC[15].

In conclusion, our work provides insights into binding of universal CM1-containing MT-nucleation-promoting factors and the localisation of XMAP215/chTOG/Stu2 family MT polymerases to MT nucleation sites. This enhances our fundamental understanding of how CM1-containing proteins, MT polymerases and γ-TuCs cooperate to nucleate MTs.

## Methods

### Molecular cloning

DNA fragments coding for *SPC72* (orf19.6583) Arg410 to Asp1006 and full-length *STU2* (orf19.6610) Met1 to Glu924 were codon-optimised and synthesised by Integrated DNA Technologies (IDT, USA). *SPC72* and *STU2* were individually cloned into plasmids pET28b (NdeI/XhoI sites) and pFastBac1 (BamHI site), respectively. DNA fragments coding for Stu2[882-924] (wild-type) and the Stu2[882-924LIM] (L906A, I910A, M913A) mutant with a FLAG tag followed by an HRV 3 C cleavage site on the 5' end, as well as gene fragments for the *SPC72[300-350ELLY]* (E317R, L319A, L321R, Y326A) mutant and the *SPC72[430-480EDID]* (E455A, D456A, I458A, D462A) mutant were synthesised by Twist Bioscience (TWIST, USA). FLAG-tagged *STU2[882-924]* (wild-type) and *STU2[882-924LIM]* DNA fragments were cloned into a pETM41 backbone and *SPC72[300-350ELLY]* and *SPC72[430-480EDID]* DNA fragments were cloned into a pET28b backbone with NEBuilder® HiFi DNA Assembly Master Mix (New England Biolabs®) to obtain MBP-FLAG-Stu2 and His-GFP-Spc72 constructs.

For constructs used for insect cell expression, a FLAG tag followed by an HRV 3C cleavage site was introduced into the 5' end of *STU2* by PCR (Q5 DNA Polymerase, NEB) and then cloned into the pIDC vector of the MultiBac™ system (GENEVA Biotech) using NEBuilder® HiFi DNA Assembly Master Mix (New England Biolabs®). His-tagged *SPC97*, His-tagged *SPC98*, and *TUB4* (γ-tubulin) were amplified from constructs used in a previous study[11], *Spc72[1-599]* was amplified from pET28b-*SPC72* and then, the gene fragments were assembled into pACEBac1 and pIDK vectors separately as described above. Plasmids were then cre-recombined (Cre-recombinase, NEB) one by one and assembled into one construct according to the MultiBac™ manual (GENEVA Biotech).

Gene fragments of *SPC72*[1-599] carrying point mutations T234P/T236A (termed PA mutant) or L243R/l247R/L254R (termed 3R mutant) were each cloned with the polyhedrin expression cassette into the pIDK-*TUB4* plasmid by using the construction strategy described earlier[57]. For experiments involving Mzt1, a FLAG tag followed by an HRV 3 C cleavage site was fused to the N-terminus of SPC97 via PCR. Afterwards, FLAG-tagged *SPC97* and *SPC98* genes were cloned into a pACEBac1 vector, His-tagged *SPC72*[1-599] and *TUB4* genes were cloned into a pIDK vector and GFP-tagged *MZT1* gene was cloned into a pIDC vector. GFP-tagged *MZT1* was PCR amplified from pET28b-*GFP-MZT1*[20]. Plasmids were then Cre-recombined step by step to produce the following constructs: (1) pACEBac1 *His-SPC97*, *His-SPC98*; pIDK *TUB4*, *His-SPC72*[1-599]; pIDC *FLAG-STU2*, (2) pACEBac1 *His-SPC97*, *His-SPC98*; pIDK *TUB4*, *His-SPC72*[1-599,PA]; pIDC *FLAG-STU2*, (3) pACEBac1 *His-SPC97*, *His-SPC98*; pIDK *TUB4*, *His-SPC72*[1-599,3R]; pIDC *FLAG-STU2*, (4) pACEBac1 *FLAG-SPC97*, *SPC98*; pIDK *TUB4*, *His-SPC72*[1-599]; pIDC *His-GFP-MZT1*.

For pull-down experiments of Stu2 and Spc72, FLAG-tagged wild-type *STU2*, the Stu2 coiled-coil (666-768aa) deletion mutant S*tu2*[Δ666-768] and the C-terminal helix (894-924aa) deletion mutant *Stu2*[Δ894-924] genes were cloned into pFastBac1 vector (BamHI site) using the NEBuilder® HiFi DNA Assembly Master Mix (New England Biolabs®). Pull-down experiments of γ-TuSC and Spc72[1-599]/Spc72[1-599,3R], a FLAG-tagged *SPC98* gene was cloned into a pFastBac1 vector (BamHI site) using the NEBuilder® HiFi DNA Assembly Master Mix (New England Biolabs®).

For His-MBP-Spc72 constructs (MBP-CM1[231-268]; MBP-CM1[231-268,3R]), a pETM41 backbone and Spc72 templates were used for PCR amplification and a NEBuilder® reaction as described for cloning of insect cell expression constructs, with primers (synthesised by Merck) listed in Supplementary Table 2. To generate GST-Spc72 constructs (CM1[231-268]; CM1[231-268,3R]), the pETM41 His-MBP plasmid was used as backbone and the GST sequence was amplified from a pGEX plasmid, exchanging the His-MBP sequence with GST. Additionally, for the Spc72 mutants used for *E. coli* expression, PCR-amplified products of *SPC72* containing codons 1-239, 1-290, 291-559 and 1-599 were subcloned into double-digested pET28b vector (NdeI and XhoI) with the NEBuilder® HiFi DNA Assembly Master Mix (New England Biolabs®). A 6×His tag was on the N-terminus of the Spc72 constructs (His-Spc72).

Moreover, PCR-amplified products of *SPC72* containing codons 230-290, 291-428, 300-350, 429-599, and 430-480 with a GFP tag on the 5' end were subcloned into pET28b[20] with the NEBuilder® HiFi DNA Assembly Master Mix (New England Biolabs®) to obtain His-GFP-Spc72 constructs. For the plasmids used in the yeast experiment, the PCR-amplified products of *SPC72* (from *S. cerevisiae*) and its mutants of *spc72*[ΔP55-N62] or *spc72*[3R] were subcloned and assembled onto pRS305, pRS315 or pRS316-*GAL1* backbones. For the *SPC72* gene tagging with *IAA7*, primers S2-Spc72 and S3-Spc72 were used to amplify the *IAA7* gene fragment from the plasmid of pFA6a-*IAA7-3xFLAG-Tubc6-kanMX* (obtained from Dr. M. Knop, ZMBH; *Tubc6* stands for "Terminator of the *UBC6* gene"). Used primers (synthesised by Merck) are listed in Table S2; plasmids are listed in Table S3.

## Protein expression in insect cells and purification

MultiBac constructs containing γ-TuSC subunits, *SPC72*, *STU2* or *MZT1* were transformed into DH10EmBacY competent cells (obtained from Prof. Imre Berger, Grenoble) and recombinant bacmid DNA was produced as described in the MultiBac™ manual (GENEVA Biotech). For Stu2-Spc72 pull-down experiments, pFastBac1-*FLAG-STU2*, pFastBac1-*FLAG-STU2*[Δ666-768] and pFastBac1-*FLAG-STU2*[Δ894-924] bacmid DNA, and for the FLAG-γ-TuSC and Spc72[1-599,3R] pull-down experiment, pFastBac1-*FLAG-SPC98*, pFastBac-*His-SPC97*[11] and pFastBac1-*TUB4*[11] bacmid DNA was produced according to the Bac-to-Bac Baculovirus Expression System manual (ThermoFisher Scientific). The purified bacmid DNA was transfected into Sf9 cells with the Cellfectin™ II Reagent (ThermoFisher Scientific) to obtain P1 baculovirus 72 h post-transfection at 27 °C. For protein expression, P2 baculovirus was amplified in 25 ml ($1 \times 10^6$ cells/ml), harvested and added to Sf21 cells ($1.5–2.0 \times 10^6$ cells/ml) at a ratio of 1:100 (v/v). Cells were harvested 60 h post-infection, pelleted, snap frozen in liquid nitrogen, and then stored at -80 °C. Insect cells were cultured in Sf-900 III medium supplemented with 100 units/ml penicillin/100 μg/ml streptomycin (ThermoFisher Scientific).

For protein purification of γ-TuSC/His-Spc72[1-599]/FLAG-Stu2 or FLAG-γ-TuSC (FLAG-Spc97)/His-Spc72[1-599] with or without His-GFP-Mzt1, the cell pellet was resuspended in TBS buffer (50 mM Tris-HCl, pH 7.4; 150 mM NaCl, 1 mM $MgCl_2$, 1 mM EGTA) supplemented with 1 mM DTT, 0.05% Tween-20 (vol/vol), and one complete EDTA-free protease inhibitor tablet (PI tablet, Roche) per 50 ml, lysed by sonication (Polytron PT3100, 4000 rpm for 4 min) and incubated with Benzonase (1:500 vol/vol) for 20 min. After centrifugation of the lysate at $245000 \times g$ for 35 min (Sorvall Discovery 90SE ultracentrifuge, 50.2 Ti rotor), the supernatant was incubated with anti-FLAG M2 Affinity Gel (Sigma-Aldrich) for 2 h at 4 °C, and protein was eluted with 500 μg/ml 3 × FLAG peptide (Gentaur) for 20 min twice after washing with wash buffer (50 mM Tris-HCl pH 7.4, 150 mM NaCl, 1 mM $MgCl_2$, 1 mM EGTA and one PI tablet per 50 ml) three times. The elution was analysed by SDS-PAGE. Glycerol was added to the purified protein to a final concentration of 2.5%, flash frozen in liquid $N_2$ and stored at −80 °C until further use.

For FLAG-Stu2, 800 ml pellet from insect cell expression was resuspended in lysis buffer (50 mM Tris-HCl pH 7.4, 400 mM NaCl, 1 mM EGTA, 1 mM $MgCl_2$, 1 mM DTT, 0.25% Brij35, 1 mM PMSF and one complete protease inhibitor tablet (Roche). FLAG purification was performed as described for Stu2-containing insect cell constructs. Afterwards, FLAG elutions were pooled and loaded on Capto Hires Q (5/50, Cytiva) equilibrated with Buffer A (50 mM Tris-HCl pH 7.4; 50 mM NaCl, 0.5 mM EGTA; 1 $MgCl_2$) for anion exchange chromatography (AEC). Complexes were eluted with a gradient to 100 % Buffer B (50 mM Tris pH 7.4, 1 M NaCl, 0.5 mM EGTA; 1 $MgCl_2$) over 20 column volumes. After AEC, peak fractions of FLAG-Stu2 were concentrated and buffer exchanged to 50 mM Tris-HCl pH 7.5, 150 mM NaCl, 1 mM $MgCl_2$ using Amikon-Ultra-0.5 (50 KDa, Merck). For control SEC experiments (Supplementary Fig. 1e, f), γ-TuSC (His-tagged *SPC97*, His-tagged *SPC98*, and *TUB4*) was purified from 400 ml of cell pellet as described previously[11], using His-affinity purification (Protino® Ni-TED Resin, MACHEREY-NAGEL).

## Protein expression in *E. coli* and purification

Truncated and point-mutated *SPC72* and *STU2* variants (*STU2* variants used in Supplementary Fig. 10d), and His-GFP-Mzt1[20] used for pull-down experiments were expressed in *E. coli BL21 CodonPlus-RIL* (Stratagene) in cultures up to 100 ml (proteins used in Supplementary Fig. 10d) or 1 l (proteins used in Figs. 4f, 6a, b, Supplementary Fig. 1b, Supplementary Fig. 10b, c) using LB medium (Roth). Cells were grown at 37 °C until reaching $OD_{600}$ values of 0.6-0.8, as checked using an Ultrospec 2100 pro (Amersham Biosciences). After 15 min incubation at room temperature, protein expression was induced with 0.5 mM isopropyl-β-D-thiogalactopyranoside (IPTG) and proteins were expressed at 18 °C for 20 h. After induction, cells were harvested by centrifugation (Sorvall RC 6, Thermo Scientific) at $20000 \times g$ for 10 min. Cell pellets were flash-frozen and stored at −80 °C.

For the His purification of His-Spc72, His-GFP-Spc72 truncated proteins and His-GFP-Mzt1 used in Figs. 4f, 6a, b, Supplementary Fig. 1b, Supplementary Fig. 10b, c, 1 l *E. coli* cells were sonicated (3 × 1 min with 0.8 amplitude, Hielscher UP50H) in lysis buffer (50 mM Tris-HCl pH 7.4, 200 mM NaCl, 1 mM EDTA, 1 mM EGTA) supplemented with 1 mM DTT, 1 mM PMSF and one PI tablet per 50 ml and then centrifuged at $20000 \times g$ for 30 min at 4 °C. After centrifugation, the supernatant was incubated with Ni-TED resin (Macherey-Nagel, 0.5 g resin per liter of cell pellet) for 1 h, washed 3 times with wash buffer (50 mM Tris-HCl pH 7.4, 200 mM NaCl, 1 mM EDTA, 1 mM EGTA, one PI tablet per 50 ml) and

then, protein was eluted using the elution buffer (50 mM Tris-HCl pH 7.4; 200 mM NaCl, 1 mM EDTA, 1 mM EGTA, one PI tablet per 50 ml and 300 mM imidazole). Protein expression and purification were confirmed with SDS-PAGE and Coomassie blue staining. Purified protein was flash-frozen in liquid $N_2$ and then stored at -80 °C until use.

1 l $E.$ $coli$ expression cultures of $MBP\text{-}CM1^{231\text{-}268}$, $MBP\text{-}CM1^{231\text{-}268,3R}$, $GST\text{-}CM1^{231\text{-}268}$, $GST\text{-}CM1^{231\text{-}268,3R}$ and $His\text{-}Spc72^{291\text{-}599}$ constructs were resuspended in 30-35 ml cold lysis buffer (50 mM Tris-HCl pH 7.4, 300 mM NaCl, 15 mM imidazole, 1 mM $MgCl_2$, 1 mM EGTA) supplemented with 1 mM DTT and 1 mM PMSF. Resuspended cells were sonicated ($6 \times 1$ min with 100% amplitude; Hielscher UP50H) and the lysate was cleared by centrifugation at $20000 \times g$ for 30 min at 4 °C using an SS-34 rotor (Thermo Fisher Scientific). Ni-NTA beads (Qiagen) or Glutathione beads (Protino Agarose 4B) were equilibrated with lysis buffer and incubated with the total soluble fraction of the lysate for 90 min rotating at 4 °C. 400 μl of beads were used for 1 l culture. After incubation, affinity beads were separated from the flow-through by centrifugation ($800 \times g$, 3 min at 4 °C) and were subsequently washed twice with lysis buffer and twice with wash buffer (50 mM Tris- HCl pH 7.4, 150 mM NaCl, 1 mM $MgCl_2$, 1 mM EGTA). Between each washing step, beads were sedimented by centrifugation ($800 \times g$, 3 min at 4 °C). Recombinant proteins were eluted three times via incubation with 1-1.25 bead volume of elution buffer (wash buffer supplemented with 400 mM imidazole for Ni-NTA beads, wash buffer supplemented with 50 mM glutathione (Thermo Fisher) for glutathione beads) for 10-20 min at 4 °C. Elution fractions were collected via centrifugation ($800 \times g$, 3 min at 4°C) and subsequently subjected to SEC.

## Size-exclusion chromatography (SEC)
Protein complexes purified from insect cells were centrifuged at 4°C for 10 min at $20000 \times g$ after thawing from −80°C. They were then injected into an SEC column (Superose 6 increase 10/300 GL, GE healthcare) equilibrated with HB100 buffer (40 mM K-HEPES pH 7.4, 100 mM KCl, 1 mM EGTA, 1 mM $MgCl_2$). SEC was performed at a constant flow of 0.25 ml/min on the ÄKTA go™ protein purification system operated with the Unicorn software (version 7.5, Cytiva). For experiments with His-, His-MBP- and GST-tagged Spc72 constructs, SEC runs were performed at a constant flow of 0.5 ml/min using an ÄKTA Pure system (Cytiva) controlled by Unicorn software (version 7.9). SEC data were analysed using the Prism software (GraphPad Prism 10).

## Size-exclusion chromatography – multi-angle light scattering (SEC-MALS)
FLAG-Stu2 (10 μM) and His-Spc72$^{291\text{-}599}$ (60 μM) were diluted 1:1 with SEC buffer (in 50 mM Tris-HCl, pH7.5, 150 mM NaCl, 1 mM $MgCl_2$) or mixed 1:1 (vol), incubated for 30 min and injected onto a Superdex 200 increase 5/150 GL gel-filtration column (Cytiva) on an Agilent 1260 Infinity II HPLC system (Agilent) equilibrated in SEC buffer at room temperature, with a flow rate of 0.3 ml/min. The column was coupled to a MALS system (MiniDAWN and Optilab, Wyatt Technology). Data were analysed using the Astra 8.2.2 software (Wyatt Technology). The generic protein dn/dc value of 0.1850 was applied.

## Mass Photometry
Mass Photometry measurements were performed using a Refeyn TwoMP mass photometer (Refeyn Ltd, Oxford, UK). Videos of 1 min with default image size were recorded using the Refeyn AcquireMP 2024 R1 software (Refeyn Ltd, Oxford, UK). For the measurement, freshly washed and dried high-precision microscope coverslips (24x50mm) were used. A silicone gasket with 6 cavities was placed on top/centre of the coverslip to form measurement holes. A total amount of 19 μl of corresponding SEC buffer for the different protein samples was applied in each gasket hole and autofocus was performed before every measurement. 1 μl of the protein at a concentration of 400 nM was diluted with the SEC buffer to below

40 nM. Data analysis and data plotting was performed using the Refeyn DiscoverMP 2024 R1 software (Refeyn Ltd, Oxford, UK). Bovin serum albumin (BSA, 66 kDa) and Immunoglobulin G (IgG, 150 kDa and 300 kDa) proteins were used to generate the standard contrast-to-mass calibration curve.

## Negative stain EM and data analysis
5 μl of purified γ-TuSC/His-Spc72$^{1\text{-}599}$/FLAG-Stu2 in TBS or HB100 buffer or γ-TuSC/His-Spc72$^{1\text{-}599}$/His-GFP-Mzt1 in HB100 buffer was applied on a glow-discharged copper-palladium 400 EM mesh grid covered with an approximately 9–10 nm-thick continuous carbon layer and incubated for 30 s at room temperature. Grids were blotted on Whatman filter paper 50 (cat no. 1450-070, Cytiva) and rinsed on 3 drops of distilled water, followed by staining with 3% uranyl acetate in distilled water. Micrographs were acquired on a Talos L120C TEM equipped with a 4 k × 4 k Ceta CMOS camera (Thermo Fisher Scientific) using the EPU software (Thermo Fischer Scientific) at an approximate defocus of -2 μm to -2.5 μm and an object pixel size of 0.2552 nm. In total, 609 images of γ-TuSC/His-Spc72$^{1\text{-}599}$/FLAG-Stu2 sample in TBS buffer, 726 images of γ-TuSC/His-Spc72$^{1\text{-}599}$/FLAG-Stu2 and 2481 images of γ-TuSC/His-Spc72$^{1\text{-}599}$/His-GFP-Mzt1 were acquired.

Analysis of all datasets was conducted in RELION 3.0 Beta[58]. Gctf was used for micrograph contrast transfer function (CTF) estimation[59]. For γ-TuSC/His-Spc72$^{1\text{-}599}$/FLAG-Stu2 samples in TBS and HB100 buffer, 3179 and 6106 particles were manually picked and extracted at a pixel size of 0.51 nm (TBS) and 0.656 nm (HB100) in boxes of 128 pixels. For γ-TuSC/His-Spc72$^{1\text{-}599}$/His-GFP-Mzt1, 1629 ring-like particles were manually selected from 2481 images and extracted at a pixel size of 0.51 nm and box size of 128 pixels. Particles were then 2D classified into 50 classes, with a mask diameter of 400 Å. After 2D classification, the best 2D classes were subjected to 3D classification into 4 classes and a regularisation parameter T of 10 (for γ-TuSC/His-Spc72$^{1\text{-}599}$/FLAG-Stu2 in TBS buffer and for γ-TuSC/His-Spc72$^{1\text{-}599}$/His-GFP-Mzt1) or 6 classes and a regularisation parameter T of 8 (γ-TuSC/His-Spc72$^{1\text{-}599}$/FLAG-Stu2 in HB100 buffer). After 3D classification, 3D auto-refinement was performed, with the human γ-TuRC (EMD-21074) as a reference. For visualisation, no masking or sharpening was applied.

## S. cerevisiae spc72$^{ΔP55\text{-}N62}$ and spc72$^{3R}$ phenotype analysis
To analyse functionality of $spc72^{ΔP55\text{-}N62}$ and $spc72^{3R}$, $LEU2$-based CEN plasmids pRS315- $spc72^{ΔP55\text{-}N62}$, pRS315-$spc72^{3R}$, pRS315-$SPC72$ and pRS315 were transformed into the $S.$ $cerevisiae$ ESM448-1 strain (S288C background) (Table S3), plated onto SC-Leu-Ura plates and incubated for 2 days at 30°C. Cells were subsequently inoculated into SC-Leu liquid medium, cultured at 30°C overnight, serially diluted and dropped onto SC-Leu and 5-FOA plates, which were incubated at 30°C and 23°C for 2 or 3 to 6 days. A similar drop test was performed with $S.$ $cerevisiae$ YJP287-1 (W303 background) (Table S3) containing the $LEU2$-based integration plasmids pRS305-$spc72^{3R}$, pRS305-$SPC72$ and pRS305 at 30°C, 23°C and 16 °C for 2, 3 and 5 days, respectively.

The phenotype of $spc72^{ΔP55\text{-}N62}$ was analysed by integrating plasmids pRS305-$spc72^{ΔP55\text{-}N62}$, pRS305-$SPC72$ and pRS305 (HpaI digest) into the genome of an $S.$ $cerevisiae$ YJP287-1 strain (see above). Transformants were plated onto SC-Leu-Ura plates for 2 days at 30 °C, followed by incubation on 5-FOA plates to remove pRS316-$SPC72$. For MT phenotype analysis, $SPC72$ and $spc72^{ΔP55\text{-}N62}$ cells were cultured in SC-Leu (+ adenine) liquid medium at 16°C for 2 days and stained with DAPI to visualise DNA or analysed by fluorescence microscopy (DeltaVision microscope, see below) for GFP-Tub1 and Spc42-mCherry.

To test whether the $spc72^{3R}$ mutation is as a dominant or recessive mutation, pRS316-$GAL1$-$SPC72$, pRS316-$GAL1$-$spc72^{3R}$, and pRS316-$GAL1$ plasmids were transformed into the $S.$ $cerevisiae$ strain ESM356-1 (Table S3) and plated on SC-Ura plates at 30 °C for 2 days. The transformants were cultured in SC-Ura/raffinose liquid medium overnight. Cell densities were adjusted to $OD_{600}$ value of 1. Cells were

10-fold serially diluted and plated on raffinose-Ura plates containing various galactose concentrations (0%, 1%, and 2%). Plates were then incubated at 30 °C for 2 days. For documentation of cell growth, a CanoScan 5600 F scanning machine running MP Navigator EX - CanoScan 5600 F software (Canon U.S.A., Inc.) was used for imaging the dish plates.

For spc72[3R] mutant cell MT phenotype analysis, auxin-inducible degron (AID) technology was utilised, in which *S. cerevisiae SPC72* was tagged with an IAA7 tag on the C-terminus. After treatment with 0.5 mM IAA (3-Indoleacetic acid, Sigma-Aldrich I2886) at 30 °C for 0, 1, 2, 3, and 4 h, depletion efficiency was analysed by Western blotting using home-made anti-Spc72 antibodies[29]. Protein extracts from yeast cells were prepared using trichloroacetic acid (TCA)[60]. After confirming that the degron-based Spc72-IAA system was working, pRS315-*SPC72*, pRS315-*spc72[3R]* and pRS315 plasmids were transformed into strain YAJZ023 (Table S3). Transformants were selected on SC-Leu plates. Colonies were used for depletion experiments at 30 °C. 0.5 mM IAA was added for the degradation of the endogenous Spc72-IAA7 and after 0 and 3 h IAA treatment, cells were analysed by microscopy. For microscopy, 3 µl of cells were dropped onto glass slides and then covered with a 35-mm glass dish. Images were acquired with a DeltaVision RT system (Applied Precision) on an Olympus IX71 microscope equipped with 100X NA UPlanSAPO objective lens (Olympus) with the same exposure time and illumination settings and 2 × 2 binning. For DAPI-stained cells, the DAPI channel was used and for Spc42-mCherry and GFP-Tub1 analysis the TRITC/FITC channels were selected. Image processing and analysis were performed with the open-source software package ImageJ 1.46r (National Institutes of Health)[61].

## Microscale thermophoresis (MST)

The Monolith NT.115 instrument (NanoTemper Technologies) was used for the MST measurements. The analyte was serially diluted in a 1:1 ratio (vol/vol) 15 times to obtain 16 measurement points and a constant concentration of fluorescein-labeled ligand was added. To measure the binding affinity with Spc72, purified γ-TuSC was fluorescently labeled using an amine-reactive protein labeling kit (GREEN-NHS; NanoTemper Technologies). Increasing concentrations (~1 nM to ~50 µM) of recombinant GST-CM1[231-268] and GST-CM1[231-268,3R] were titrated against 100 nM labeled γ-TuSC. Experiments were performed in TBS buffer supplemented with 0.05% (wt/vol) Tween-20. Samples were loaded into standard glass capillaries (Monolith NT capillaries; NanoTemper Technologies). MST assays were performed with 20% LED power and 40% MST power by using a green filter. The normalised fluorescence readings (thermophoresis plus T-jump) were plotted to generate the binding curves and the dissociation constant $K_D$ was determined using the NanoTemper software.

## Pull-down experiments

To map Stu2-Spc72 interaction regions, 1 ml of insect cell pellet from FLAG-tagged Stu2, Stu2[Δ666-768] and Stu2[Δ894-924] samples was used for each reaction. The cell pellet was lysed in lysis buffer (50 mM Tris-HCl pH 7.4, 200 mM NaCl, 1 mM EGTA, 1 mM MgCl₂, PMSF (1:100 vol/vol), Brij35 (0.25% vol/vol) and one PI tablet per 50 ml). To map the influence of site-directed mutations on Stu2-Spc72 interactions, 10 ml of *E. coli* expressed cell pellet from MBP-FLAG-tagged Stu2[882-924] and Stu2[882-924LIM] samples were used as bait. Correspondingly, 10 ml of His-GFP-tagged Spc72[300-350], Spc72[300-350ELLY], Spc72[430-480EDID], and Spc72[430-480] samples from *E. coli* expression were used as prey. The cell pellet was lysed in lysis buffer (50 mM Tris-HCl pH 7.4, 200 mM NaCl, 1 mM EGTA, 1 mM MgCl₂, PMSF (1:100 vol/vol), Tween-20 (0.1% vol/vol) and one PI tablet per 50 ml). To test the binding of the Spc72[3R] mutant to FLAG-tagged γ-TuSC, 1 ml of cell pellet from FLAG-tagged γ-TuSC (FLAG-Spc98) sample was used for each reaction. The cell pellet was lysed in lysis buffer (50 mM Tris-HCl pH 7.4, 150 mM NaCl, 1 mM EGTA, 1 mM

MgCl₂, PMSF (1:100 vol/vol)), Tween-20 (0.1% vol/vol) and one PI tablet per 50 ml).

For both experiments, cells were then sonicated (3 × 1 min with 0.8 amplitude, Hielscher UP50H) and centrifuged at 20000 × *g* for 30 min at 4 °C. The supernatant was incubated with Anti-FLAG M2 Affinity Gel (Sigma-Aldrich) and incubated for 2 h at 4 °C. After incubation, affinity gel was washed 3 times with wash buffer (50 mM Tris-HCl pH 7.4; 200 mM NaCl, 1 mM EGTA, 1 mM MgCl₂, one PI tablet per 50 ml for Stu2-Spc72 interactions, same buffer with 150 mM NaCl for γ-TuSC-Spc72[3R] interactions) before the addition of the Spc72 fragments (20 µg) or with the same volume of wash buffer (as negative control). As a complementary negative control, meanwhile, the same amount of Anti-FLAG M2 Affinity Gel was used and incubated with Spc72 fragments (20 µg) in the absence of FLAG-Stu2. Then, the mixtures were incubated for 1 h. For elution, wash buffer was supplemented with 300 µg/ml 3 × FLAG peptide. Eluted proteins were separated by 4-20% precast polyacrylamide gels (Bio-Rad), Western blotted with their corresponding antibodies and visualised with a LAS-3000 imaging system (Fujifilm Life Science). Antibodies used in this study: anti-FLAG (mouse; clone 9A3, 1:1000; Cell Signaling); anti-penta-His (mouse, 1:2000; Qiagen 34660), anti-GAPDH (mouse, 1:10000; Proteintech®), anti-N-Spc72 (rabbit; 1:300; home-made, *S. cerevisiae* Spc72) and anti-γ-tubulin (guinea pig; 1:1000; home-made, *C. albicans* γ-tubulin (against 331-498aa fragment) was sent to Eurogentec for antibody production). Anti-Mouse HRP-conjugated IgG (H + L) (donkey, 1:5000, JacksonImmunoResearch EUROPE LTD.); Anti-Guinea pig HRP-conjugated IgG (H + L) (donkey, 1:5000, JacksonImmunoResearch EUROPE LTD.); Anti-Rabbit HRP-Conjugated IgG (H + L) (donkey, 1:5000, JacksonImmunoResearch EUROPE LTD.).

## Cryo-EM sample preparation and data acquisition

R2/1 Cu 200 mesh grids (Quantifoil) were glow-discharged for 1 min in a PELCO easiGlow. 4 µl of sample was applied to the grid inside the chamber of a Vitrobot Mark IV (Thermo Fisher Scientific, Eindhoven), operated at 4 °C and a relative humidity of 100%, blotted from both sides for 5 s with blot force 5 after 10 s of waiting and, after 1 s of draining, plunge-frozen into liquid ethane.

All data was acquired using EPU (Thermo Fisher Scientific) on a 300 kV Titan Krios (Thermo Fisher Scientific/FEI, Eindhoven) equipped with a K3 camera and a Quantum Gatan Imaging Filter (Gatan) operated at 20 kV energy slit width. An initial dataset of 2231 movies were acquired at a magnification of 33000 x (2.54 Å/px) in 50 fractions at an exposure rate of 24 e⁻/px/s and a cumulative electron exposure of 43 e⁻/Å², with a nominal defocus of -1 µm to -3 µm at a 0.5 µm increment. Two more datasets of 15871 and 17910 movies were acquired at 81000 x magnification (1.07 Å/px) with a 100 µm objective aperture at exposure rates of 16 e⁻/px/s and 17 e⁻/px/s for dataset 1 and 2, respectively, resulting in a cumulative electron exposure of 47 e⁻/Å² spread over 50 fractions. Movies were acquired at a nominal defocus of -1 µm to -2 µm (dataset 1) or -1.6 µm to -2.6 µm (dataset 2) in steps of 0.5 µm.

## Cryo-EM data processing

The initial steps of processing were performed in RELION 3.1[58], unless mentioned otherwise. All micrograph movies were motion-corrected with MotionCor2[62], using 5 × 5 patches, and CTF parameters were estimated using Gctf[59].

The initial dataset acquired at 2.54 Å/px was solely used for generating 2D templates for auto-picking and a reference map for 3D classification of the two datasets at higher magnification (Supplementary Fig. 4a). To this end, 331 particles were manually identified on 50 randomly selected micrographs, extracted at 5.08 Å/px and used as input for training a resnet8 model using Topaz[63]. Topaz was used to pick 48852 particles from the full dataset of 2231 micrographs, which were extracted in RELION 3.1 and subsequently classified in 2D using CryoSPARC 3.2.0[64]. Particles from classes representing higher

oligomers of γ-TuSCs were then selected. For obtaining 2D templates for auto-picking, particles were converted back to RELION format using csparc2star.py (part of the pyEM package[65]) and star.py, re-extracted and classified in 2D in RELION 3.1. For generating a reference map of oligomeric γ-TuSC for 3D classification, particles were subjected to a round of ab initio reconstruction into two classes, followed by heterogeneous refinement in two classes and finally homogeneous refinement (this reconstruction is referred to as 'the initial reference' below), all in CryoSPARC.

On the high magnification datasets, 575310 (dataset 1) and 819738 particles (dataset 2) were identified using auto-picking (Supplementary Fig. 4b). In addition, for dataset 1, 316371 particles were picked with a resnet8 Topaz model, trained with 602 particles that had been manually picked on 50 randomly select micrographs. The three sets of picked particles were separately extracted at 4.28 Å/px.

Using the reference map generated as described above, 3D classification was then separately performed on the auto-picked particles (1 round) and Topaz-picked particles (2 rounds) of dataset 1 with a spherical reference mask. The best class of both auto-picked particles and Topaz-picked particles was joined, duplicates were removed and the remaining particles were then 3D classified together with a spherical reference mask, resulting in 20122 particles for the class that was retained for further processing. These particles were subjected to 3D auto-refinement, recentered and re-extracted at 2.14 Å/px. For the auto-picked particles of dataset 2, three random subsets were subjected to 3D classification with a spherical reference mask, the best class of each subset joined and subjected to another round of 3D classification with a spherical reference mask. The 27114 particles in the class that was used for further processing were then recentered and re-extracted at 2.14 Å/px. For both datasets, the retained particles were then separately subjected to 3D auto-refinement. Reference masks for 3D refinement were generated by extending and softening a low-pass filtered version of the initial reference by a number of pixels, unless otherwise noted. All 3D refinement steps were performed using solvent flattening. All subsequent processing was then done in RELION 3.0, where another round of 3D auto-refinement was performed, followed by Bayesian polishing[66] and another round of 3D auto-refinement.

As each particle represents an oligomer of γ-TuSCs of different sizes up to this point, each particle was re-extracted 5 times at 2.14 Å/px, each time re-centering a different one of the 5 best resolved γ-TuSC units to the box center, similarly to symmetry expansion. For the γ-TuSC unit that displayed substoichiometry, particles containing this unit were first enriched by focused 3D classification without sampling. Reference masks for focused 3D classifications of γ-TuSC units were generated by extending and softening a low-pass filtered version of the 3D refinement output, segmented to keep only the central γ-TuSC unit and resampled to the appropriate position, by a number of pixels. This was followed by local refinement and re-extraction with recentering (dataset 1) or re-extraction directly following 3D classification (dataset 2); for the other 4 γ-TuSC units, re-extraction with recentering directly followed aforementioned global 3D auto-refinement. This resulted in 97040 (dataset 1) or 126685 (dataset 2) particles of centered γ-TuSC units. After performing 3D auto-refinement separately for both datasets, they were joined and the resulting 223725 particles were refined together, from here on using a reference mask generated from a segmented and low-pass filtered version of the initial reference that was extended and softened. Afterwards, particles were recentered and re-extracted at full spatial resolution (Supplementary Fig. 4c). Particles were then subjected to 3D auto-refinement, Bayesian polishing and 3D auto-refinement, followed by focused 3D classification on the centered γ-TuSC unit, retaining 129908 particles. The remaining particles were again subjected to several cycles of 3D auto-refinement, CTF refinement (per-particle defocus and beamtilt)[67] and Bayesian polishing. After a final round of 3D refinement, the final density was sharpened by

post-processing (using a solvent mask generated from the output of 3D refinement, segmented to cover exactly one γ-TuSC, low-pass filtered, extended and softened) and subsequently filtered to local resolution, resulting in a reconstruction of a single γ-TuSC within larger oligomers at 3.6 Å (Supplementary Fig. 3c). In this reconstruction, inter-γ-TuSC interactions to neighbouring spokes are clearly resolved to side chain resolution (Supplementary Fig. 3b).

To assess the geometry of the full, 14-spoked rings of γ-TuSC units (Supplementary Fig. 4d), the merged set of 223725 particles from both datasets was CTF refined after Bayesian polishing. Per higher oligomer of γ-TuSC, one particle was retained, and 3D refinement was performed with a solvent mask for a full ring of γ-TuSC units. Subsequently, multiple rounds of focused 3D classification on substoichiometric γ-TuSCs and 3D refinement were performed, retaining a set of 8261 particles of a full 14-spoked ring of γ-TuSC units and reaching 8.2 Å resolution (Supplementary Fig. 3e). Finally, multibody refinement[68] was performed with 3 bodies (covering spokes 1-4, 5-10 and 11-14, masks created based on strongly low-passed filtered molmaps generated from an atomic model of the γ-TuSC unit) reaching resolutions between 7.1 and 8.4 Å. For visualisation, the density of the γ-TuSC within higher oligomers were sharpened (B-factor was automatically determined in RELION during post-processing) and filtered to local resolution. For the 14-spoked ring of γ-TuSC units, multi-body refined reconstructions were sharpened by post-processing, using the same masks as for multi-body refinement, and dust was hidden. In either case, masking during post-processing was used for resolution determination, but not applied during visualisation.

## Protein identification and model building

As an unbiased means of determining the identity of two coiled-coil α-helices outside of the Spc97 GRIP2 domain, ModelAngelo 0.2.4[69] was run on the post-processed reconstruction focused on a single γ-TuSC unit without providing prior information on protein sequences. Both α-helices were unambiguously identified as Spc72 by comparison of the amino acid sequences of *C. albicans* Spc72[1-599] (UniProt[70] ID Q5AGV5), Stu2 (A0A1D8PTZ8), Spc97 (Q59PZ2), Spc98 (A0A1D8PS42) and γ-tubulin (O93807) with the sequence predicted by ModelAngelo.

Initial atomic models for the γ-TuSCs within higher oligomers, flanked by one spoke of neighbouring γ-TuSCs on either side, were built using ModelAngelo, based on post-processed reconstruction, providing all abovementioned amino acid sequences as input. Missing segments were manually built in Coot[71], which was also used to trim away unjustifiably built segments and fuse fragmented chains. For some regions, the density quality only allowed Cβ-truncated residues to be built: Spc97 residues 699-714, 728-735; Spc98 640-647; γ-tubulin 431-435 (Spc97 spokes only). Model errors were corrected by iterative rounds of manual correction in Coot and real-space refinement in Phenix[72], with a restraint weight of 0.6. Backbone geometry errors and clashes were resolved using MDFF simulation in ISOLDE[73].

In order to build a model of a 14-spoked ring of γ-TuSC units, atomic models of the γ-TuSC were rigid-body docked into the reconstruction of the 14-spoked ring of γ-TuSCs using UCSF Chimera[74].

To extend the available atomic model for the interaction of the human CDK5RAP2 CM1 motif with GCP2 at spoke 13 and GCP6 at spoke 12 of the γ-TuRC, i.e., a combination of GCP2 and GCP6 from PDB-6V6S[6] and CDK5RAP2 from PDB-6X0V[21], an atomic model was built by ModelAngelo providing EMD-21985[21] and protein sequence information as input. From the model built by ModelAngelo, GCP2 residues 442-455, 611-620, 869-883 and 767-774 were added to the composite model of GCP2, GCP6 and CDK5RAP2 from PDB-6V6S and PDB-6X0V. The pseudoatomic model for the Mzt2/GCP2[N] module present in PDB-6X0V was replaced by the Mzt2/GCP2[N] module from an AlphaFold2 Multimer (version 2.3.1)[75,76] prediction of Mzt2/GCP2[N] module in complex with a dimer of CDK5RAP2 (residues 60-200), trimmed to

exclude parts not represented by the cryo-EM density. Models were corrected by iterative rounds of manual correction in Coot, MDFF simulation in ISOLDE and real-space refinement in Phenix, focusing on segments that were not present in the originally available models. For visualisation, the available reconstruction was segmented to retain only secondary structure elements of the CDK5RAP2 CM1 or its interaction elements on GCP2.

All map-to-model FSC curves were generated using the comprehensive validation function Phenix.

## AlphaFold predictions

Structures of dimeric Spc72$^{1-599}$ as well as dimeric Spc72$^{1-599}$ in complex with Stu2$^{882-924}$ were predicted using AlphaFold Multimer (version 2.3.1). For the top-ranked models representing the primary and secondary Stu2 interaction sites, Spc72 was trimmed to match the corresponding fragment (residues 291-428 or 429-599) and the models of interacting Stu2 and Spc72 interacting were relaxed as implemented in ColabFold[77]. Full-length dimeric Stu2 was predicted and relaxed using AlphaFold Multimer version 2.3.2.

## Hydrogen exchange mass spectrometry (HX-MS)

By detecting the exchange of backbone amide hydrogen atoms under different conditions, HX-MS reports on stabilisation of secondary structure elements such as α-helices and β-sheets, in which hydrogen bond formation prevents exchange of amide hydrogens in D$_2$O[78]. His-Spc72$^{291-599}$ was pre-incubated for 10 min at 20 °C in the absence or presence of FLAG-Stu2 (1:1 and 1:2; with 1 being 20 μM) and FLAG-Stu2 was pre-incubated for 10 min at 20 °C in the absence or presence of His-Spc72$^{291-599}$ (1:1 and 1:3; with 1 being 20 μM) in HX buffer (25 mM Tris-HCl, pH 8, 150 mM NaCl$_2$, 0.5 mM MgCl$_2$, 0.5 mM EGTA) and, subsequently diluted 10-fold in deuterated HX buffer to a final volume of 100 μl and incubated for 30 s at 30 °C. The deuteration reaction was quenched by adding 100 μl of ice-cold quench buffer (2 M NaCl, 50 mM Na-phosphate, pH 2.5, 4 M guanidinium-HCl, 200 mM TCEP), and the sample was immediately injected into the ice-cooled HPLC system.

In the ice-cooled HPLC setup, the protein was digested online using a column with immobilised pepsin and the peptides were desalted on a C8 trap column (POROS 10 R2, Applied Biosystems, #1-1118-02) for 2 min and eluted over an analytical C8 column (Waters GmbH, 186002876) using a 10 min gradient from 5 to 55% acetonitrile. All experiments were performed using a Maxis mass spectrometer (Bruker, Bremen, Germany) and analysed with the Data Analysis software. The calculated centroid values were corrected for the back-exchange using a 100% deuterated sample (for HX-MS details, see Supplementary Table 4). Statistical analysis of the data was performed by ordinary two-way ANOVA using Prism 9 version 9.5.1 (GraphPad Software Inc.). To aid comparison between the two coiled coil regions in His-Spc72$^{291-599}$, which have strongly different baseline hydrogen exchange patterns, we visualised protection against hydrogen exchange after ligand addition relative to the fraction that was not already protected in the baseline condition without ligand. That is, induction of full protection by the ligand would result in a value of 100%. Peptides with an initial exchange of <10% are not shown after this correction.

## Statistics and reproducibility

The number of biological replicates and other statistical parameters ($p$-values, sample sizes) are provided in the figure legends. Statistical analysis of yeast phenotypes was conducted with Prism (GraphPad Prism 10), and a two-way ANOVA test (two-sided) was used to distinguish the significance between sample conditions with a significance level of p ≤ 0.05. No statistical method for data exclusion or pre-determination of sample size was used; for the rationale, please read the reporting summary. For MST data, normalised fluorescence readings (thermophoresis plus T-jump) were plotted to generate binding curves by using the NanoTemper software, and then the curves were exported into R for visualisation. Protein band intensities on Western blots or Coomassie blue-stained gels were quantified using ImageJ. Quantification graphs for the protein band intensities on Coomassie blue-stained gels/Western blots and SEC chromatograms was performed using GraphPad Prism (GraphPad Prism v10.2.3).

## Reporting summary

Further information on research design is available in the Nature Portfolio Reporting Summary linked to this article.

## Data availability

Atomic coordinates and the associated cryo-EM densities generated in this study have been deposited in the Protein Data Bank and the Electron Microscopy Data Bank under accession codes PDB-9H9Q [https://www.rcsb.org/structure/9H9Q] / EMD-51971 (γ-TuSC in ring-like oligomer), PDB-9H9R [https://www.rcsb.org/structure/9H9R] / EMD-51972 (14-spoked γ-TuRC) and PDB-9H9P [https://www.rcsb.org/structure/9H9P] (human CDK5RAP2$^{CM1}$ in complex with GCP2/GCP6/Mzt2). The mass spectrometry proteomics data generated in this study have been deposited to the ProteomeXchange Consortium via the PRIDE partner repository with the dataset identifier PXD057184. Models predicted by AlphaFold-Multimer generated in this study have been deposited to the ModelArchive (https://modelarchive.org/) under accession codes ma-eefkg [https://modelarchive.org/doi/10.5452/ma-eefkg], ma-bktlp [https://modelarchive.org/doi/10.5452/ma-bktlp], ma-kzdh2 [https://modelarchive.org/doi/10.5452/ma-kzdh2] and ma-3kpdx [https://modelarchive.org/doi/10.5452/ma-3kpdx]. Cryo-EM data collection, refinement, and validation statistics data, experimental details for HX-MS experiments as well as primer, plasmid, and yeast strains are provided in the Supplementary Information file. Constructs generated in this study are available upon request to the corresponding authors. Published structural data used in this article: PDB-5W3F [https://www.rcsb.org/structure/5W3F], PDB-6V6S [https://www.rcsb.org/structure/6V6S], PDB-6X0V [https://www.rcsb.org/structure/6X0V], PDB-7ANZ [https://www.rcsb.org/structure/7ANZ], PDB-7M2Y [https://www.rcsb.org/structure/7M2Y], PDB-7M2Z [https://www.rcsb.org/structure/7M2Z], PDB-8QV2 [https://www.rcsb.org/structure/8QV2], PDB-8QV3 [https://www.rcsb.org/structure/8QV3], EMD-8758, EMD-21985. Source data are provided with this paper.

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

## Acknowledgements

We thank Erik Župa (Princeton University, USA) for insightful discussions, Ursula Jäkle (ZMBH, Heidelberg University, Germany) for help with insect cell expression and Ieva Baronaitė for support with data processing (ZMBH, Heidelberg University, Germany). We thank Dr. Michael Knop (ZMBH, Heidelberg University, Germany) and Dr. Diana Rüthnick (ZMBH, Heidelberg University, Germany) for providing plasmids and a yeast strain. We thank Rainer Beck (BZH, Heidelberg University, Germany) for providing the introduction to the NanoTemper Monolith NT.115 machine located at BZH. We further thank Dr. Karine Lapouge from the European Molecular Biology Laboratory (EMBL Heidelberg, Germany) Protein Expression and Purification Core Facility (PEPCF) for Mass Photometry and SEC-MALS experiments. We acknowledge the services SDS@hd and bwHPC supported by the Ministry of Science, Research and the Arts Baden-Württemberg, as well as the German Research Foundation (INST 35/1314-1 FUGG and INST 35/1597-1 FUGG). We would like to acknowledge access to the infrastructure and support provided by the Cryo-EM Network at the Heidelberg University (HDcryoNet), which is funded and supported by the German Research Foundation (DFG), the Federal Ministry of Education and Research (BMBF) and the Ministry of Science Baden-Württemberg, among others, within the framework of the Excellence Strategy of the Federal and State Governments of Germany. We acknowledge the technical support of the Core Facility for Mass Spectrometry and Proteomics of the Center for Molecular Biology of Heidelberg University (ZMBH). We thank Marcin Luzarowski for support with data deposition. The Core Facility for Mass Spectrometry and Proteomics is funded by the ZMBH and partially funded by the CellNetworks Core Technology Platform (CCTP) of Heidelberg University. The CCTP is funded in part by the Federal Ministry of Education and Research (BMBF) and the Ministry of Science Baden Württemberg within the framework of the Excellence Strategy of the Federal and State Governments of Germany. This work is supported by grants of the Deutsche Forschungsgemeinschaft (DFG) to E.S. (DFG Schi 295/4-4) and to S.P. (DFG PF 963/1-4; DFG PF 963/4-1). S.P. also acknowledges funding by the Aventis Foundation and the Chica and Heinz Schaller Foundation.

## Author contributions

A.Z. performed yeast experiments, and A.Z. and M.W. performed cloning, protein expression, protein purification, and biochemical analyses. B.J.A.V. performed cryo-EM data collection, cryo-EM data analysis, model building, structural interpretation and computational predictions. A.N. performed negative staining and data acquisition. A.Z., A.N., B.J.A.V., and M.W. analysed negative stain EM data. N.L. acquired HX-MS data, which was analyzed by N.L., M.P.M., and B.J.A.V. E.S. and S.P. supervised the experiments. B.J.A.V., E.S., S.P., A.Z., and M.W. wrote the manuscript. All authors discussed the data and gave final approval for publication.

## Funding

## Competing interests

The authors declare no competing interests.
