## [Transparent Peer Review file · Nature Communications]

Structural insights into the interplay between microtubule polymerases, γ -tubulin complexes and their receptors

Corresponding Author: Dr Stefan Pfeffer

Version 0:

Reviewer comments:

Reviewer #1

(Remarks to the Author)

In this manuscript, the authors present a structure of recombinantly expressed *C. Albicans* γ -TuRC co-expressed with Spc72 and Stu2. Notably, the determined γ -TuRC structure adopts an active conformation compatible with microtubule nucleation, suggesting that the SPC72-bound γ -TuRC is constitutively active, in contrast to previously determined structures of the *S. Cerevisiae* γ -TuRC bound to Spc110, in which only a small number of γ -TuSCs sample the closed “active” conformation unless it is stabilized via crosslinks. The authors suggest this results from the Spc72 CM1 helix forming a dimer whereas the Spc110 CM1 helix is monomeric, or that it may point to differences in the regulation of microtubule nucleation at the nuclear and cytoplasmic faces of the spindle pole body. The authors extend their structural work with a mutational analysis of Spc72 to characterize various interactions observed in the presented structure which may contribute to oligomerization and activation of the γ -TuRC. Finally, the authors use AlphaFold2-multimer structural predictions to guide the design of immunoprecipitation experiments defining two binding sites of the Stu2 C-terminal helix to Spc72.

Taken together, these results are deserving of publication. However, I believe the manuscript should be significantly edited to address major concerns respecting the interpretation of the author’s results.

Major comments:

1) The authors refer to their structure as a “cryo-EM reconstruction of the *C. Albicans* γ -TuSC oligomer in complex with Spc72 and Stu2”. While the authors show that Stu2 copurifies with their assembled complex, the 3D model shows no density for Stu2. The authors suggest that this is due to flexible linkers in the between CM1 and the Spc72 coiled-coil domains that interact with Stu2. However, it is also possible that Stu2 dissociates from the complex during or just prior to vitrification. It may be possible to use labeling methods or additional processing to establish the stoichiometry of Stu2 to the presented structures. In the absence of such data, it should be made clearer that the presented structures show a complex of γ -TuSC oligomer in complex with Spc72, but that Stu2 may not be bound.

The section entitled “Stu2 is tethered to the TuSC/Spc72 complex by flexibly linked coiled-coil modules” should be toned down to reflect that it is unknown if Stu2 is bound in the presented data, or at what stoichiometry it is bound in the cryo-EM data.

2) Similarly to point 1 above, it is unclear if Mzt1 remains bound to the assembled γ -TuSC complexes presented in Supplementary Figure 2. The presented data does not strongly establish that “Mzt1 had no impact on oligomer formation or the overall architecture of γ -TuSC rings”. It is likely to be difficult to directly show the presence of Mzt1 to the γ -TuSC rings, given its small size and the low resolution of the structure. It is possible that the GFP tag on Mzt1 may be visible even at low resolution, or that the GFP could be used as a tag for labeling studies establishing that Mzt1 remains bound to the γ -TuSC rings shown. I would suggest that it should be clearer that the 3D models may not represent a Mzt1-bound state and that the data does not support drawing strong conclusions.

3) There is significant work from the Agard lab (Kollman et al. Nature 2010, NSMB 2015, Brilot et al. eLife 2021) and more recently by Dendooven and colleagues (NSMB 2024) that characterizes the conformational changes of the γ -TuSC during activation. The discussion of the conformational changes in the TuSC upon Spc72 binding could be significantly improved if it included a comparison examining to what extent there are conserved similarities or significant differences between the conformational changes in the structures presented in this manuscript and previously determined structures γ -TuSC structures.

4) The authors suggest that CM1 dimerization performs an important and conserved function in γ -TuRC assembly and activation. However, significant evidence indicates this may not be accurate: 1) previous work by Brilot et al. (eLife 2021), and more recently by Dendooven et al. (NSMB 2024) shows that a γ -TuSC rings bound to a monomeric CM1 sample the active conformation and that a monomeric CM1 is present at the inter-TuSC when microtubules are nucleated at the nuclear

face of the spindle pole body. 2) All of the major contacts between Spc97, Spc98 and Spc72 highlighted in Figure 3 and Supplementary Figure 5 appear to be between Spc97 or Spc98 and the CM1in of Spc72, further suggesting the CM1out helix may in fact be dispensable. The evidence presented (3R mutant, pulldowns and in vivo characterization of mutations) does not clearly establish whether the triple mutant disrupts γ -TuSC oligomerization due to a disruption of CM1 dimerization or through some other mechanism, which could include disrupting one of the binding interfaces on Spc97 or Spc98. The authors suggest that coiled coils C-terminal to the CM1 in Spc110 may help it function in the absence of CM1 dimerization, but the authors model similarly dimeric C-terminal coiled coils in Figure 6c,d. This logic should then also apply to the 3R mutant constructs presented in this work, and the 3R mutants should also stimulate γ -TuSC assembly if the only effect of the mutations is to render the CM1 helix monomeric.

Supplementary Figure 7 should include a comparison, perhaps including an overlay, with the Spc110 CM1-bound closed γ -TuSC structure as it may be helpful in examining whether any substantive differences in the structures of γ -TuSCs bound to monomeric and dimeric CM1 helices suggest a role for CM1 dimerization.

The SEC trace in Figure 4f for the 3R mutant does not appear to show pronounced peaks for either assembled γ -TuSC oligomers or monomers. This suggests the 3R mutations may not simply inhibit γ -TuSC oligomerization.

To better address the role of CM1 dimerization in γ -TuSC oligomerization, the authors could generate Spc72 heterodimers using SpyCatcher-SpyTag where only one wildtype Spc72 CM1 helix is present.

The evidence presented here also does not suggest a defined mechanism for the importance of dimerization in γ -TuSC assembly. The authors should more directly establish the importance of dimerization before stating that it is essential to γ -TuRC assembly and activation.

5) Supplementary Figure 3 should include the relevant map-to-model FSC curves.

6) Supplementary Figure 4 indicates that multi-body refinement was performed on the TuSC rings. Is this data discussed or presented elsewhere in the manuscript? If yes, FSCs should also be included for this data.

7) The authors should explain in the methods how masks used in refinement, focused classification, multi-body refinement, and for FSC estimation were generated and used. The method used to calculate the map-to-model FSCs should also be described. Describe which regions only allowed C-beta truncated residues to be built.

8) The appearance of the data in Figures, 1, 3 and supplementary figure 5 all appears consistent with the resolution claimed and the FSC curves. At this resolution (3.6Å), it can be difficult, especially in lower quality regions of maps, to clearly define side-chain conformations and interactions. The authors should include more data clearly showing the fit of the model to the experimental map, focusing on the CM1 helix region presented in Supplementary Figure 5 and the various detailed interactions presented there.

9) The authors use structural prediction tools to guide the identification of two binding sites of the Stu2 C-terminal helix to Spc72. Using immunoprecipitation experiments, they show that Spc72 residues 300-350 and 430-480 form two binding sites for the Stu2 C-terminal helix. I do not agree with the authors' claim that "by combining systematic co-IP experiments and AlphaFold2-based structure predictions, we could characterize the interaction between Stu2 and Spc72 at the residue level". The authors should tone down these claims or perform experiments such as extensive site-directed mutagenesis to probe or validate this interaction at the residue level.

10) It is unclear if one or both Stu2 binding sites on Spc72 are required, or if the higher-affinity binding site located between residues 430-480 functions as the main binding site in vivo.

11) Could the authors comment on why the newly modelled regions in EMD-21985 were not previously modelled, or why they could now be modelled? I am concerned that this region may have poor density that led to this region not being modelled previously.

Minor comments

1) In Supplementary Table 1, the estimated B-factor for the masked single γ -TuSC map suggests a higher resolution should have been achieved given the amount of data included in the averaged model. Danev et al. (Nature Commun. 2021) presents a number of structures with higher B-factors that require significantly less data than presented here to achieve 3A resolution. Is it possible the B-factor estimation is inaccurate?

2) The authors should clarify, either in figure legends or methods, how the EM maps were sharpened, filtered and masked in the various panels where this data is presented.

3) The data presented in Fig 3e and supplementary figure 7 are similar in design and conclusions to the work performed by Lyon et al (Mol. Biol. Cell. 2016) on Spc110. This work should probably be cited as corroborating evidence of the central role of CM1 in γ -TuSC assembly.

4) On line 644 "As an independent means of determining..." should be changed to "As an unbiased means of determining..." to match the main text.

5) Update Supplementary Table 1 with EMDB and PDB IDs.

Reviewer #2

(Remarks to the Author)

This manuscript reports the cryo-EM structure of the cytoplasmic γ -TuRC from *C. albicans*, representative of ascomycete yeast. The reported structure was reconstituted from co-expression of Spc97, Spc98, γ -tubulin, Spc72 and Stu2. The authors determined a high resolution (3 Å) of the γ -TuSC protomers. They used the resultant model to build the entire γ -TuRC complex at 8 Å resolution. The authors observe the Spc97, Spc98 and γ -tubulin subunits of γ -TuSC protomers, and the CM1 motif of Spc72. Although present in the sample used for cryo-EM analysis, cryo-EM density was not visible for Stu2. Based on AlphaFold2 predictions, and biochemistry, the authors propose a mechanism of Stu2 interaction with Spc72, suggesting that multiple copies of Stu2 might bind to an Spc72 dimer.

The authors test (and validate) the mode of binding of Spc72 and suggest it plays a role in assembly of γ -TuRC. Co-expression of Spc97, Spc98, γ -tubulin with Mtz1 indicated that this subunit associates with γ -TuRC.

This is an interesting paper describing significant and important results. In general the experiments are performed vigorously and clearly explained. The figures are of high quality.

The main findings include:

1. The cytoplasmic γ -TuRC with Spc72 adopts a closed helical conformation that matches the 13-protofilament geometry of the minus end of a microtubule filament in the absence of binding the minus end of a microtubule filament. This is in contrast to the structure of the in vitro reconstituted *S. cerevisiae* γ -TuRC with Spc110 that adopts an open conformation in the absence of microtubules.
2. Spc72 bridges neighbouring γ -TuSC protomers, explaining how it can promote the closed conformation of γ -TuRC.
3. Spc72 binds to γ -TuRC through CM1 motifs as a helical dimer in contrast to how Spc110 binds γ -TuRC through a single α -helix.
4. The model for how Stu2 interacts with Spc72 is of interest – although note caveats below.

Subject to revisions, this manuscript warrants publication in Nature Communications.

Major points.

1. The major concern is the description of how Stu2 binds to Spc72. The authors propose that the absence of cryo-EM density for Stu2 is due to Stu2 binding to a disordered region of γ -TuRC. This is quite likely, but it is also possible that Stu2 dissociated from γ -TuRC during cryo-EM grid sample preparation. The authors should include this possibility in the text.
2. The authors present AlphaFold models of Spc72 and complexes of the C-terminal α -helix of Stu2 bound to Spc72. Unfortunately it is not possible for the reader to assess the quality of these models because the authors did not include either the coordinates colour-coded by pLDDT score or the PAE plots. Both should be included in the paper.
3. Although the authors test the prediction of Stu2 binding to the two modules of Spc72, quite an imprecise mapping is used, ie. deletions of Spc72. Since the authors do not include NMR or CLMS data to test/validate their model, a more precise mutagenesis study is called for. The authors could mutate interacting residues in Spc72 and Stu2 to better validate their model.
4. The model that Stu2 binds to module 1 and module 2 potentially simultaneously could be tested by determining the mass of the Stu2-Spc72 complex using SEC-MALS or mass photometry.
5. Fig. 1a. Stoichiometry of subunits. The authors state that all subunits are present in an approximately stoichiometric ratio. 'Ratio' is a bit ambiguous. Does it mean equal ratio? An equal stoichiometric ratio isn't obvious from the SDS PAGE gel. Stu2 is more abundant. A more convincing gel is that shown in after SEC (Supplementary Fig. 1d). As Spc72 is a dimer bound to an Spc97-Spc98 heterodimer, it isn't formally correct to state that all subunits have equal stoichiometry.
6. Supplementary Fig. 10a is difficult to follow. The locations of CM1 and the hinge between the coiled-coil modules 1 and 2 are not clear. Possibly colour N- to C-termini with a blue-to-red colour ramp, and label more clearly using arrows.

Minor points.

1. The authors state that *S. cerevisiae* nuclear γ -TuRC adopts an open conformation, referencing 13 and 14 from 2010, 2015. The Brilot paper (ref 10) describes reconstituted γ -TuRC. More recent higher resolution cryo-ET reconstructions of the nuclear γ -TuRC from enriched spindles showed that all γ -TuRC are in a closed conformation capping the minus end of microtubules. This section of the Introduction does not reflect the current knowledge of the field, and incorrectly conflates the structures of in vitro reconstituted γ -TuRC with Spc110 with the actual structure of the nuclear γ -TuRC in situ.
2. Show colour guide of all subunits in Fig. 2 (or otherwise label subunits).
3. Table 1: *S. cerevisiae* not *S. pombe*.
4. Please show side chains in Supplementary Fig. 7b.
5. There are two lanes 1-5 in Fig. 6a, b. This is a little confusing. Similar for Supplementary Figs. 1b, 9b, c.
6. Reference 47 should be updated. PMID: 38609662.
7. Reference 60 should be updated. PMID: 38408488.

Reviewer #3

(Remarks to the Author)

Microtubule (MT) is a highly conserved cytoskeleton that plays fundamental roles in cellular activities. Obtaining a mechanistic explanation of the MT nucleation process in a wide range of organisms helps understand the essential principles of MT regulation.

In this work, the authors revealed how budding yeast Spc72, a yeast homologue of CDK5RAP2, contributes to the oligomerisation of the γ -TuSC through the CM1 motif by solving the reconstituted assembly of the γ -TuSC using cryo-EM. The authors also tried to unveil the structural arrangement of Stu2, a yeast homologue of XMAP215/chTOG. They mapped the domains responsible for the Spc72-Stu2 interaction, but the structural information could not be obtained, presumably due to the flexibility of the complex. The overall data quality is excellent and compelling, although some biochemical data can be further improved. Hence, I believe the work is suitable to be published in Nature Communications after addressing the issues listed below.

Major issues

1. Molar stoichiometry of Spc72 in the γ -TuSC assembly

The manuscript proposes that a dimer of Spc72 interacts with Spc97/98 of the γ -TuSC assembly (hence a total of 12-14 Spc72 in one γ -TuRC?). However, direct evidence is not provided; as far as I am aware, the sole relevant information would be the Coomassie-stained gel image (Fig. 1a), which is described as "all γ -TuSC components and Spc72 in an approximately stoichiometric ratio (page 4)", which can be misleading if two molecules of Spc72 bind to one molecule of Spc97 (and the neighbouring Spc98). Therefore, I would like the authors to examine the molecular weight (MW) of the complex containing Spc72. It may be achieved by techniques such as multi-angle light scattering (MALS), small-angle X-ray scattering (SAXS) or mass photometry (MP). For this experiment, I would suggest the authors exclude Stu2 as Stu2 inclusion may make the analysis over-complicated. Instead, I would like the authors to examine one of the following two samples. First option: MW of the SEC-purified γ -TuC assembly consisting of the γ -TuSC (with FLAG-Spc98, Spc97 and γ -Tubulin) and Spc72(1-599). Second option: the MW of the SEC-purified γ -TuSC decorated with Spc72.PA(1-599). The latter sample may be more homogenous, and the data analysis may be more straightforward.

2. The minimum Spc72 motif sufficient for the γ -TuSC oligomerisation

The authors demonstrate that the Spc72 CM1 motif interacts with Spc97, and T234/T236 at the CM1 N-term is in contact with the neighbouring Spc98 by analysing the cryo-EM structure. They also show that the Spc72.PA mutant cannot help the oligomerisation of γ -TuSC. However, whether the Spc72 CM1 motif (including T234/T236) is sufficient for the γ -TuSC oligomer is not shown. Therefore, I would like the authors to examine whether adding the Spc72 CM1 motif (Spc72.231-268) to the γ -TuSC causes the γ -TuSC oligomerisation. In this experiment, Stu2 can be excluded.

3. The Spc72.3R phenotype

3-(1)

The authors claim that the Spc72.3R mutant still can bind to the γ -TuSC based on the pull-down experiment (Fig.4e). However, the pull-down experiment is not quantitative, and the band intensity of Spc72.3R in the FLAG-IP panel (Fig. 4e) seems to be reduced. Furthermore, in Fig. 4(g), fraction 11.75 ml hardly contains the component of the γ -TuSC, unlike the case shown in Fig. 3(e) where signals of γ -TuSC components and Spc72.PA are evident in fraction 11.75 ml. Instead, Fig. 4(g) shows that fractions 12.5 ml and 13.25 ml contain the stronger γ -Tubulin signals as if the γ -TuSC is not associated with Spc72. In fact, the signal of His-Spc72 seems much weaker than the signals of Spc98 and Spc97 in these fractions in Fig. 4(g).

Therefore, I would like the authors to conduct a quantitative method (such as ITC, SPR, biolayer interferometry (BLI) or microscale thermophoresis as the authors used in their previous studies) to examine whether the 3R mutation affects the interaction between the γ -TuSC and Spc72. Stu2 can be excluded from this experiment.

As I explain below in (2), there might be a possibility that Spc72.3R may interfere with the formation of the γ -TuSC.

Therefore, if the γ -TuSC - Spc72 interaction kinetics shows complex outcomes using ITC/SPR/BLI, it may be a good idea to simply look at the KD of Spc97 and Spc72.WT/Spc72.3R (without including Spc98 and γ -Tubulin).

3-(2)

In Fig. 4(f), an intriguing elution profile is presented for the γ -TuSC/Spc72.3R/FLAG-Stu2 sample. There is hardly any detectable peak that corresponds to the monomeric γ -TuSC. This is in a striking contrast against the result presented in Fig. 3(d) where Spc72.PA interacts with the γ -TuSC. Furthermore, the *in vivo* phenotypes between them are different. It could be that the expression of Spc72.3R could negatively affect the γ -TuSC formation.

The authors may argue that Fig. 4(g) shows the successful formation of γ -TuSC. However, the γ -TuSC formation efficacy seems to be greatly reduced because fractions 14.0 – 17.0 ml show substantial signals of Spc98, Spc97 and γ -tubulin, which would represent the various associating status and monomeric status of these molecules. This trend is distinct from the result presented in Fig. 3(e), where most of the signals of Spc98, Spc97 and γ -tubulin are found in fractions 11 – 12.5 ml. Therefore, I would like the authors to conduct the following experiments.

(a) Examine whether Spc72.3R is a dominant or recessive mutation. The spc72. Δ P55-N62 mutant may be used as a reference, as it represents an example that does not disturb the γ -TuSC formation.

(b) In the images presented in Supplementary Fig. 6(c) and Supplementary Fig. 8, the expression of Spc72.3R seems to cause defects in the spindle formation. I would like the authors to quantify the spindle formation defect phenotype of the cells expressing Spc72.3R and Spc72. Δ P55-N62.

(c) A possible cause of the Spc72.3R *in vivo* phenotype could be its capability to replace Spc110 at the SPB. I would like the authors to examine the localisation of Spc110 in the presence of Spc72.3R.

4. The structural arrangement of Stu2

I understand that the authors aimed to solve the structure of the oligomerised γ -TuSC decorated with both Stu2 and Spc72 (1-599). However, the cryo-EM structure did not show an obvious contribution by Stu2. This is interesting as previous studies (including the ones from the authors' group) showed that Stu2/XMAP215 directly binds γ -TuSC. As the authors discussed, Stu2 flexibility can be the reason, and the limitation in the cryo EM-based approach may be demonstrated in this instance. Meanwhile, I would like the authors to probe the molar stoichiometry of Spc71(1-599) and Stu2 complex as the authors identified two Stu2 binding sites on Spc72 (1-599). It can ideally be conducted using the Stu2: γ -TuSC:Spc72(1-599) complex, and the molar stoichiometry can be examined both by Coomassie staining of the gels and MW analysis by SAXS/MALS/MP. In this case, it may be better to use sucrose gradient ultracentrifugation to purify the complex as fraction 8.5 ml of the SEC seems to be very close to the void volume, causing the sample heterogeneity, which is shown in Supplementary Figure 1 (d). Alternatively, a complex of Stu2 and Spc72(1-599) can be purified and analysed without the presence of the γ -TuSC.

Minor issues

- In order to avoid confusion, I would like to ask the authors to keep stating "Spc72(1-599)" throughout the manuscript, including all the figures. Also, I would like to ask the authors to clearly indicate the presence of a His-tag, if that is the case,

in all the figures.

- It is unclear whether Stu2 was included in the result shown in Fig. 4(e). It would be helpful if the authors clarify the experimental set-up of Fig. 4(e).
- The authors concluded that GFP-Mzt1 has no impact on oligomer formation and the overall structure of γ -TuSC oligomers. However, Supplementary Figure 2 (d) shows that the sample is highly heterogeneous, and it is unclear whether the authors could pick the oligomers involving GFP-Mzt1. Furthermore, it is unclear whether the authors could identify the GFP-Mzt1 density in the structure presented in Supplementary Figure 2(e). Therefore, the statement that “Mzt1 has no impact” can be misleading unless the authors show the GFP-Mzt1 density in the cryo-EM structure presented in Supplementary Figure 2(e).
- There are some typos – for example, figure 4(g) legend “...shown in panel (e) by western blotting” -> maybe “...shown in panel (f) by Western blotting”?

Version 1:

Reviewer comments:

Reviewer #1

(Remarks to the Author)

The revised manuscript provided by the authors substantively addresses all the points previously raised in the prior review. The authors have made significant revisions to the text throughout, added panels to compare their structural results to prior work in the field, included data to clarify the oligomeric state of Spc72 in the 3R mutant and further probe its binding to γ -TuSC, re-analyzed their SEC data, and significantly improved their analysis of the Spc72/Stu2 binding sites, and provided *in vivo* data establishing that the secondary binding site may be dispensable. The authors have further included important data used in validating their structures, including several panels showing the quality of the structures, and made their maps and models available to the reviewers. Overall, the manuscript is significantly improved, with many of the major conclusions better supported by the data.

While the new data significantly improve the manuscript, there are a few significant questions the new data raises that require additional revisions:

1. The additional experiments performed by the authors establishes that the 3R mutant exhibits reduced dimerization, reduces the affinity of the CM1 interaction with γ -TuSC, and still binds to γ -TuSC. However, there are still significant questions as to whether the oligomeric state of CM1 in the Spc721-599.3R constructs and whether the 3R mutant's effect is due to its disruption of dimerization.
 - 1.1 While the SEC and mass photometry data provided to establish the assembly state of the CM1 3R mutant supports the authors' claim that the 3R mutant disrupts dimerization, the presence of a tetramer peak in the GST-CM1 3R mutant mass photometry data (Supp. Fig. 9g) suggests that the CM1 helices in the 3R mutant may still dimerize even at relatively low concentrations. It seems likely the CM1 helices in the 3R mutants might still be dimerized at the high local concentration expected in the constructs used in this study including the C-terminal coiled-coils in Spc72. Unfortunately, the authors also show in Figure 3R that the CM1 dimer alone cannot induce γ -TuSC assembly, making it difficult to directly probe the role of CM1 dimerization using the MBP constructs generated. As there is some remaining ambiguity about whether CM1 is dimerized in the Spc721-599.3R constructs, I might suggest a more direct method such as using EPR or attaching FRET probes to the CM1 helices to more directly probe the structural state of the CM1 helices in the Spc721-599 WT and 3R constructs. A related experiment that may provide useful data is to examine the affinity of the MBP-tagged CM1 constructs for the γ -TuSC.
 - 1.2 As the Spc72.1-599.3R mutants' CM1 helix may still be dimerized, it remains unclear whether the 3R mutant's inability to stimulate γ -TuSC assembly is due to a disruption of CM1 dimerization, diminished affinity for γ -TuSC, or some other effect. The authors could target the CM1 out interacting residues for mutagenesis to further probe whether the main contribution of CM1 dimerization in this system is to increase the affinity for the complex with these additional contacts, but this would not address the proposed role of structural rigidity induced by CM1 dimerization in the “wedge” model cited by the authors.
 - 1.3 In Figure 4e, the WT CM1 binding curve appears very similar to the 3R CM1 binding curve, with both appearing to show a midpoint in the rise of F_{norm} slightly above 1000 nM. The error bars in the WT CM1 appear to be similar in size during its rise to the observed dynamic range of the curve, suggesting there may be significant uncertainty in the concentration measurement that is not reflected in the affinity measurement provided at the top of the panel. Additional information may be helpful in establishing the accuracy of the binding curves. As the GST-CM1 3R construct still forms dimers, is it possible the observed affinity in these plots is still due to CM1 dimers binding to γ -TuSC?
 - 1.4 In the western blot shown in figure 4h, γ -TuSC appears to co-elute with Stu2 and Spc72.1-599.3R. The trace shows no peak consistent with the peak at 12 ml that the authors attribute to γ -TuSC/Spc72.1-599/Stu2. This suggests that the 3R mutant's effect may not be well characterized, or that the peaks may be misattributed in the revised manuscript. Overall, if the authors are unable to better establish the monomer state of the CM1 helix in their Spc72.1-599.3R constructs, I would suggest that the authors reduce their claims on the importance of dimerization.

2. The additional experiments performed by the authors provide important additional validation of the Stu2/Spc72 interfaces. However, the interfaces may not be mapped precisely enough to be described as validated at the residue level:

- 2.1 In figure S10d, the authors show that the Stu2 LIM mutant and the SPC72 EDID mutants both have strong effects on assembly with the combination appearing to completely abolish it. However, the Spc72 ELLY mutant displays a much weaker effect on assembly, indicating the structural prediction for this interaction site may not be accurate.
- 2.2 While both the Stu2 LIM mutant and the SPC72 EDID mutants have strong effects on assembly, the additive effect observed when both sets of mutations are present does not establish that these residues interact with each other. This leaves the register of the interaction in question. This also applies to the ELLY mutants.

2.3 The HDX data appears quite noisy, and only shows a strong effect for a small portion of the N-terminal weaker binding site.

I would suggest the authors tone down the language on the interface being mapped at the residue level. I would also suggest that if the authors would like to include a discussion of the electrostatic interactions in the interfaces, it be made clear that this remains speculative due to the relatively low “resolution” of the data provided.

3. The panels showing density and the fitted models in Figures 6 and 9 all appear to be relatively tightly masked via segmentation. While the quality of the structures presented in this manuscript is visually consistent with the quoted resolutions, it may be easier for the reader to assess their quality if the structures are less tightly masked, so the noise level present can be readily assessed without examining the deposited data. I would suggest some of the panels in these two figures be unmasked and unsegmented, with only clip planes used to focus on the region of interest.

Minor points:

In the overlay in Figure 4c, the authors note that the structures were superposed based on an alignment of Spc97 with GCP2.473-895. I wonder if this may cause the alignment to report more on differences in the conformations of Spc97/GCP2 than on differences in the CM1 motifs in the structures. Perhaps an alignment based on the CM1 motifs and the surrounding conserved interaction regions may yield some additional insights?

Similarly to the point above, would the panel Supp. Fig 6e report more directly on differences or similarities in the CM1 motifs in *S. cerevisiae* Spc110 and *C. albicans* Spc72 if the alignment were focused on the CM1 helix and surrounding interaction regions? Such an alignment may also yield additional insights into differences in regulation between the systems.

During the previous round of review (point 4.4), I raised a question on the authors' claim on the importance of the C-terminal coiled coils in *S. Cerevisiae* Spc110 in compensating for the monomeric state of its CM1 helix. After reading their rebuttal, I now agree with their assessment that the C-terminal coiled coils present in Spc110, as well as the loop between the CM1 motif and these coiled-coils may help to increase the affinity of the Spc110 for the γ -TuSC due to the additional contacts formed and thereby compensate for the loss of the interactions between CM1out and the γ -TuSC observed in the Spc72-bound γ -TuSC structure the authors present in this manuscript. I would support adding the text back to the manuscript.

At low thresholds, the raw unsharpened data for the γ -TuRC ring shows low resolution density that resembles an eighth γ -TuSC subunit. Could the authors comment on whether this is due to some rotational misalignment, or a subpopulation present in their purification?

In the sentence “CM1-mediated dimerisation of dimers induced tetramer formation predominantly in case of MBP-CM1231-268”, I believe the authors intend to refer to the GST-CM1 construct.

Reviewer #2

(Remarks to the Author)

The authors have conducted extensive experiments and revisions that satisfy my questions and concerns. The manuscript warrants publication in Nature Communications.

Reviewer #3

(Remarks to the Author)

The authors have addressed my concerns, and the manuscript warrants publication in Nature Communications.

Point-by-Point Reply to Reviewers

We thank the reviewers for their thoughtful and supportive comments. Below we comprehensively address all the specific points raised by the reviewers and elaborate on the corresponding changes in the manuscript.

Importantly, we have included an extensive set of new experiments, which overall significantly strengthen our manuscript, including experiments that now establish a clear mechanistic role for CM1 motif dimerization in γ -TuSC oligomerization and validate the binding interfaces between Stu2 and Spc72.

Reviewer #1 (Remarks to the Author):

In this manuscript, the authors present a structure of recombinantly expressed *C. Albicans* γ -TuRC co-expressed with Spc72 and Stu2. Notably, the determined γ -TuRC structure adopts an active conformation compatible with microtubule nucleation, suggesting that the SPC72-bound γ -TuRC is constitutively active, in contrast to previously determined structures of the *S. Cerevisiae* γ -TuRC bound to Spc110, in which only a small number of γ -TuSCs sample the closed “active” conformation unless it is stabilized via crosslinks. The authors suggest this results from the Spc72 CM1 helix forming a dimer whereas the Spc110 CM1 helix is monomeric, or that it may point to differences in the regulation of microtubule nucleation at the nuclear and cytoplasmic faces of the spindle pole body. The authors extend their structural work with a mutational analysis of Spc72 to characterize various interactions observed in the presented structure which may contribute to oligomerization and activation of the γ -TuRC. Finally, the authors use Alphafold2-multimer structural predictions to guide the design of immunoprecipitation experiments defining two binding sites of the Stu2 C-terminal helix to Spc72.

Taken together, these results are deserving of publication. However, I believe the manuscript should be significantly edited to address major concerns respecting the interpretation of the author’s results.

We thank Reviewer #1 for the positive evaluation of the manuscript.

Major comments:

1). The authors refer to their structure as a “cryo-EM reconstruction of the *C. Albicans* γ -TuSC oligomer in complex with Spc72 and Stu2”. While the authors show that Stu2 copurifies with their assembled complex, the 3D model shows no density for Stu2. The authors suggest that this is due to flexible linkers in the between CM1 and the Spc72 coiled-coil domains that interact with Stu2. However, it is also possible that Stu2 dissociates from the complex during or just prior to vitrification. It may be possible to use labeling methods or additional processing to establish the stoichiometry of Stu2 to the presented structures. In the absence of such data, it should be made clearer that the presented structures show a complex of γ -TuSC oligomer in complex with Spc72, but that Stu2 may not be bound.

The section entitled “Stu2 is tethered to the TuSC/Spc72 complex by flexibly linked coiled-coil modules” should be toned down to reflect that it is unknown if Stu2 is bound in the presented data, or at what stoichiometry it is bound in the cryo-EM data.

Based on the identified interactions between Stu2 and the γ -TuSC-Spc72 complex, which all involve flexible segments, we indeed would not expect to observe density for Stu2. However, we agree with Reviewer #1 that dissociation of Stu2 prior to or during vitrification cannot be entirely ruled out, although the complex was purified via Stu2. We have updated all references to our cryo-EM reconstruction, which now no longer explicitly mention the presence of Stu2 in the complex, and have discussed the possibility of Stu2 dissociation during sample preparation in the text:

“, although it cannot be entirely excluded that Stu2 has dissociated from the γ -TuSC/Spc72¹⁻⁵⁹⁹ complex during cryo-EM sample preparation.”

And in the discussion:

“Although Stu2 dissociation or destabilisation during cryo-EM sample preparation cannot be excluded entirely, ...”

2) Similarly to point 1 above, it is unclear if Mzt1 remains bound to the assembled γ -TuSC complexes presented in Supplementary Figure 2. The presented data does not strongly establish that “Mzt1 had no impact on oligomer formation or the overall architecture of γ -TuSC rings”. It is likely to be difficult to directly show the presence of Mzt1 to the γ -TuSC rings, given its small size and the low resolution of the structure. It is possible that the GFP tag on Mzt1 may be visible even at low resolution, or that the GFP could be used as a tag for labeling studies establishing that Mzt1 remains bound to the γ -TuSC rings shown. I would suggest that it should be clearer that the 3D models may not represent a Mzt1-bound state and that the data does not support drawing strong conclusions.

While Mzt1 was clearly present in the γ -TuSC-Spc72¹⁻⁵⁹⁹ complex purified through FLAG-Spc97, similarly to comment 1 above, we cannot exclude that Mzt1 dissociated from the complex during EM grid preparation. Therefore, we have toned down our conclusions based on 3D reconstructions from negative stain data and now explicitly refer to “ γ -TuSC-Spc72 complexes co-expressed with Mzt1”, rather than γ -TuSC/Spc72/Mzt1 complexes:

“Using size-exclusion chromatography (SEC, Supplementary Fig. 2b,c) combined with negative stain EM analysis (Supplementary Fig. 2d,e), we could not find evidence that co-expression of Mzt1 impacts oligomer formation or the overall architecture of γ -TuSC rings.”

And in the legend of Supplementary Figure 2: “His-GFP-Mzt1 dissociation during EM grid preparation cannot be entirely excluded, because no additional density attributable to Mzt1 can be observed.”

3) There is significant work from the Agard lab (Kollman et al. Nature 2010, NSMB 2015, Brilot et al. eLife 2021) and more recently by Dendooven and colleagues (NSMB 2024) that characterizes the conformational changes of the γ -TuSC during activation. The discussion of the conformational changes in the TuSC upon Spc72 binding could be significantly improved if it included a comparison examining to what extent there are conserved similarities or significant

differences between the conformational changes in the structures presented in this manuscript and previously determined structures γ -TuSC structures.

We thank the reviewer for this suggestion. We have included a comparison of conformational changes involved in γ -TuSC oligomerization by *C. albicans* Spc72 and *S. cerevisiae* Spc110 in the result section and have added a new Supplementary Figure 5 to support this comparison. Furthermore, we now compare the binding modes of CM1 motifs from *C. albicans* Spc72 and *S. cerevisiae* Spc110 in detail in Supplementary Fig. 6e.

4) The authors suggest that CM1 dimerization performs an important and conserved function in γ -TuRC assembly and activation. However, significant evidence indicates this may not be accurate:

4.1) previous work by Brilot et al. (eLife 2021), and more recently by Dendooven et al. (NSMB 2024) shows that a γ -TuSC rings bound to a monomeric CM1 sample the active conformation and that a monomeric CM1 is present at the inter-TuSC when microtubules are nucleated at the nuclear face of the spindle pole body.

Our analyses of CM1 sequences over a wide range of organisms, combined with the structural work by Brilot et al. (eLife 2021) and Dendooven et al. (NSMB 2024), strongly indicate that CM1 dimerization is a conserved feature with the one exception being Spc110 in *S. cerevisiae*. This suggests that alternative mechanisms may underly γ -TuRC assembly and activation specifically in the case of Spc110 from *S. cerevisiae*. This notion is supported by the fact that the interactions between γ -TuSCs and CM1 motifs strongly differ between *S. cerevisiae* Spc110 and *C. albicans* Spc72 (Supplementary Fig. 6e), while they are rather similar in case of *C. albicans* and humans (Fig. 4b), both with dimeric CM1 motif binding. Similarly, binding of the dimeric CM1 motif transitions CM1-bound γ -TuSC units stoichiometrically into MT-compatible geometry, both in the case of *C. albicans* Spc72 (this study) as well as human CDK5RAP2 CM1 (spokes 1-8; Xu et al.¹, Serna et al.²), while this is the case only for a minority fraction of γ -TuSCs in complex with the monomeric *S. cerevisiae* Spc110 CM1 motif, which form oligomerized γ -TuSC rings in a mostly ‘open’ conformation before MT assembly. This suggests distinct effects of monomeric and dimeric CM1 motif binding on γ -TuRC activation.

Overall, this indicates that observations made specifically for *S. cerevisiae* may most likely not be broadly applicable to other systems.

4.2) All of the major contacts between Spc97, Spc98 and Spc72 highlighted in Figure 3 and Supplementary Figure 5 appear to be between Spc97 or Spc98 and the CM1_{in} of Spc72, further suggesting the CM1_{out} helix may in fact be dispensable.

CM1_{out} directly contacts Spc97 (contact 1 in Supplementary Fig. 6c) and our new MST measurements (see Reviewer #1, comment 4.3 for details) indicate that these contacts significantly contribute to the binding affinity between the dimeric Spc72 CM1 motif and γ -TuSCs.

Additionally, as raised in the Discussion section, dimerization may impart mechanical rigidity or stability on the CM1_{in} helix that could be required for its function as a ‘molecular wedge’, as suggested by Xu et al.¹. We have expanded the respective discussion section to make this clearer.

Overall, our data indicate that the CM1_{out} helix is needed for γ -TuSC oligomerization and activation.

4.3) The evidence presented (3R mutant, pulldowns and in vivo characterization of mutations) does not clearly establish whether the triple mutant disrupts γ -TuSC oligomerization due to a disruption of CM1 dimerization or through some other mechanism, which could include disrupting one of the binding interfaces on Spc97 or Spc98.

In response to this and other comments, we have added extensive new experiments to better characterize the Spc72^{3R} mutant. These experiments allowed us to derive a causal link for why CM1 motif dimerization is essential for γ -TuSC oligomerization.

First, we confirmed that the 3R mutation disrupts CM1 dimerization. Using SEC and mass photometry experiments of wild-type and mutated MBP-CM1²³¹⁻²⁶⁸, we observed that MBP-CM1 dimerizes in a concentration-dependent manner, while MBP-CM1^{231-268,3R} remains a monomer even at high concentrations, as expected (Supplementary Fig. 9a-d).

Second, we characterized the binding affinity of wild-type and mutated CM1 to γ -TuSC using MST measurements. For these experiments, we used CM1 variants with an N-terminal GST tag (GST-CM1²³¹⁻²⁶⁸ and GST-CM1^{231-268,3R}), aiming to mimic an Spc72 coiled-coil region directly C-terminal of the CM1 motif that may impact on CM1 dimerization (Supplementary Fig. 9e-g). In MST measurements, we observed a 4-fold increased K_D value for GST-CM1^{231-268,3R} compared to wild-type GST-CM1²³¹⁻²⁶⁸ (Fig. 4e). Since the mutated CM1 motif residues clearly do not contribute to the interface between CM1_{in} and the γ -TuSC in our cryo-EM reconstruction (Fig. 4d), this indicates that the CM1_{out} helix significantly participates in binding of the dimeric CM1 coiled-coil to the γ -TuSC via contact 1 (Supplementary Fig. 6c).

Thus, this provides an explanation for why CM1 motif dimerization is essential for γ -TuSC oligomerization.

4.4) The authors suggest that coiled coils C-terminal to the CM1 in Spc110 may help it function in the absence of CM1 dimerization, but the authors model similarly dimeric C-terminal coiled coils in Figure 6c,d. This logic should then also apply to the 3R mutant constructs presented in this work, and the 3R mutants should also stimulate γ -TuSC assembly if the only effect of the mutations is to render the CM1 helix monomeric.

In *S. cerevisiae*, these coiled coil segments stably interact with the γ -TuSC oligomer, as indicated by cryo-EM reconstructions of *S. cerevisiae* γ -TuSC bound to Spc110 (see Brilot et al. 2021³), while these coiled coil segments in *C. albicans* Spc72 are completely flexible with respect to the γ -TuSC, as indicated by their absence in our cryo-EM reconstruction. Furthermore, there is no sequence homology between the respective coiled coil segments in *C. albicans* Spc72 and *S. cerevisiae* Spc110. Overall, this indicates that the function of coiled coil segments C-terminal to the CM1 motif are most likely different between *C. albicans* Spc72 and *S. cerevisiae* Spc110, suggesting that any putative compensatory role in *S. cerevisiae* Spc110 may not be conserved in *C. albicans* Spc72. In response to this comment and due to the speculative character, we decided to remove this section from the discussion.

4.5) Supplementary Figure 7 should include a comparison, perhaps including an overlay, with the Spc110 CM1-bound closed γ -TuSC structure as it may be helpful in examining whether any substantive differences in the structures of γ -TuSCs bound to monomeric and dimeric CM1 helices suggest a role for CM1 dimerization.

We thank the reviewer for this suggestion and have now included a comparison between the binding modes of dimeric *C. albicans* Spc72 and monomeric *S. cerevisiae* Spc110 CM1 motifs in Supplementary Fig. 6e. The *S. cerevisiae* Spc110 CM1 motif binding site is considerably shifted towards the neighboring γ -TuSC unit compared to *C. albicans* Spc72, illustrating the overall divergent architecture of the CM1- γ -TuSC interactions in these two cases. However, how these substantially divergent binding sites are linked to the mono-/oligomeric state of CM1 motifs in Spc72 and *S. cerevisiae* Spc110 is difficult to rationalize. We now discuss this in the main text, as part of a comparison between available structures of ascomycete yeast γ -TuSCs, as was requested by the Reviewer #1, point 3.

4.6) The SEC trace in Figure 4f for the 3R mutant does not appear to show pronounced peaks for either assembled γ -TuSC oligomers or monomers. This suggests the 3R mutations may not simply inhibit γ -TuSC oligomerization.

In response to this and other comments, we have revisited our SEC data. We compared the SEC trace of unbound γ -TuSC monomers (Supplementary Fig. 1e,f) with the SEC traces of γ -TuSC monomers co-expressed with Stu2/Spc72¹⁻⁵⁹⁹ or Stu2/Spc72^{1-599,3R}. We observed that γ -TuSC monomers co-expressed with Stu2/Spc72^{1-599,3R} elute at the same volume as unbound γ -TuSC monomers (now added to Figure 4g), while γ -TuSC monomers co-expressed with Stu2/Spc72¹⁻⁵⁹⁹ are shifted to higher molecular weight. This suggests that Spc72^{1-599,3R} mostly dissociates from γ -TuSC monomers during SEC, while Spc72¹⁻⁵⁹⁹ remains bound. This is fully consistent with MST measurements, which indicated lower γ -TuSC binding affinity for GST-CM1^{231-268,3R}.

Thus, the peak annotated as ' γ -TuSC monomer' in the initial manuscript version in fact referred to 'monomeric γ -TuSC/Stu2/Spc72¹⁻⁵⁹⁹'. We have updated all figures including SEC traces accordingly.

Please also see our response to Reviewer #3, comment 3-(2).

4.7) To better address the role of CM1 dimerization in γ -TuSC oligomerization, the authors could generate Spc72 heterodimers using SpyCatcher-SpyTag where only one wildtype Spc72 CM1 helix is present.

We thank the reviewer for this suggestion. Indeed, we had considered generating Spc72 heterodimers in a controlled and targeted manner, e.g. using the SpyCatcher-SpyTag system, but even if heterodimers would successfully form, wild-type CM1 motifs in two different heterodimers are likely to dimerize and function as wild-type homodimers of Spc72, as suggested by SEC of MBP-CM1²³¹⁻²⁶⁸ (Supplementary Fig. 9b).

The evidence presented here also does not suggest a defined mechanism for the importance of dimerization in γ -TuSC assembly. The authors should more directly establish the importance of dimerization before stating that it is essential to γ -TuRC assembly and activation.

Based on our MST measurements, we could clearly establish a role for the second CM1 motif (CM1_{out}) in high affinity binding of Spc72 to γ -TuSCs and, as a consequence, in γ -TuSC oligomerization.

In response to this comment, we have modified the text to reflect this mechanism.

5) Supplementary Figure 3 should include the relevant map-to-model FSC curves.

Map-to-model FSC curves have been added to Supplementary Fig. 3c,d and Supplementary Fig. 8c.

6) Supplementary Figure 4 indicates that multi-body refinement was performed on the TuSC rings. Is this data discussed or presented elsewhere in the manuscript? If yes, FSCs should also be included for this data.

We now state more clearly that multi-body refinement was performed on the γ -TuSC rings and, as requested by the Reviewer, we have included the corresponding FSCs in Supplementary Figure 3d.

7) The authors should explain in the methods how masks used in refinement, focused classification, multi-body refinement, and for FSC estimation were generated and used. The method used to calculate the map-to-model FSCs should also be described. Describe which regions only allowed C-beta truncated residues to be built.

The regions only allowing building of C-beta truncated residues, the method for map-to-model FSC generation and the generation and use of masks throughout data processing have now been outlined in the Methods section, as requested.

8) The appearance of the data in Figures, 1, 3 and supplementary figure 5 all appears consistent with the resolution claimed and the FSC curves. At this resolution (3.6Å), it can be difficult, especially in lower quality regions of maps, to clearly define side-chain conformations and interactions. The authors should include more data clearly showing the fit of the model to the experimental map, focusing on the CM1 helix region presented in Supplementary Figure 5 and the various detailed interactions presented there.

We have included new panels illustrating the fit of the model to the map in detail in Supplementary Figure 6b of the revised manuscript.

9) The authors use structural prediction tools to guide the identification of two binding sites of the Stu2 C-terminal helix to Spc72. Using immunoprecipitation experiments, they show that

Spc72 residues 300-350 and 430-480 form two binding sites for the Stu2 C-terminal helix. I do not agree with the authors' claim that "by combining systematic co-IP experiments and AlphaFold2-based structure predictions, we could characterize the interaction between Stu2 and Spc72 at the residue level". The authors should tone down these claims or perform experiments such as extensive site-directed mutagenesis to probe or validate this interaction at the residue level.

In response to this and other comments, we have thoroughly validated the two Stu2 binding sites on Spc72, as well as the involved interfaces on both proteins, by $^1\text{H}/^2\text{H}$ -exchange mass spectrometry (HX-MS) and detailed site-directed mutagenesis. Please refer to Reviewer #2, comment 3, where we have addressed this comment in detail.

10) It is unclear if one or both Stu2 binding sites on Spc72 are required, or if the higher-affinity binding site located between residues 430-480 functions as the main binding site in vivo.

Because *C. albicans* Spc72 is not functional in *S. cerevisiae* (our unpublished data), we have pursued yeast-two-hybrid (Y2H) experiments to address this comment (Fig. R1). To avoid strong overexpression of Spc72 constructs, we took advantage of the leakiness of the *GALI* promoter in the presence of glucose, instead of inducing the *GALI* promoter with galactose. While this approach minimizes the risk of artificial interactions due to very high concentrations of the protein of interest, color development takes longer. Stu2 interacted clearly with Spc72, but failed to bind to an Spc72 mutant lacking the primary Stu2 binding site, while reduced Stu2 binding was observed for an Spc72 mutant lacking the secondary Stu2 binding site. Thus, the primary binding site is essential for robust Spc72-Stu2 interaction in this system, while the secondary binding site plays a supporting role. This is fully consistent with our model for Spc72-Stu2 interaction derived from co-IP experiments, HX-MS and AlphaFold-based structure predictions.

Figure R1. Yeast two-hybrid (Y2H) experiments confirm the role of the primary and secondary Stu2 binding sites on Spc72 *in vivo*. a) *C. albicans* *SPC72*, *Spc72*^{Δ300-350} and *Spc72*^{Δ430-480} were subcloned into Y2H vector pMM5-Myc⁴. Expression levels were tested in *S. cerevisiae* SGY37 cells by Western blotting with an anti-Myc antibody, using GAPDH as a loading control. Note, *Spc72*^{Δ300-350}-Myc-LexA and *Spc72*^{Δ430-480}-Myc-LexA are slightly downshifted compared to *Spc72*-Myc-LexA because of the deletion of 50 amino acids. b) Indicated plasmid constructs from *C. albicans* *SPC72*, *STU2*, and *S. cerevisiae* *SFI1*, *SPC29*⁵ were tested for interactions *in vivo* using Y2H experiments using an X-Gal overlay assay after 5 days at 30°C without induction of the *GALI* promoter. The weak *S. cerevisiae* Sfi1-*C. albicans* Stu2 interaction can be explained by the known interaction of Sfi1 with Kar1⁶, which in turn interacts with *Spc72*⁷, which interacts with Stu2.

11) Could the authors comment on why the newly modelled regions in EMD-21985 were not previously modelled, or why they could now be modelled? I am concerned that this region may have poor density that led to this region not being modelled previously.

Previously, EMD-21985 was only used for modelling the CM1 motif of CDK5RAP2 and the associated N-GCP2/MZT2 module.

Reported local resolution for the density segments that were not modelled was in the range of 3.5-4.5 Å, very similar to the modelled regions⁸. Also, visible density features in the unmodelled and modelled GCP2 and CDK5RAP2 CM1 regions of EMD-21985 are comparable and consistent with the reported local resolution range.

Thus, while we can only speculate about the reason, time constraints and/or a race for publication may have been the reason for the authors to not update their original model for CDK5RAP2-associated elements in GCP2.

Minor comments

1) In Supplementary Table 1, the estimated B-factor for the masked single γ -TuSC map suggests a higher resolution should have been achieved given the amount of data included in the averaged model. Danev et al. (Nature Commun. 2021) presents a number of structures with higher B-factors that require significantly less data than presented here to achieve 3Å resolution. Is it possible the B-factor estimation is inaccurate?

Danev et al. provide a B-factor determined through running refinements and relating the number of particles needed to reach a certain resolution (the ‘Rosenthal-Henderson plot’). In contrast, we provided a B-factor as automatically determined through Guinier plot fitting in RELION. We have determined the B-factor in the same manner as performed by Danev et al., leading to a higher value consistent with the Reviewer’s expectation of needed particle number to achieve 3.6 Å resolution (Figure R2a). Conversely, determining Guinier plot-based B-factors in RELION for the three reconstructions deposited with Danev et al. similarly yields values that are considerably lower than expected based on the relationship between particle number and achieved resolution (Figure R2b-d).

Figure R2. B-factor determination by different methods. a) B-factor determination of our γ -TuSC reconstruction using the Rosenthal-Henderson plot. Fit B-factor is 195 \AA^2 . b) Guinier plot of EMD-0993 with B-factor fit superimposed (dashed line), yielding a B-factor of -82 \AA^2 (at a final resolution of 2.65 \AA from 390k particles). c) Guinier plot of the EMD-21992 with B-factor fit superimposed (dashed line), yielding a B-factor of -44 \AA^2 (at a final resolution of 2.1 \AA from 636k particles). d) Guinier plot of the EMD-22883 with B-factor fit superimposed (dashed line), yielding a B-factor of -70 \AA^2 (at a final resolution of 2.5 \AA from 625k particles).

2) The authors should clarify, either in figure legends or methods, how the EM maps were sharpened, filtered and masked in the various panels where this data is presented.

The requested information has been provided in the Methods section.

3) The data presented in Fig 3e and supplementary figure 7 are similar in design and conclusions to the work performed by Lyon et al (Mol. Biol. Cell. 2016) on Spc110. This work should probably be cited as corroborating evidence of the central role of CM1 in γ -TuSC assembly.

While we studied the importance of γ -TuC receptor dimerization at the level of the CM1, Lyon et al. study the effect of (higher order) oligomerization of the γ -TuC receptor outside the CM1 motif, using Spc110, in which the CM1 motif itself does not dimerize. We acknowledge the importance of both modes of oligomerization and have added the following sentence to the discussion, citing the work of Lyon et al.:

“Beyond dimerisation at the level of the CM1 motif, higher-order oligomerisation of γ -TuC receptor proteins imposed by the SPB components Spc42⁹ or Nud1¹⁰ likely additionally contributes to γ -TuSC oligomerisation at physiological γ -TuSC concentrations¹¹.”

4) On line 644 “As an independent means of determining...” should be changed to “As an unbiased means of determining...” to match the main text.

The sentence has been updated accordingly.

5) Update Supplementary Table 1 with EMDB and PDB IDs.

EMDB and PDB IDs will be provided in Supplementary Table 1 and the data availability section upon provisional acceptance of the manuscript.

Reviewer #2 (Remarks to the Author):

This manuscript reports the cryo-EM structure of the cytoplasmic γ -TuRC from *C. albicans*, representative of ascomycete yeast. The reported structure was reconstituted from co-expression of Spc97, Spc98, γ -tubulin, Spc72 and Stu2. The authors determined a high resolution (3 \AA) of the γ -TuSC protomers. They used the resultant model to build the entire γ -TuRC complex at 8 \AA resolution. The authors observe the Spc97, Spc98 and γ -tubulin subunits of γ -TuSC protomers,

and the CM1 motif of Spc72. Although present in the sample used for cryo-EM analysis, cryo-EM density was not visible for Stu2. Based on AlphaFold2 predictions, and biochemistry, the authors propose a mechanism of Stu2 interaction with Spc72, suggesting that multiple copies of Stu2 might bind to an Spc72 dimer.

The authors test (and validate) the mode of binding of Spc72 and suggest it plays a role in assembly of γ -TuRC. Co-expression of Spc97, Spc98, γ -tubulin with Mtz1 indicated that this subunit associates with γ -TuRC.

This is an interesting paper describing significant and important results. In general the experiments are performed vigorously and clearly explained. The figures are of high quality.

The main findings include:

1. The cytoplasmic γ -TuRC with Spc72 adopts a closed helical conformation that matches the 13-prot filament geometry of the minus end of a microtubule filament in the absence of binding the minus end of a microtubule filament. This is in contrast to the structure of the in vitro reconstituted *S. cerevisiae* γ -TuRC with Spc110 that adopts an open conformation in the absence of microtubules.
2. Spc72 bridges neighbouring γ -TuSC protomers, explaining how it can promote the closed conformation of γ -TuRC.
3. Spc72 binds to γ -TuRC through CM1 motifs as a helical dimer in contrast to how Spc110 binds γ -TuRC through a single α -helix.
4. The model for how Stu2 interacts with Spc72 is of interest – although note caveats below.

Subject to revisions, this manuscript warrants publication in Nature Communications.

We thank Reviewer #2 for the positive evaluation of our manuscript.

Major points.

1. The major concern is the description of how Stu2 binds to Spc72. The authors propose that the absence of cryo-EM density for Stu2 is due to Stu2 binding to a disordered region of γ -TuRC. This is quite likely, but it is also possible that Stu2 dissociated from γ -TuRC during cryo-EM grid sample preparation. The authors should include this possibility in the text.

Please see our response to Reviewer #1, point 1.

2. The authors present AlphaFold models of Spc72 and complexes of the C-terminal α -helix of Stu2 bound to Spc72. Unfortunately it is not possible for the reader to assess the quality of these models because the authors did not include either the coordinates colour-coded by pLDDT score or the PAE plots. Both should be included in the paper.

We have included models coloured by pLDDT score and PAE plots in Supplementary Figure 12. Both indicate high confidence prediction of the 3D structures and interactions of Spc72 and Stu2.

3. Although the authors test the prediction of Stu2 binding to the two modules of Spc72, quite an imprecise mapping is used, ie. deletions of Spc72. Since the authors do not include NMR or CLMS data to test/validate their model, a more precise mutagenesis study is called for. The authors could mutate interacting residues in Spc72 and Stu2 to better validate their model.

In response to this comment, we have thoroughly validated the two Stu2 binding sites on Spc72, as well as the involved interfaces on both proteins.

First, we analyzed the interaction between Spc72²⁹¹⁻⁵⁹⁹ and Stu2 by ¹H/²H-exchange mass spectrometry (HX-MS), which reports on stabilization of protein secondary structure elements originating from protein-protein interactions. We observed strong and specific protection of Spc72 peptides located in the primary and secondary Stu2 binding sites. Notably, strong protection of the secondary Stu2 binding site was achieved only under higher Stu2 concentrations (2:1 molar ratio), which suggests lower affinity compared to the primary binding site, in line with our co-IP experiments. Thus, HX-MS analysis confirmed the two Stu2 binding sites on Spc72 at peptide level. We have included these data in Figure 6d and Supplementary Figs. 13 and 14).

Secondly, to confirm the interfaces predicted for the two Stu2-Spc72 binding sites, we mutated the residues predicted to interact either on Stu2 or Spc72 and tested interaction by co-immunoprecipitation experiments. Consistent with the predictions, mutating the Stu2 residues predicted to directly interact with Spc72 completely abrogated or substantially reduced the interaction with wild-type Spc72 fragments harboring either the primary or secondary Stu2 binding site, respectively. Vice versa, mutating Spc72 residues predicted to interact with Stu2 completely abrogated or substantially reduced the interaction with wild-type Stu2 for the primary or secondary Stu2 binding site, respectively.

Cumulatively, these new experiments thoroughly validate our model.

4. The model that Stu2 binds to module 1 and module 2 potentially simultaneously could be tested by determining the mass of the Stu2-Spc72 complex using SEC-MALS or mass photometry.

In response to this and other comments, we analyzed the oligomerization status of Spc72²⁹¹⁻⁵⁹⁹ and Stu2, as well as the Spc72²⁹¹⁻⁵⁹⁹/Stu2 complex using mass photometry and SEC-MALS. Mass photometry analysis, as well as SEC-MALS indicated that both Stu2 and Spc72²⁹¹⁻⁵⁹⁹ predominantly form stable homodimers in isolation (Supplementary Fig. 15b,c). Under conditions of mass photometry, Spc72²⁹¹⁻⁵⁹⁹ and Stu2 failed to form stable complexes, most likely due to the very low concentrations necessary for mass photometry (Supplementary Fig. 15d). At higher concentrations used in SEC-MALS experiments, 20% of total protein mass formed high molecular weight Stu2/Spc72²⁹¹⁻⁵⁹⁹ complexes of heterogeneous size (Supplementary Fig. 15d,e), consistent with the formation of large Spc72²⁹¹⁻⁵⁹⁹/Stu2 networks, possibly due to the dimer status of both interacting proteins and the presence of multiple binding sites on each dimer. The formation of such networks could be promoted by *in vitro* incubation of Stu2 and Spc72²⁹¹⁻⁵⁹⁹ in the absence of γ -TuSC and may not reflect the situation in cells.

Indeed, the formation of much more defined complexes is indicated by quantification of band intensities for Stu2 and Spc72¹⁻⁵⁹⁹ in Coomassie-stained gels of γ -TuRC/Spc72¹⁻⁵⁹⁹/Stu2 complex

co-expressed and purified by SEC. The relative intensities suggest that each Spc72 dimer binds approximately two Stu2 dimers (Fig. 6e). This approach was validated by relative quantification of band intensities for Spc97, Spc98 and γ -tubulin, for which the expected 1:1:2 ratio was obtained.

Finally, yeast-two-hybrid experiments suggested that both Stu2 binding sites in Spc72 contribute to stable complex formation in the cellular context (Fig. R1). See Reviewer 1, comment 10.

Cumulatively, these experiments indicate that each Spc72 dimer binds two Stu2 dimers.

5. Fig. 1a. Stoichiometry of subunits. The authors state that all subunits are present in an approximately stoichiometric ratio. 'Ratio' is a bit ambiguous. Does it mean equal ratio? An equal stoichiometric ratio isn't obvious from the SDS PAGE gel. Stu2 is more abundant. A more convincing gel is that shown in after SEC (Supplementary Fig. 1d). As Spc72 is a dimer bound to an Spc97-Spc98 heterodimer, it isn't formally correct to state that all subunits have equal stoichiometry.

We agree with the reviewer and have removed the claim of approximate stoichiometric ratio.

6. Supplementary Fig. 10a is difficult to follow. The locations of CM1 and the hinge between the coiled-coil modules 1 and 2 are not clear. Possibly colour N- to C-termini with a blue-to-red colour ramp, and label more clearly using arrows.

We have improved the figure according to the Reviewer's suggestions.

Minor points.

1. The authors state that *S. cerevisiae* nuclear γ -TuRC adopts an open conformation, referencing 13 and 14 from 2010, 2015. The Brilot paper (ref 10) describes reconstituted γ -TuRC. More recent higher resolution cryo-ET reconstructions of the nuclear γ -TuRC from enriched spindles showed that all γ -TuRC are in a closed conformation capping the minus end of microtubules. This section of the Introduction does not reflect the current knowledge of the field, and incorrectly conflates the structures of in vitro reconstituted γ -TuRC with Spc110 with the actual structure of the nuclear γ -TuRC in situ.

We agree with the Reviewer and have updated the introduction to accurately reflect the current state of knowledge:

“When reconstituted, the majority of nuclear γ -TuSC oligomers were characterised by an ‘open conformation’^{12,13}, in which γ -tubulin molecules were spaced further apart than in a 13-protofilament MT, and only a minority fraction of γ -TuSC rings sampled a MT-compatible and more active ‘closed’ arrangement^{3,12}. In contrast, γ -TuSC rings imaged in native SPBs were observed exclusively in the closed conformation while capping MTs¹⁴, where the conformation was presumably stabilised by lateral interactions between MT protofilaments.”

2. Show colour guide of all subunits in Fig. 2 (or otherwise label subunits).

A colour guide has been implemented in Fig. 2.

3. Table 1: *S. cerevisiae* not *S. pombe*.

This is not a typo; we initially used the available microtubule structure from *S. pombe* (PDB-5MJS), because searching of the RCSB PDB for a ‘microtubule’ or ‘tubulin’ structure of source organisms *S. cerevisiae* or *C. albicans* yielded no atomic model for a high-resolution structure. Prompted by the Reviewer’s comment, however, we revisited this and found that the RCSB PDB search algorithm is yeast strain-specific, i.e. searching for source organism *S. cerevisiae* does not identify structures deposited with source organism ‘*S. cerevisiae* S288C’. Taking this into account, we identified a high-resolution microtubule structure of ‘*S. cerevisiae* S288C’ and updated the comparison in Table 1 with the new model. As expected, the numbers are virtually identical.

4. Please show side chains in Supplementary Fig. 7b.

We have added side chains to the extended parts of the model when displaying the density fit in Supplementary Fig. 8b.

5. There are two lanes 1-5 in Fig. 6a, b. This is a little confusing. Similar for Supplementary Figs. 1b, 9b, c.

We opted to label lanes corresponding to the input and immunoprecipitated sample of the same experiment with the same number to aid comparison between matching input and IP lanes. We have now clarified this in the respective figure legends.

6. Reference 47 should be updated. PMID: 38609662.

7. Reference 60 should be updated. PMID: 38408488.

We have updated references for all cited preprints that were published in the meantime.

Reviewer #3 (Remarks to the Author):

Microtubule (MT) is a highly conserved cytoskeleton that plays fundamental roles in cellular activities. Obtaining a mechanistic explanation of the MT nucleation process in a wide range of organisms helps understand the essential principles of MT regulation.

In this work, the authors revealed how budding yeast Spc72, a yeast homologue of CDK5RAP2, contributes to the oligomerization of the γ -TuSC through the CM1 motif by solving the reconstituted assembly of the γ -TuSC using cryo-EM. The authors also tried to unveil the structural arrangement of Stu2, a yeast homologue of XMAP215/chTOG. They mapped the

domains responsible for the Spc72-Stu2 interaction, but the structural information could not be obtained, presumably due to the flexibility of the complex. The overall data quality is excellent and compelling, although some biochemical data can be further improved. Hence, I believe the work is suitable to be published in Nature Communications after addressing the issues listed below.

We thank Reviewer #3 for the positive evaluation of our manuscript.

Major issues

1. Molar stoichiometry of Spc72 in the γ -TuSC assembly

The manuscript proposes that a dimer of Spc72 interacts with Spc97/98 of the γ -TuSC assembly (hence a total of 12-14 Spc72 in one γ -TuRC?). However, direct evidence is not provided; as far as I am aware, the sole relevant information would be the Coomassie-stained gel image (Fig. 1a), which is described as “all γ -TuSC components and Spc72 in an approximately stoichiometric ratio (page 4)”, which can be misleading if two molecules of Spc72 bind to one molecule of Spc97 (and the neighbouring Spc98).

Motivated by this and Reviewer #2's comments, we have removed the statement about all components being in an approximately stoichiometric ratio.

Therefore, I would like the authors to examine the molecular weight (MW) of the complex containing Spc72. It may be achieved by techniques such as multi-angle light scattering (MALS), small-angle X-ray scattering (SAXS) or mass photometry (MP). For this experiment, I would suggest the authors exclude Stu2 as Stu2 inclusion may make the analysis over-complicated. Instead, I would like the authors to examine one of the following two samples. First option: MW of the SEC-purified γ -TuC assembly consisting of the γ -TuSC (with FLAG-Spc98, Spc97 and γ -Tubulin) and Spc72(1-599). Second option: the MW of the SEC-purified γ -TuSC decorated with Spc72.PA(1-599). The latter sample may be more homogenous, and the data analysis may be more straightforward.

To address this comment, we have quantified band intensities in Coomassie-stained gels of γ -TuRC/Spc72¹⁻⁵⁹⁹/Stu2 complex co-expressed and purified by SEC (Fig. 6e). Here, the relative intensities suggest that each γ -TuSC unit binds one Spc72¹⁻⁵⁹⁹ dimer. This approach was also validated by relative quantification of band intensities for Spc97, Spc98 and γ -tubulin, for which the expected 1:1:2 ratio was obtained.

The presence of one Spc72¹⁻⁵⁹⁹ dimer on each γ -TuSC unit is further supported by our cryo-EM reconstruction, in which we observe clear and unambiguous density for two copies of the Spc72 CM1 motif bound to each γ -TuSC unit (Fig. 3a), as well as mass photometry and SEC-MALS experiments, in which the vast majority of Spc72 forms stable dimers (Supplementary Fig. 15).

Hence, we conclude that one Spc72 dimer binds per γ -TuSC unit.

2. The minimum Spc72 motif sufficient for the γ -TuSC oligomerization

The authors demonstrate that the Spc72 CM1 motif interacts with Spc97, and T234/T236 at the CM1 N-term is in contact with the neighbouring Spc98 by analysing the cryo-EM structure. They

also show that the Spc72.PA mutant cannot help the oligomerization of γ -TuSC. However, whether the Spc72 CM1 motif (including T234/T236) is sufficient for the γ -TuSC oligomer is not shown. Therefore, I would like the authors to examine whether adding the Spc72 CM1 motif (Spc72.231-268) to the γ -TuSC causes the γ -TuSC oligomerisation. In this experiment, Stu2 can be excluded.

Motivated by the Reviewer's comment, we first aimed to generate a minimal CM1 motif construct able to dimerize and bind to the γ -TuSC at high affinity. We initially expressed and purified an MBP-tagged minimal Spc72 CM1 motif (MBP-CM1²³¹⁻²⁶⁸, Supplementary Fig. 9a-c) and observed that CM1 dimerization was concentration-dependent (< 40 nM: monomeric; 100 μ M: predominantly dimeric), which suggested, based on our analysis of the dimerization deficient CM1-3R mutant, that the monomeric MBP-CM1²³¹⁻²⁶⁸ motif alone may most likely not be able to bind to γ -TuSCs at high affinity. In Spc72, the CM1 motif is directly followed by a coiled-coil region that most likely promotes CM1 dimerisation by keeping CM1 helices in spatial proximity. To mimic this effect and promote CM1 dimerisation also under low concentrations, we generated a GST-tagged construct (GST-CM1²³¹⁻²⁶⁸, Supplementary Fig. 9e,f). Using MST measurements, we confirmed that the dimerized CM1 motif in GST-CM1²³¹⁻²⁶⁸ was able to bind to the γ -TuSC with a K_D of about 100 nM (Fig. 4e), providing an experimental basis for investigating CM1²³¹⁻²⁶⁸-induced γ -TuSC oligomerization.

To test whether the dimerized CM1 motif in GST-CM1²³¹⁻²⁶⁸ was sufficient to efficiently induce γ -TuSC oligomerization, we incubated 1 μ M γ -TuSC with 15 μ M GST-CM1²³¹⁻²⁶⁸ or a buffer control and analyzed γ -TuSC oligomerization by negative stain EM and 2D class averaging (Figure R3). Surprisingly, we did not observe enhanced oligomerization induced by GST-CM1²³¹⁻²⁶⁸ under the experimental conditions used. Our observations contrast those recently made with the human system, where addition of the CDK5RAP2 CM1 motif promotes oligomerization of the human γ -TuSC² although the concentration of human γ -TuSC in the CDK5RAP2 oligomerization experiment was similar to what we used in our experiment with *C. albicans* γ -TuSC and GST-CM1²³¹⁻²⁶⁸ (Fig. R3). This difference may originate from the substantially higher intrinsic oligomerization propensity of human compared to *C. albicans* γ -TuSC, as evidenced by formation of human γ -TuSC oligomers even in the absence of the CDK5RAP2 CM1 motif^{2,15}.

Overall, our results suggest that parts of Spc72 other than the CM1 motif might be required, e.g. to facilitate higher order spatial arrangement of CM1-bound γ -TuSCs to promote γ -TuSC oligomerization. However, we cannot rule out that the CM1 motif alone can induce oligomerization under specific experimental conditions. In the absence of conclusive evidence, we prefer not comment in the manuscript on whether the dimeric Spc72 CM1 motif is sufficient for γ -TuSC oligomerization.

Figure R3. GST-CM1²³¹⁻²⁶⁸ alone is not sufficient for γ -TuSC oligomerization: (a,b), representative negative stain EM micrograph and 2D class averages of γ -TuSC without (a) and with 15 μ M GST-CM1²³¹⁻²⁶⁸ (b). Scale bars are given.

3. The Spc72.3R phenotype

3-(1)

The authors claim that the Spc72.3R mutant still can bind to the γ -TuSC based on the pull-down experiment (Fig.4e). However, the pull-down experiment is not quantitative, and the band intensity of Spc72.3R in the FLAG-IP panel (Fig. 4e) seems to be reduced. Furthermore, in Fig. 4(g), fraction 11.75 ml hardly contains the component of the γ -TuSC, unlike the case shown in Fig. 3(e) where signals of γ -TuSC components and Spc72.PA are evident in fraction 11.75 ml. Instead, Fig. 4(g) shows that fractions 12.5 ml and 13.25 ml contain the stronger γ -Tubulin signals as if the γ -TuSC is not associated with Spc72. In fact, the signal of His-Spc72 seems much weaker than the signals of Spc98 and Spc97 in these fractions in Fig. 4(g).

Based on a series of new experiments (see directly below), we agree with the Reviewer that Spc72^{1-599,3R} most likely has lower binding affinity compared to wild-type Spc72¹⁻⁵⁹⁹. Our cryo-EM reconstruction clearly indicates that the interface between CM1_{in} and the γ -TuSC is not affected in Spc72^{1-599,3R}, which suggests that the reduced binding affinity of Spc72^{1-599,3R} most likely originates from the missing contribution of the CM1_{out} helix in γ -TuSC binding. As a consequence, Spc72^{1-599,3R}, but not Spc72¹⁻⁵⁹⁹ or Spc72^{1-599,PA}, most likely dissociates from the γ -TuSC during SEC experiments, as proposed by the Reviewer. Consistently, we observed that γ -TuSC monomers in the presence of Spc72^{1-599,3R} predominantly elute at the same volume as uncomplexed γ -TuSC monomers (Figure 4g,h). This provides an explanation and a clear causal link for why CM1 dimerization is essential for γ -TuSC oligomerization and activation and explains the observed results in Fig. 4g (now Fig. 4h).

Please also refer to our response to Reviewer #1, comment 4.6.

Therefore, I would like the authors to conduct a quantitative method (such as ITC, SPR, bilayer interferometry (BLI) or microscale thermophoresis as the authors used in their previous studies) to examine whether the 3R mutation affects the interaction between the γ -TuSC and Spc72. Stu2

We have conducted MST measurements and observed a 4-fold increased K_D value for γ -TuSC binding by GST-CM1^{3R} compared to wild-type GST-CM1 (see Reviewer #1, comment 4.3 for details).

Moreover, quantification of the pull down experiment in Fig. 4f indicates that the pull-down efficiency of Spc72^{1-599,3R} is reduced 1.7 ± 0.2 fold (mean \pm SD).

As I explain below in (2), there might be a possibility that Spc72.3R may interfere with the formation of the γ -TuSC. Therefore, if the γ -TuSC - Spc72 interaction kinetics shows complex outcomes using ITC/SPR/BLI, it may be a good idea to simply look at the K_D of Spc97 and Spc72.WT/Spc72.3R (without including Spc98 and γ -Tubulin).

Please see our response to Reviewer #3, comment 3-2 (directly below), where we establish that Spc72^{1-599,3R} does not interfere with γ -TuSC formation.

3-(2)

In Fig. 4(f), an intriguing elution profile is presented for the γ -TuSC/Spc72.3R/FLAG-Stu2 sample. There is hardly any detectable peak that corresponds to the monomeric γ -TuSC. This is in a striking contrast against the result presented in Fig. 3(d) where Spc72.PA interacts with the the γ -TuSC. Furthermore, the in vivo phenotypes between them are different. It could be that the expression of Spc72.3R could negatively affect the γ -TuSC formation. The authors may argue that Fig. 4(g) shows the successful formation of γ -TuSC. However, the γ -TuSC formation efficacy seems to be greatly reduced because fractions 14.0 – 17.0 ml show substantial signals of Spc98, Spc97 and γ -tubulin, which would represent the various associating status and monomeric status of these molecules. This trend is distinct from the result presented in Fig. 3(e), where most of the signals of Spc98, Spc97 and γ -tubulin are found in fractions 11 – 12.5 ml.

When comparing the SEC traces of γ -TuSC monomers co-expressed with Stu2/Spc72¹⁻⁵⁹⁹ or Stu2/Spc72^{1-599,3R} with the SEC trace of uncomplexed γ -TuSC monomers, we observed that γ -TuSC monomers co-expressed with Stu2/Spc72^{1-599,3R} elute at the same volume as uncomplexed γ -TuSC monomers. This indicates that γ -TuSC monomers are still intact and properly formed in the presence of Spc72^{1-599,3R}.

The shift in elution volume compared to γ -TuSC monomers co-expressed with Stu2/Spc72¹⁻⁵⁹⁹ can be explained by partial dissociation of Spc72^{1-599,3R} and Stu2 under the conditions of SEC (see also Reviewer #1, comment 4.6 for more details), which is consistent with MST measurements suggesting reduced γ -TuSC binding affinity of Spc72^{1-599,3R} compared to Spc72¹⁻⁵⁹⁹.

Thus, the peak annotated as ‘ γ -TuSC monomer’ in the initial manuscript version in fact referred to ‘monomeric γ -TuSC/Stu2/Spc72¹⁻⁵⁹⁹’. We have updated the figure labeling accordingly.

Therefore, I would like the authors to conduct the following experiments.

(a) Examine whether SPC72.3R is a dominant or recessive mutation. The spc72. Δ P55-N62 mutant may be used as a reference, as it represents an example that does not disturb the γ -TuSC formation.

As indicated above, we have established that Spc72^{1-599,3R} does not interfere with γ -TuSC formation.

Still, to directly address the Reviewer’s comment, we have expressed *GALI*-induced *S. cerevisiae* spc72^{3R} in wild type yeast cells and compared growth of cells to *SPC72* overexpression (Fig. 5b). In this experiment, *GALI*- spc72^{3R} impaired cell growth, whereas no such effect was observed with *GALI*-*SPC72* when compared to the *GALI* vector control. This suggests that wild-type Spc72 is replaced by Spc72^{3R} that is unable to induce γ -TuSC oligomerisation at the cytoplasmic side of the SPB.

(b) In the images presented in Supplementary Fig. 6(c) and Supplementary Fig. 8, the expression of Spc72.3R seems to cause defects in the spindle formation. I would like the authors to quantify the spindle formation defect phenotype of the cells expressing Spc72.3R and Spc72. Δ P55-N62.

The spindle phenotype of Spc72. Δ P55-N62 in the experiment was quantified, as requested by the Reviewer. This data was added to Supplementary Fig. 7f.

Previously, we have analyzed the phenotype of spc72^{3R} in W303 Δ spc72 cells. Although *SPC72* is not absolutely essential in the W303 strain background, the defects are strong. This gives rise to dying yeast cells that have broad general defects.

To circumvent this problem for spc72^{3R}, we followed a new strategy. We used a budding yeast strain in which chromosomal *SPC72* was modified to contain an auxin-inducible degron (AID), IAA7. Addition of the auxin analogue IAA triggered the rapid (several hours) depletion of Spc72-IAA7, as shown in Fig. 5d. In these cells, we expressed *SPC72* or spc72^{3R} from the *SPC72* promoter. After 3 h of IAA addition when Spc72-IAA7 was completely degraded, we were able to determine the phenotype of spc72^{3R} without prolonged incubation, avoiding secondary defects. This analysis showed that cytoplasmic microtubules are specifically affected in spc72^{3R} cells, while nuclear MTs are mostly functional, forming metaphase and anaphase spindles (Fig. 5e). We quantified these defects and added the quantification to Fig. 5f. The degron experiment was not repeated for Spc72. Δ P55-N62, because the original experimental setup using W303 Δ spc72 cells was appropriate.

(c) A possible cause of the Spc72.3R in vivo phenotype could be its capability to replace Spc110 at the SPB. I would like the authors to examine the localisation of Spc110 in the presence of Spc72.3R.

This comment was likely prompted by the unexpected nuclear MT defects that we observed in W303 $\Delta spc72 spc72^{3R}$ cells, which were likely caused by dying yeast cells that have broad general defects. Note, W303 $\Delta spc72 spc72^{3R}$ cells were incubated for several days on 5-FOA followed by incubation on SC-Leu plates for 6 days followed by culturing in SC-Leu liquid medium for 2 days for microscopy. As can be seen in Figure R4, the cell size of $spc72^{3R}$ cells is greatly increased and cells are arrested in mitosis for prolonged times. This analysis has now been replaced by analysis in *SPC72-IAA7* cells, where nuclear MTs are shown not to be affected (see our response to Reviewer #3, comment 3-2b).

Still, to address this comment, we measured Spc42 and Spc110 fluorescence intensity at the SPB in *Spc42-mCherry-hgh Spc110-GFP* W303 cells either expressing *SPC72* or $spc72^{3R}$. This analysis showed that Spc110 localization at the SPB was not reduced by $spc72^{3R}$. Because experiments using W303 $\Delta spc72 spc72^{3R}$ cells were replaced by experiments using IAA-induced degradation of wild-type Spc72-IAA7, where such nuclear MT defects were not observed anymore, we opted to include the analysis in the point-by-point response only (Figure R4).

Figure R4. The localization of Spc110 is not negatively affected in $spc72^{3R}$ cells. a) YJP283-1 yeast strain (W303 $\Delta spc72::KanMx6$ pRS316-*SPC72 SPC42-mCherry-hgh*) was transformed with the SacI digested plasmid YIp211-*SPC110-GFP*¹⁶. After confirming that *SPC110-GFP* was successfully tagged via microscopy, the *LEU2*-based integration plasmids pRS305- $spc72^{3R}$, pRS305-*SPC72* and pRS305 were inserted into the genome of the modified YJP283-1 yeast strain (W303 $\Delta spc72::KanMx6$ pRS316-*SPC72 SPC42-mCherry-hgh SPC110-GFP*). Cells were selected on SC-Leu plates at 23°C followed by 5-FOA to remove pRS316-*SPC72*. The localization of Spc110-GFP was determined by fluorescence microscopy in *SPC72* and $spc72^{3R}$ cells. Spc42-mCherry was used as a marker for the SPB. Scale bar, 5 μ m. Three independent experiments were performed. b) The fluorescence intensity of Spc42-mCherry and Spc110-GFP were quantified in ImageJ software and the Spc110-GFP signal was normalized to the Spc42-mCherry signal. Three independent experiments were used for the quantification. n > 30 cells per experiment. p values were produced using the unpaired t test. ****p<0.0001; **p=0.0039.

4. The structural arrangement of Stu2

I understand that the authors aimed to solve the structure of the oligomerised γ -TuSC decorated with both Stu2 and Spc72 (1-599). However, the cryo-EM structure did not show an obvious contribution by Stu2. This is interesting as previous studies (including the ones from the authors' group) showed that Stu2/XMAP215 directly binds γ -TuSC. As the authors discussed, Stu2 flexibility can be the reason, and the limitation in the cryo EM-based approach may be demonstrated in this instance.

Meanwhile, I would like the authors to probe the molar stoichiometry of Spc72(1-599) and Stu2 complex as the authors identified two Stu2 binding sites on Spc72 (1-599). It can ideally be conducted using the Stu2: γ -TuSC:Spc72(1-599) complex, and the molar stoichiometry can be examined both by Coomassie staining of the gels and MW analysis by SAXS/MALS/MP. In this case, it may be better to use sucrose gradient ultracentrifugation to purify the complex as fraction 8.5 ml of the SEC seems to be very close to the void volume, causing the sample heterogeneity, which is shown in Supplementary Figure 1 (d). Alternatively, a complex of Stu2 and Spc72(1-599) can be purified and analysed without the presence of the γ -TuSC.

Please see our response to Reviewer #2, comment 4, in which we describe extensive new experiments to probe the stoichiometry of Spc72 and Stu2 in the complex. Cumulatively, these experiments indicate that each Spc72 dimer binds two Stu2 dimers.

Minor issues

- In order to avoid confusion, I would like to ask the authors to keep stating “Spc721-599” throughout the manuscript, including all the figures. Also, I would like to ask the authors to clearly indicate the presence of a His-tag, if that is the case, in all the figures.

We have modified the manuscript accordingly.

- It is unclear whether Stu2 was included in the result shown in Fig. 4(e). It would be helpful if the authors clarify the experimental set-up of Fig. 4(e).

Stu2 was not included in the experiment. The pull down of γ -TuSC and Spc72¹⁻⁵⁹⁹ or Spc72^{1-599,3R} was performed through the FLAG tag on Spc98. Both is now explicitly mentioned in the figure legend.

- The authors concluded that GFP-Mzt1 has no impact on oligomer formation and the overall structure of γ -TuSC oligomers. However, Supplementary Figure 2 (d) shows that the sample is highly heterogeneous, and it is unclear whether the authors could pick the oligomers involving GFP-Mzt1. Furthermore, it is unclear whether the authors could identify the GFP-Mzt1 density in the structure presented in Supplementary Figure 2(e). Therefore, the statement that “Mzt1 has no impact” can be misleading unless the authors show the GFP-Mzt1 density in the cryo-EM structure presented in Supplementary Figure 2(e).

We have not found any density corresponding to GFP-Mzt1 in our reconstruction, which would be consistent with GFP-Mzt1 binding to the N-terminus of Spc98, flexibly positioned with respect to the γ -TuSC oligomers. While the sample is heterogeneous, the structure of γ -TuSC rings obtained in the presence of Spc72¹⁻⁵⁹⁹ and His-GFP-Mzt1 shows no considerable differences to the structure of γ -TuSC rings obtained in the presence of Spc72¹⁻⁵⁹⁹ and Stu2. However, we cannot exclude that Mzt1 has partially dissociated during EM grid preparation, so we have toned down conclusions based on the negative stain EM data. Please see our response to Reviewer #1, comment 2, for more details.

• There are some typos – for example, figure 4(g) legend “...shown in panel (e) by western blotting” -> maybe “...shown in panel (f) by Western blotting”?

We apologies and have corrected the pointed out typo, as well as critically read the manuscript to fix other typos.

References

- 1 Xu, Y. *et al.* Partial closure of the gamma-tubulin ring complex by CDK5RAP2 activates microtubule nucleation. *Dev Cell* (2024). <https://doi.org:10.1016/j.devcel.2024.09.002>
- 2 Serna, M. *et al.* CDK5RAP2 activates microtubule nucleator gammaTuRC by facilitating template formation and actin release. *Dev Cell* (2024). <https://doi.org:10.1016/j.devcel.2024.09.001>
- 3 Brilot, A. F. *et al.* CM1-driven assembly and activation of yeast gamma-tubulin small complex underlies microtubule nucleation. *Elife* **10** (2021). <https://doi.org:10.7554/eLife.65168>
- 4 Schramm, C., Janke, C. & Schiebel, E. Molecular dissection of yeast spindle pole bodies by two hybrid, in vitro binding, and co-purification. *Methods Cell Biol* **67**, 71-94 (2001). [https://doi.org:10.1016/s0091-679x\(01\)67006-7](https://doi.org:10.1016/s0091-679x(01)67006-7)
- 5 Rüttnick, D., Vitale, J., Neuner, A. & Schiebel, E. The N-terminus of Sfi1 and yeast centrin Cdc31 provide the assembly site for a new spindle pole body. *J Cell Biol* **220** (2021). <https://doi.org:10.1083/jcb.202004196>
- 6 Seybold, C. *et al.* Kar1 binding to Sfi1 C-terminal regions anchors the SPB bridge to the nuclear envelope. *J Cell Biol* **209**, 843-861 (2015). <https://doi.org:10.1083/jcb.201412050>
- 7 Pereira, G., Grueneberg, U., Knop, M. & Schiebel, E. Interaction of the yeast γ -tubulin complex-binding protein Spc72p with Kar1p is essential for microtubule function during karyogamy. *The EMBO Journal* **18**, 4180-4195 (1999). <https://doi.org:https://doi.org/10.1093/emboj/18.15.4180>

- 8 Wieczorek, M., Huang, T. L., Urnavicius, L., Hsia, K. C. & Kapoor, T. M. MZT Proteins Form Multi-Faceted Structural Modules in the gamma-Tubulin Ring Complex. *Cell Rep* **31**, 107791 (2020). <https://doi.org:10.1016/j.celrep.2020.107791>
- 9 Drennan, A. C. *et al.* Structure and function of Spc42 coiled-coils in yeast centrosome assembly and duplication. *Mol Biol Cell* **30**, 1505-1522 (2019). <https://doi.org:10.1091/mbc.E19-03-0167>
- 10 Gruneberg, U., Campbell, K., Simpson, C., Grindlay, J. & Schiebel, E. Nud1p links astral microtubule organization and the control of exit from mitosis. *EMBO J* **19**, 6475-6488 (2000). <https://doi.org:10.1093/emboj/19.23.6475>
- 11 Lyon, A. S. *et al.* Higher-order oligomerization of Spc110p drives gamma-tubulin ring complex assembly. *Mol Biol Cell* **27**, 2245-2258 (2016). <https://doi.org:10.1091/mbc.E16-02-0072>
- 12 Kollman, J. M. *et al.* Ring closure activates yeast gammaTuRC for species-specific microtubule nucleation. *Nat Struct Mol Biol* **22**, 132-137 (2015). <https://doi.org:10.1038/nsmb.2953>
- 13 Kollman, J. M., Polka, J. K., Zelter, A., Davis, T. N. & Agard, D. A. Microtubule nucleating gamma-TuSC assembles structures with 13-fold microtubule-like symmetry. *Nature* **466**, 879-882 (2010). <https://doi.org:10.1038/nature09207>
- 14 Dendooven, T. *et al.* Structure of the native gamma-tubulin ring complex capping spindle microtubules. *Nat Struct Mol Biol* (2024). <https://doi.org:10.1038/s41594-024-01281-y>
- 15 Wurtz, M. *et al.* Modular assembly of the principal microtubule nucleator gamma-TuRC. *Nat Commun* **13**, 473 (2022). <https://doi.org:10.1038/s41467-022-28079-0>
- 16 Huisman, S. M., Smeets, M. F. & Segal, M. Phosphorylation of Spc110p by Cdc28p-Clb5p kinase contributes to correct spindle morphogenesis in *S. cerevisiae*. *J Cell Sci* **120**, 435-446 (2007). <https://doi.org:10.1242/jcs.03342>

Point-by-Point Reply to Reviewers

We are very pleased that Reviewers #2 and #3 did not raise any further concerns and support publication of our revised manuscript in *Nature Communications*.

We thank Reviewer #1 for thoughtful comments. Below we comprehensively address all the specific points raised by Reviewer #1 and elaborate on the corresponding changes in the manuscript.

Reviewer #1 (Remarks to the Author):

The revised manuscript provided by the authors substantively addresses all the points previously raised in the prior review. The authors have made significant revisions to the text throughout, added panels to compare their structural results to prior work in the field, included data to clarify the oligomeric state of Spc72 in the 3R mutant and further probe its binding to γ -TuSC, re-analyzed their SEC data, and significantly improved their analysis of the Spc72/Stu2 binding sites, and provided in vivo data establishing that the secondary binding site may be dispensable. The authors have further included important data used in validating their structures, including several panels showing the quality of the structures, and made their maps and models available to the reviewers. Overall, the manuscript is significantly improved, with many of the major conclusions better supported by the data.

We thank Reviewer #1 for the positive and supportive assessment of our revised manuscript.

While the new data significantly improve the manuscript, there are a few significant questions the new data raises that require additional revisions:

1. The additional experiments performed by the authors establishes that the 3R mutant exhibits reduced dimerization, reduces the affinity of the CM1 interaction with γ -TuSC, and still binds to γ -TuSC. However, there are still significant questions as to whether the oligomeric state of CM1 in the Spc721-599.3R constructs and whether the 3R mutant's effect is due to its disruption of dimerization.

- 1.1 While the SEC and mass photometry data provided to establish the assembly state of the CM1 3R mutant supports the authors' claim that the 3R mutant disrupts dimerization, the presence of a tetramer peak in the GST-CM1 3R mutant mass photometry data (Supp. Fig. 9g) suggests that the CM1 helices in the 3R mutant may still dimerize even at relatively low concentrations. It seems likely the CM1 helices in the 3R mutants might still be dimerized at the high local concentration expected in the constructs used in this study including the C-terminal coiled-coils in Spc72. Unfortunately, the authors also show in Figure 3R that the CM1 dimer alone cannot induce γ -TuSC assembly, making it difficult to directly probe the role of CM1 dimerization using the MBP constructs generated. As there is some remaining ambiguity about whether CM1 is dimerized in the Spc721-599.3R constructs, I might suggest a more direct

method such as using EPR or attaching FRET probes to the CM1 helices to more directly probe the structural state of the CM1 helices in the Spc721-599 WT and 3R constructs. A related experiment that may provide useful data is to examine the affinity of the MBP-tagged CM1 constructs for the γ -TuSC.

The data we presented in the revised version of our manuscript establish that the 3R mutant strongly affects CM1 dimerization, which impairs high-affinity γ -TuSC binding and has functional consequences *in vitro* and *in vivo*. However, we cannot entirely exclude a low level of CM1 dimerization even in the 3R mutant and therefore revised our manuscript to reflect this:

“A low level of tetramer formation was also observed in case of GST-CM1^{231-268,3R}, which may suggest that CM1 motif dimerisation was not completely abolished by the 3R mutation under these experimental conditions.”

1.2 As the Spc72.1-599.3R mutants' CM1 helix may still be dimerized, it remains unclear whether the 3R mutant's inability to stimulate γ -TuSC assembly is due to a disruption of CM1 dimerization, diminished affinity for γ -TuSC, or some other effect. The authors could target the CM1out interacting residues for mutagenesis to further probe whether the main contribution of CM1 dimerization in this system is to increase the affinity for the complex with these additional contacts, but this would not address the proposed role of structural rigidity induced by CM1 dimerization in the “wedge” model cited by the authors.

We are convinced that the data we presented in the revised version of our manuscript establish strongly reduced CM1 dimerization levels for the 3R mutant. Nevertheless, to address this comment, we edited the manuscript text to reflect that alternative mechanisms may contribute to the effects observed.

“While we cannot exclude entirely that alternative defects induced by the 3R mutant may contribute to the observed effects, this likely suggests that dimerisation of the Spc72 CM1 motif plays a central role for establishing high affinity binding to γ -TuSCs, which is essential for the formation of γ -TuSC oligomers in *C. albicans*.”

1.3 In Figure 4e, the WT CM1 binding curve appears very similar to the 3R CM1 binding curve, with both appearing to show a midpoint in the rise of F_{norm} slightly above 1000 nM. The error bars in the WT CM1 appear to be similar in size during its rise to the observed dynamic range of the curve, suggesting there may be significant uncertainty in the concentration measurement that is not reflected in the affinity measurement provided at the top of the panel. Additional information may be helpful in establishing the accuracy of the binding curves. As the GST-CM1 3R construct still forms dimers, is it possible the observed affinity in these plots is still due to CM1 dimers binding to γ -TuSC?

Similar to above, we are convinced that the data we presented in the revised version of our manuscript clearly establish strongly reduced CM1 dimerization levels for the 3R mutant.

However, since we cannot entirely exclude a low level of CM1 dimerization even in the 3R mutant, binding of dimerized GST-CM1^{231-268,3R} to γ -TuSCs may contribute to the binding curve of GST-CM1^{231-268,3R}. In fact, this would further strengthen the importance of CM1 dimerization for high-affinity binding of CM1 to γ -TuSCs. We modified the manuscript accordingly:

“Notably, a low level of CM1^{231-268,3R} dimerization may contribute to the remaining γ -TuSC binding observed for GST-CM1^{231-268,3R}.”

1.4 In the western blot shown in figure 4h, γ -TuSC appears to co-elute with Stu2 and Spc72.1-599.3R. The trace shows no peak consistent with the peak at 12 ml that the authors attribute to γ -TuSC/Spc72.1-599/Stu2. This suggests that the 3R mutant’s effect may not be well characterized, or that the peaks may be misattributed in the revised manuscript.

We apologize for the ambiguity of the Western blot in Fig. 4h. The elution volumes of monomeric γ -TuSC (Supplementary Fig. 1e) and free Spc72^{1-599,3R} are very similar (Fig. R1), which may give the impression that both components elute as a complex in Fig. 4h, although they most likely predominantly represent separate components eluting individually. To avoid misinterpretation of the Western blot, we now included Fig. R1b into Fig. 4h.

Moreover, to avoid confusion, we removed the negative stain 2D class average of the monomeric γ -TuSC/His-Spc72¹⁻⁵⁹⁹/Stu2 complex in Fig. 4h that was originally used to illustrate in which fractions the complex elutes when co-expressed with **wild-type His-Spc72¹⁻⁵⁹⁹**.

Figure R1. Elution volume of isolated His-Spc72^{1-599,3R}. **a)** Visualization of the protein content in fractions from a SEC profile of γ -TuSC/His-Spc72^{1-599,3R}/FLAG-Stu2 shown in Fig. 4g by Western blotting. This panel corresponds to Fig. 4h of the revised manuscript. **b)** Visualization of the protein content in fractions from a SEC profile of isolated His-Spc72^{1-599,3R} by Western blotting.

Overall, if the authors are unable to better establish the monomer state of the CM1 helix in their Spc72.1-599.3R constructs, I would suggest that the authors reduce their claims on the importance of dimerization.

While we are convinced that our data clearly establish strongly reduced CM1 dimerization levels for the 3R mutant, we cannot entirely exclude that low levels of CM1 dimerization remain in the 3R mutant. Thus, we have edited the manuscript at multiple instances highlighted in the revised manuscript version to tone down our conclusions, as requested.

2. The additional experiments performed by the authors provide important additional validation of the Stu2/Spc72 interfaces. However, the interfaces may not be mapped precisely enough to be described as validated at the residue level:

2.1 In figure S10d, the authors show that the Stu2 LIM mutant and the SPC72 EDID mutants both have strong effects on assembly with the combination appearing to completely abolish it. However, the Spc72 ELLY mutant displays a much weaker effect on assembly, indicating the structural prediction for this interaction site may not be accurate.

A likely explanation for the observed effects in case of the Spc72 ELLY mutant is that the interface involves residues in addition to those mutated. We now mention this in the result section:

“However, the weak interaction remaining for the Spc72³⁰⁰⁻³⁵⁰ ELLY mutant may suggest that the secondary binding site involves residues additional to those mutated.”

2.2 While both the Stu2 LIM mutant and the SPC72 EDID mutants have strong effects on assembly, the additive effect observed when both sets of mutations are present does not establish that these residues interact with each other. This leaves the register of the interaction in question. This also applies to the ELLY mutants.

We thank the Reviewer for pointing this out. Our experiment clearly establishes that the Stu2 LIM mutant abolishes binding of WT Spc72⁴³⁰⁻⁴⁸⁰ (lane 8); similarly the Spc72⁴³⁰⁻⁴⁸⁰ EDID mutant strongly reduces binding to WT Stu2 (lane 3). The very weak band in lane 3 is within the error range of the experiment. When both sets of mutations are combined (lane 7), binding is again abolished. All combinations of mutations strongly reduce binding of Stu2 to Spc72⁴³⁰⁻⁴⁸⁰. We believe that conclusions going beyond cannot be drawn from this experiment.

2.3 The HDX data appears quite noisy, and only shows a strong effect for a small portion of the N-terminal weaker binding site.

Observing strong protection only for a section of the secondary Stu2 binding site is most likely related to the peptide-level resolution of HX-MS, which reports on protection averaged over the total number of residues in a measured peptide. All Spc72 peptides appearing strongly protected in the primary and secondary Stu2 binding sites are fully encompassed in the predicted binding

sites (Supplementary Fig. 13d,e). In contrast, the seemingly ‘non-protected’ portion of the secondary Stu2 binding site that the Reviewer refers to is part of a comparably long Spc72 peptide of which less than half is covered by the predicted binding site (Supplementary Fig. 13d,e). Thus, any protection arising from Stu2 binding is likely to be averaged out over the length of the peptide, explaining the low apparent protection observed for this part of the binding site.

I would suggest the authors tone down the language on the interface being mapped at the residue level. I would also suggest that if the authors would like to include a discussion of the electrostatic interactions in the interfaces, it be made clear that this remains speculative due to the relatively low “resolution” of the data provided.

While we are highly confident regarding the interfaces involved in the Stu2-Spc72 interactions based on our experiments, we acknowledge the Reviewer’s concerns and removed any claims of having identified the interfaces at residue level from the manuscript, as requested.

Regarding the electrostatic interactions predicted to be involved in the Stu2-Spc72 interfaces, we confirmed that in all instances it has been made clear that these interactions are purely based on an AlphaFold2 structure prediction.

3. The panels showing density and the fitted models in Figures 6 and 9 all appear to be relatively tightly masked via segmentation. While the quality of the structures presented in this manuscript is visually consistent with the quoted resolutions, it may be easier for the reader to assess their quality if the structures are less tightly masked, so the noise level present can be readily assessed without examining the deposited data. I would suggest some of the panels in these two figures be unmasked and unsegmented, with only clip planes used to focus on the region of interest.

We believe that the Reviewer refers to Supplementary Figures 6 and 8, in which we show segmented cryo-EM densities to visualize the overlap between atomic models and cryo-EM densities. To generate these figure panels, cryo-EM densities were segmented in ChimeraX with a segmentation radius of 5 Å, which is sufficiently wide to faithfully preserve the shape of densities at the resolution range of the cryo-EM densities shown.

However, to address this comment, we have updated the figure panels as requested by the Reviewer. Specifically, we now provide views of atomic models fitted into unmasked and unsegmented cryo-EM densities wherever possible without compromising visual clarity.

Minor points:

In the overlay in Figure 4c, the authors note that the structures were superposed based on an alignment of Spc97 with GCP2.473-895. I wonder if this may cause the alignment to report more on differences in the conformations of Spc97/GCP2 than on differences in the CM1 motifs in the structures. Perhaps an alignment based on the CM1 motifs and the surrounding conserved interaction regions may yield some additional insights?

As suggested, we superposed the two atomic models shown in Fig. 4c based only on the structurally conserved interaction elements of GCP2/Spc97. The outcome was very similar to our original superposition according to GCP2 residues 473-895, suggesting that any conformational differences between GCP2 and Spc97 seem to have little impact on the CM1-coordinating segments and the relative positioning of CM1 motifs.

Following the Reviewer's suggestion, we updated Fig. 4c with the superposition according to the structurally conserved interaction elements of GCP2/Spc97.

Similarly to the point above, would the panel Supp. Fig 6e report more directly on differences or similarities in the CM1 motifs in *S. cerevisiae* Spc110 and *C. albicans* Spc72 if the alignment were focused on the CM1 helix and surrounding interaction regions? Such an alignment may also yield additional insights into differences in regulation between the systems.

The binding modes of monomeric CM1 in *S. cerevisiae* Spc110 and dimeric CM1 in *C. albicans* Spc72 are vastly different (Supplementary Fig. 6e). Consistently, many of the interactions towards Spc97 are not conserved between *S. cerevisiae* and *C. albicans*. This makes a superposition of the two models shown in Supplementary Fig. 6e only according to the CM1 motif and its interaction elements on Spc97 very challenging. Since the overall structure of γ -TuSCs in the closed conformation is very similar in *S. cerevisiae* and *C. albicans* (Supplementary Fig. 5a), we believe that superposition of the two models according to the full γ -TuSC as reference system is more robust and we therefore opted to keep the original mode of superposition in Supplementary Fig. 6e.

During the previous round of review (point 4.4), I raised a question on the authors' claim on the importance of the C-terminal coiled coils in *S. cerevisiae* Spc110 in compensating for the monomeric state of its CM1 helix. After reading their rebuttal, I now agree with their assessment that the C-terminal coiled coils present in Spc110, as well as the loop between the CM1 motif and these coiled-coils may help to increase the affinity of the Spc110 for the γ -TuSC due to the additional contacts formed and thereby compensate for the loss of the interactions between CM1_{out} and the γ -TuSC observed in the Spc72-bound γ -TuSC structure the authors present in this manuscript. I would support adding the text back to the manuscript.

We thank the Reviewer for this suggestion. We have added the text back into the manuscript and integrated some aspects mentioned by the Reviewer:

"Notably, *S. cerevisiae* Spc110, the only protein for which monomeric CM1 motif binding was observed, contains a loop and a specific coiled-coil region C-terminal to the CM1 motif, both of which bind to Spc97 GRIP1¹⁰ and may thus compensate for the loss of CM1_{out} in high-affinity γ -TuSC binding."

At low thresholds, the raw unsharpened data for the γ -TuRC ring shows low resolution density that resembles an eighth γ -TuSC subunit. Could the authors comment on whether this is due to some rotational misalignment, or a subpopulation present in their purification?

We agree that the low-resolution density visible at very low threshold levels most likely corresponds to a small fraction of γ -TuRC particles for which the particle alignment is shifted in register by one γ -TuSC unit. Notably, these particles still contribute to the overall signal of the cryo-EM density due to the inherent symmetry of the γ -TuRC. Such low-resolution density can be frequently observed in reconstructions of γ -TuRCs in the closed conformation, e.g. in EMDB-18182, EMDB-43482, EMD-43483 and EMDB-18665.

In the sentence “CM1-mediated dimerisation of dimers induced tetramer formation predominantly in case of MBP-CM1231-268”, I believe the authors intend to refer to the GST-CM1 construct.

We corrected as suggested.

Reviewer #2 (Remarks to the Author):

The authors have conducted extensive experiments and revisions that satisfy my questions and concerns. The manuscript warrants publication in Nature Communications.

We are very pleased that Reviewer #2 does not raise any further concerns and supports publication of our manuscript in *Nature Communications*.

Reviewer #3 (Remarks to the Author):

The authors have addressed my concerns, and the manuscript warrants publication in Nature Communications.

We are very pleased that Reviewer #3 does not raise any further concerns and supports publication of our manuscript in *Nature Communications*.

Point-by-Point Reply to Reviewers

There were no Reviewer comments left to address.